# Anchor Sampling for Federated Learning with Partial Client Participation

## Abstract

In federated learning, the support of partial client participation offers a flexible training strategy, but it deteriorates the model training efficiency. In this paper, we propose a framework FedAMD to improve the convergence property and maintain flexibility. The core idea is anchor sampling, which disjoints the partial participants into anchor and miner groups. Each client in the anchor group aims at the local bullseye with the gradient computation using a large batch. Guided by the bullseyes, clients in the miner group steer multiple near-optimal local updates using small batches and update the global model. With the joint efforts from both groups, FedAMD is able to accelerate the training process as well as improve the model performance. Measured by $\epsilon$-approximation and compared to the state-of-the-art first-order methods, FedAMD achieves the convergence by up to $O(1/\epsilon)$ fewer communication rounds under non-convex objectives. In specific, we achieve a linear convergence rate under PL conditions. Empirical studies on real-world datasets validate the effectiveness of FedAMD and demonstrate the superiority of our proposed algorithm: Not only does it considerably save computation and communication costs, but also the test accuracy significantly improves.

## 1 Introduction

Federated learning (FL) (Konečný et al., 2015; 2016; McMahan et al., 2017) has attained an increasing interest over the past few years. As a distributed training paradigm, it enables a group of clients to collaboratively train a global model from decentralized data under the orchestration of a central server. By this means, sensitive privacy is basically protected because the raw data are not shared across the clients. Due to the unreliable network connection and the rapid proliferation of FL clients, it is infeasible to require all clients to be simultaneously involved in the training. To address the issue, recent works (Li et al., 2019b; Philippenko & Dieuleveut, 2020; Gorbunov et al., 2021a; Karimireddy et al., 2020b; Yang et al., 2020; Li et al., 2020; Eichner et al., 2019; Yan et al., 2020; Ruan et al., 2021; Gu et al., 2021; Lai et al., 2021) introduce a practical setting where merely a portion of clients participates in the training. The partial-client scenario effectively avoids the network congestion at the FL server and significantly shortens the idle time as compared to traditional large-scale machine learning (Zinkevich et al., 2010; Bottou, 2010; Dean et al., 2012; Bottou et al., 2018).

However, a model trained with partial client participation is much worse than the one trained with full client participation (Yang et al., 2020). This phenomenon is account for two reasons, namely, data heterogeneity (a.k.a. non-i.i.d. data) and the lack of inactive clients' updates. With data heterogeneity, the optimal model is subject to the local data distribution, and therefore, the local updates on the clients' models greatly deviate from the update towards optimal global parameters (Karimireddy et al., 2020b; Malinovskiy et al., 2020; Pathak & Wainwright, 2020; Wang et al., 2020; 2021; Mitra et al., 2021; Rothchild et al., 2020; Zhao et al., 2018; Wu et al., 2021). FedAvg (McMahan et al., 2017; Li et al., 2019b; Yu et al., 2019a;b; Stich, 2018), for example, is less likely to follow a correct update towards the global minimizer because the model aggregation on the active clients critically deviates from the aggregation on the full clients, an expected direction towards global minimizer (Yang et al., 2020).

As a family of practical solutions to data heterogeneity, variance reduced techniques (Karimireddy et al., 2020b; Gorbunov et al., 2021a; Wu et al., 2021; Gorbunov et al., 2021b; Liang et al., 2019; Shamir et al., 2014; Li et al., 2019a; 2021b; Karimireddy et al., 2020a; Murata & Suzuki, 2021) achieve an improved convergence rate when compared to FedAvg. With multiple local updates, each client corrects the SGD steps with reference to an estimated global target, which is synchronized at

| Convexity | Method | Partial Clients | Communication Rounds |
|---|---|---|---|
| Non-convex | Minibatch SGD (Wang & Srebro, 2019) | ✗ | $\frac{1}{MK\epsilon^2} + \frac{1}{\epsilon}$ |
| | FedAvg (Yang et al., 2020) | ✓ | $\frac{K}{A\epsilon^2} + \frac{1}{\epsilon}$ |
| | SCAFFOLD (Karimireddy et al., 2020b) | ✓ | $\frac{\sigma^2}{AK\epsilon^2} + \left(\frac{M}{A}\right)^{2/3}\frac{1}{\epsilon}$ |
| | BVR-L-SGD (Murata & Suzuki, 2021) | ✗ | $\frac{1}{MK\epsilon^{3/2}} + \frac{1}{\epsilon}$ |
| | VR-MARINA(Gorbunov et al., 2021a) | ✗ | $\frac{\sigma}{M\epsilon^{3/2}} + \frac{\sigma^2}{M\epsilon} + \frac{1}{\epsilon}$ |
| | FedAMD (Sequential) (Corollary 1) | ✓ | $\frac{M}{A\epsilon}$ |
| | FedAMD (Constant) (Corollary 2) | ✓ | $\frac{M}{A\epsilon}$ |
| PL condition (or *strongly-convex) | Minibatch SGD* (Woodworth et al., 2020b) | ✗ | $\frac{\sigma^2}{\mu MK\epsilon} + \frac{1}{\mu}\log\frac{1}{\mu\epsilon}$ |
| | FedAvg (Karimireddy et al., 2020a) | ✓ | $\frac{1+\sigma^2/K}{\mu A\epsilon} + \frac{\sqrt{1+\sigma^2/K}}{\mu\sqrt{\epsilon}} + \frac{1}{\mu}\log\frac{1}{\epsilon}$ |
| | SCAFFOLD* (Karimireddy et al., 2020b) | ✓ | $\frac{\sigma^2}{\mu AK\epsilon} + \left(\frac{M}{A} + \frac{1}{\mu}\right)\log\frac{M\mu}{A\epsilon}$ |
| | VR-MARINA (Gorbunov et al., 2021a) | ✗ | $\left(\frac{\sigma^2}{\mu M\epsilon} + \frac{\sigma}{\mu^{3/2}M\sqrt{\epsilon}} + \frac{1}{\mu}\right)\log\frac{1}{\epsilon}$ |
| | FedAMD (Constant) (Corollary 3) | ✓ | $\left(\frac{1}{\mu} + \frac{M}{\mu^2 A} + \frac{M}{A}\right)\log\frac{1}{\epsilon}$ |

Table 1: Number of communication rounds that achieve $\mathbb{E}\left\|\nabla F\left(\tilde{x}_{out}\right)\right\|_2^2 \le \epsilon$ for non-convex objectives (or $\mathbb{E}F(\tilde{x}_{out}) - F_* \le \epsilon$ for PL condition or strongly-convex with the parameter of $\mu$). We optimize an online scenario and set the small batch size to 1. The symbol ✓ or ✗ for "Partial Clients" is determined by the following footnote 1.

the beginning of every round. Although, in each transmission round, variance-reduced algorithms require the communication overhead twice as more as FedAvg, their improved performances are likely to eliminate the cost increments. Recent studies (Gorbunov et al., 2021a; Murata & Suzuki, 2021; Tyurin & Richtárik, 2022; Zhao et al., 2021) have demonstrated great potential using large batches under full client participation[1]. Measured by $\epsilon$-approximation, MARINA (Gorbunov et al., 2021a), for instance, realizes $O(1/M\epsilon^{1/2})$ faster while using large batches, where $M$ indicates the number of clients.

However, none of the prior studies address the drawbacks of using large batches. Typically, a large batch update involves several gradient computations compared to a small batch update. This increases the burden of FL clients, especially on IoT devices like smartphones, because their hardware hardly accommodates all samples in a large batch simultaneously. Instead, they must partition the large batch into several small batches to obtain the final gradient. Furthermore, regarding the critical convergent differences between various participation modes, the effect of using large batches under partial client participation cannot be affirmative. BVR-L-SGD (Murata & Suzuki, 2021) and FedPAGE (Zhao et al., 2021) claim that they can work under partial client participation, but they require all clients' participation when the algorithms come to the synchronization using a large batch.

Motivated by the observation above, we propose a framework named FedAMD under federated learning with anchor sampling that disjoints the partial participants into two groups, i.e., anchor and miner group. In the anchor group, clients (a.k.a. anchors) compute the gradient using a large batch cached in the server to estimate the global orientation. In the miner group, clients (a.k.a. miners) perform multiple updates corrected according to the previous and the current local parameters and the last local update volume. The objective for the latter group is twofold. First, multiple local updates without serious deviation can effectively accelerate the training process. Second, we update the global model using the local models from the latter group only. Since anchor sampling disjoints the clients with time-varying probability, we separately consider constant and sequential probability settings.

**Contributions.** We summarize our contributions as follows:

- **Algorithmically**, we propose a unified federated learning framework FedAMD that identifies a participant as an anchor or a miner. Clients in the anchor group aim to obtain the bullseyes of their local data with a large batch, while the miners target to accelerate the training with multiple local updates using small batches.

- **Theoretically**, we establish the convergence rate for FedAMD under non-convex objectives under both constant and sequential probability settings. To the best of our knowledge, this is the first work to analyze the effectiveness of large batches under partial client participation. Our theoretical

---

[1]In this paper, partial client participation refers to the case where only a portion of clients take part at every round during the entire training.

results indicate that, with the proper setting for the probability, FedAMD can achieve a convergence rate of $O(\frac{M}{AT})$ under non-convex objective, and linear convergence under Polyak-Łojasiewicz (PL) condition (Polyak, 1963; Lojasiewicz, 1963). Comprehensive comparisons with previous works are presented in Table 1.

- **Empirically**, we conduct extensive experiments to compare FedAMD with the most representative approaches. The numerical results provide evidence of the superiority of our proposed algorithm. Achieving the same test accuracy, FedAMD utilizes less computational power metered by the cumulative gradient complexity.

## 2 RELATED WORK

In this section, we discuss the state-of-the-art works that are strongly relevant to our research. A more comprehensive review is provided in Appendix A.

**Mini-batch SGD vs. Local SGD.** Distributed optimization is required to train large-scale deep learning systems. Local SGD (also known as FedAvg) (Stich, 2018; Dieuleveut & Patel, 2019; Haddadpour et al., 2019; Haddadpour & Mahdavi, 2019) performs multiple (i.e., $K \geq 1$) local updates with $K$ small batches, while mini-batch SGD computes the gradients averaged by $K$ small batches (Woodworth et al., 2020b;a) (or a large batches (Shallue et al., 2019; You et al., 2018; Goyal et al., 2017)) on a given model. There has been a long discussion on which one is better (Lin et al., 2019; Woodworth et al., 2020a;b; Yun et al., 2021), but no existing work considers how to disjoint the nodes such that both can be trained at the same time.

**Variance Reduction in FL.** The variance-reduced techniques have critically driven the advent of FL algorithms (Karimireddy et al., 2020b; Wu et al., 2021; Liang et al., 2019; Karimireddy et al., 2020a; Murata & Suzuki, 2021; Mitra et al., 2021) by correcting each local computed gradient with respect to the estimated global orientation. However, a concern is addressed on how to attain an accurate global orientation to mitigate the update drift from the global model. Roughly, the estimation lies in two types, namely, precalculated and cached. The former methods (Murata & Suzuki, 2021; Mitra et al., 2021) required precalculation typically require full worker participation, which is infeasible for federated learning settings. As for the global orientation estimated by cached information, existing approaches (Karimireddy et al., 2020b; Wu et al., 2021; Liang et al., 2019; Karimireddy et al., 2020a) utilize small batches, which derives a biased estimation and misleads the training. This work explores the effectiveness of large-batch estimation for the global orientation under partial client participation.

## 3 FEDAMD

In this section, we comprehensively describe the technical details of FedAMD, a federated learning framework with anchor sampling. In specific, it disjoints the active participants into the anchor group and the miner group with time-varying probabilities. The pseudo-code is illustrated in Algorithm 1.

**Problem Formulation.** In an FL system with a total of $M$ clients, the objective function is formalized as

$$\min_{\mathbf{x} \in \mathbb{R}^d} F(\mathbf{x}) = \frac{1}{M} \sum_{m \in [M]} F_m(\mathbf{x}) \tag{1}$$

where we define $[M]$ for a set of $M$ clients. $F_m(\cdot)$ indicates the local expected loss function for client $m$, which is unbiased estimated by empirical loss $f_m(\cdot)$ using a random realization $\mathcal{B}_m$ from the local training data $\mathcal{D}_m$, i.e., $\mathbb{E}_{\mathcal{B}_m \sim \mathcal{D}_m} f_m(\mathbf{x}, \mathcal{B}_m) = F_m(\mathbf{x})$. We denote $n$ by the size of a client's local dataset, i.e., $|\mathcal{D}_m| = n$ for all $m \in [M]$, and $n$ can be infinite large in the streaming/online cases. $F_*$ represents the minimum loss for Equation (1).

**Algorithm Description.** In FedAMD, a global model is initialized with arbitrary parameters $\tilde{x}_0 \in \mathbb{R}^d$. By distributing the model to all clients (Line 1), clients $m \in [M]$ are required to generate a $b$-sample batch $\mathcal{B}_{m,0}$ and compute the gradient $v_0^{(m)}$ (Line 3). Then, clients send $v_0^{(m)}$ to the server (Line 4), and server caches these gradients and span them as a matrix $v_0 = \left\{ v_0^{(m)} \right\}_{m \in [M]}$ (Line 6).

After the initialization steps above, the algorithm comes to the model training (Line 7–27). At the beginning of each round $t$, the server randomly picks an $A$-client subset $\mathcal{A}$ from $M$ clients (Line 8). Since each client is independently selected without replacement, under the setting of Equation (1), clients have an equal chance to be selected with the probability of $\frac{A}{M}$. Subsequently, the server

---

**Algorithm 1** FedAMD

---

**Input:** local learning rate $\eta_l$, global learning rate $\eta_s$, minibatch size $b$, $b' < b$, local updates $K$, probability $\{p_t \in [0,1]\}_{t \geq 0}$, initial model $\tilde{x}_0$.

1: Communicate the initial model $\tilde{x}_0$ with all clients $m \in [M]$
2: **for** $m \in [M]$ **in parallel do**
3:      $v_0^{(m)} = \nabla f_m(\tilde{x}_0, \mathcal{B}_{m,0})$ using $\mathcal{B}_{m,0} \sim \mathcal{D}_m$ with the size of $b$
4:      Communicate $v_0^{(m)}$ with the server
5: **end for**
6: Initialize caching gradient $v_0 = \left\{ v_0^{(m)} \right\}_{m \in [M]}$
7: **for** $t = 0, 1, 2, \ldots$ **do**
8:      Sample clients $\mathcal{A} \subseteq [M]$
9:      Communicate the model $\tilde{x}_t$ and the caching gradient $\tilde{g}_t = avg(v_t)$ with clients $i \in \mathcal{A}$
10:      Initialize subsequent caching gradient $v_{t+1} = v_t$
11:      **for** $i \in \mathcal{A}$ **in parallel do**
12:          **if** $Bernoulli(p_t) == 1$ **then**
13:              $v_{t+1}^{(i)} = \nabla f_i(\tilde{x}_t, \mathcal{B}_{i,t})$ using $\mathcal{B}_{i,t} \sim \mathcal{D}_i$ with the size of $b$
14:              Communicate $v_{t+1}^{(i)}$ with the server and indicate the update of caching gradient
15:          **else**
16:              Initialize $x_{t,-1}^{(i)} = x_{t,0}^{(i)} = \tilde{x}_t$, $g_{t,0}^{(i)} = \tilde{g}_t$
17:              **for** $k = 0, \ldots, K-1$ **do**
18:                  Generate random realization $\mathcal{B}'_{i,k} \sim \mathcal{D}_i$ with the size of $b'$
19:                  $g_{t,k+1}^{(i)} = g_{t,k}^{(i)} - \nabla f_i\left(x_{t,k-1}^{(i)}, \mathcal{B}'_{i,k}\right) + \nabla f_i\left(x_{t,k}^{(i)}, \mathcal{B}'_{i,k}\right)$
20:                  $x_{t,k+1}^{(i)} = x_{t,k}^{(i)} - \eta_l \cdot g_{t,k+1}^{(i)}$
21:              **end for**
22:              $\Delta x_t^{(i)} = \tilde{x}_t - x_{t,K}^{(i)}$
23:              Communicate $\Delta x_t^{(i)}$ with the server and indicate the update of the model
24:          **end if**
25:      **end for**
26:      $\tilde{x}_{t+1} = \tilde{x}_t - \eta_s \cdot avg(\Delta x_t)$ where $\Delta x_t$ aggregates $\Delta x_t^{(i)}$ where client $i$ updates model
27: **end for**

---

distributes the global model $\tilde{x}_t$ to the clients in the set $\mathcal{A}$, accompanying the global bullseye (i.e., the averaged caching gradient) $\tilde{g}_t = \frac{1}{M} \sum_{m \in [M]} v_0^{(m)}$ (Line 9). With the probability of $p_t$, client $i \in \mathcal{A}$ is classified for the anchor group (Line 13–14) or the miner group (Line 16–23), and different groups have different objectives and focus on different tasks.

**Anchor group (Line 13–14).** Clients in this group target to discover the bullseyes based on their local data distribution. According to Line 12, client $i \in \mathcal{A}$ has the probability of $p_t$ to become a member of this group. Then, the client utilizes a large batch $\mathcal{B}_{i,t}$ with $b$ samples to obtain the gradient $v_{t+1}^{(i)}$ (Line 13). Therefore, following the gradient $v_{t+1}^{(i)}$ can find an optimal or near-optimal solution for client $i$. Next, the client pushes the gradient to the server and updates the caching gradient (Line 14). In view that some clients do not participate in the anchor group for obtaining the bullseyes at round $t$, the server spontaneously inherits their previous calculation from $v_t$ (Line 10). As a result, $\tilde{g}_t$ in Line 9 indicates an approximate orientation towards global optimal parameters, which directs the local update in the miner group and affects the final global update. Besides, $v_{t+1}^{(i)}$ influences the training from round $t+1$ up to the next time when client $i$ is a member of anchor group.

**Miner group (Line 16–23).** Guided by the global bullseye, clients in the miner group perform multiple local updates and finally drive the update of the global model. First, client $i$ initializes the model with $\tilde{x}_t$ and the target direction with $\tilde{g}_t$ (Line 16). Ideally, in the subsequent $K$ updates (Line 17), client $i$ update the model with the gradient $\nabla F(x_{t,k}^{(i)})$ for $k \in \{0, \ldots, K-1\}$. This is impractical because clients cannot access all others' training sets to compute the noise-free gradients. Instead, the client at $k$-th iteration generates a $b'$-sample realization $\mathcal{B}'_{i,k}$ (Line 18) and calculates the update $g_{t,k+1}^{(i)}$ via a variance-reduced technique, i.e., $g_{t,k+1}^{(i)} = g_{t,k}^{(i)} - \nabla f_i\left(x_{t,k-1}^{(i)}, \mathcal{B}'_{i,k}\right) + \nabla f_i\left(x_{t,k}^{(i)}, \mathcal{B}'_{i,k}\right)$

(Line 19). The update $g_{t,k+1}^{(i)}$ is approximate to $\nabla F(x_{t,k}^{(i)})$ for two reasons: (i) the first term is used to estimate the global update because $g_{t,0}^{(i)}$ stores the global bullseye; and (ii) the rest terms remove the perturbation of data heterogeneity and reflect the true update at $x_{t,k}^{(i)}$. Therefore, the local model update follows $x_{t,k+1}^{(i)} = x_{t,k}^{(i)} - \eta_l \cdot g_{t,k+1}^{(i)}$ (Line 20). After $K$ local updates, the model changes on client $i$ is $\Delta x_t^{(i)} = \tilde{x}_t - x_{t,K}^{(i)}$. Then, the client transmits $\Delta x_t^{(i)}$ to the server for the purpose of global model update.

The proposed approach possesses threefold advantages when compared to SCAFFOLD (Karimireddy et al., 2020b) and FedLin (Mitra et al., 2021) using a consistent correction term, i.e., $\tilde{g}_t - v_t^{(i)}$. Firstly, it is a memory-efficient approach that is unnecessary to maintain the obsolete gradient. Secondly, it dynamically calibrates the local updates subject to the local model. Although BVR-L-SGD (Murata & Suzuki, 2021) also achieves such a functionality, it requires all clients to jointly obtain the global direction at the beginning of each round, leading to considerable training time. As for FedAMD, here comes the third advantage that avoids the precalculation on a global bullseye under partial-client scenarios. To the best of our knowledge, this is the first work to achieve dynamic calibration under partial-client scenarios.

**Server (Line 14 and Line 26).** Therefore, after the separate local training on the participants, the server merges the model changes from the miner group into $\Delta x_t$ (Line 26) and updates the caching gradients from the anchor group (Line 14). It is noted that the size of $\Delta x_t$ (a.k.a. $|\Delta x_t|$) can be within the range between 0 and $A$. When the size is 0, $\tilde{x}_{t+1} = \tilde{x}_t$, or otherwise, $\tilde{x}_{t+1} = \tilde{x}_t - \eta_s \sum \Delta x_t / |\Delta x_t|$ (Line 26). The reason why we solely use the changes from the miner group is that clients perform multiple local updates regulated by the global target such that the model changes walk towards the global optimal solution. While directly incorporating the new gradients from the anchor group, the global model has a degraded performance because they perform a single update that aims to find out the local bullseye deviated from the global target. Implicitly, clients in the miner group take in the update of caching gradients at iteration $k = 0$ to update the local model, which will affect the next global parameters.

**Previous Algorithms as Special Cases.** The probabilities can vary among the rounds that disjoint the participants into the anchor group and the miner group. By setting $A = M$, and the probability $\{p_t\}$ following the sequence of $\{1, 0, 1, 0, \dots\}$, FedAMD reduces to distributed minibatch SGD ($K = 1$) or BVR-L-SGD ($K > 1$). Therefore, FedAMD subsumes the existing algorithms and takes partial client participation into consideration. To obtain the best performance, we should tune the settings of $\{p_t\}$ and $K$. However, accounting for the generality of FedAMD, it faces substantial additional hurdles in its convergence analysis, which is one of our main contributions, as detailed in the following section.

**Discussion on Communication Overhead.** As the anchors are not necessary to obtain the averaged caching gradient (i.e., $\tilde{g}_t$) at $t$-th round, the centralized server solely distributes $\tilde{g}_t$ to the miners. Compared to FedAvg, the proposed algorithm requires $(1 - p_t)/2$ more communication costs, but it achieves convergence with at least $O(\frac{1}{\epsilon})$ less communication rounds (see Table 1). Therefore, from the perspective of model training progress, FedAMD is more communication efficient than FedAvg.

**Discussion on Massive-Client Settings.** A typical example of this scenario is cross-device FL (Kairouz et al., 2019). In this setting, it is not a wise option for the server to preserve all the caching gradients for clients. Therefore, the clients retain their caching gradients while the server keeps their average. Firstly, at $t$-th round, client $i \in [M]$ copies their caching gradient to $(t + 1)$-th round, i.e., $v_{t+1}^{(i)} = v_t^{(i)}$. For the client $i$ in the anchor group, they will follow Line 13 in Algorithm 1 to update $v_{t+1}^{(i)}$ and push $\wedge_t^{(i)} = v_{t+1}^{(i)} - v_t^{(i)}$ to the server. After the server receives the updates of all local caching gradients, it performs $v_{t+1} = v_t + \frac{1}{M} \sum \wedge_t$, where $\wedge_t$ aggregates $\wedge_t^{(i)}$ where client $i$ is in anchor group.

## 4 THEORETICAL ANALYSIS

In this section, we analyze the convergence rate of FedAMD under non-convex objectives with respect to $\epsilon$-approximation, i.e., $\min_{t \in [T]} \|\nabla F(\tilde{x}_t)\|_2^2 \le \epsilon$. Specifically, when it comes to PL condition, $\epsilon$-approximation refers to $F(\tilde{x}_T) - F(x_*) \le \epsilon$. In the following discussion, we particularly highlight

the setting of $\{p_t\}$ to obtain the best performance. Before showing the convergence result, we make the following assumptions, where the first two assumptions have been widely used in machine learning studies (Karimireddy et al., 2020b; Li et al., 2020), while the last one has been adopted in some recent works (Gorbunov et al., 2021a; Tyurin & Richtárik, 2022; Murata & Suzuki, 2021).

**Assumption 1** (L-smooth). *The local objective functions are Lipschitz smooth: For all $v, \bar{v} \in \mathbb{R}^d$,*

$$\|\nabla F_i(v) - \nabla F_i(\bar{v})\|_2 \leq L\|v - \bar{v}\|_2, \quad \forall i \in [M].$$

**Assumption 2** (Bounded Noise). *For all $v \in \mathbb{R}^d$, there exists a scalar $\sigma \geq 0$ such that*

$$\mathbb{E}_{\mathcal{B} \sim \mathcal{D}_i}\|\nabla f_i(v, \mathcal{B}) - \nabla F_i(v)\|_2^2 \leq \frac{\sigma^2}{|\mathcal{B}|}, \quad \forall i \in [M].$$

**Assumption 3** (Average L-smooth). *For all $v, \bar{v} \in \mathbb{R}^d$, there exists a scalar $L_\sigma \geq 0$ such that*

$$\mathbb{E}_{\mathcal{B} \sim \mathcal{D}_i}\left\|(\nabla f_i(v, \mathcal{B}) - \nabla f_i(\bar{v}, \mathcal{B})) - (\nabla F_i(v) - \nabla F_i(\bar{v}))\right\|_2^2 \leq \frac{L_\sigma^2}{|\mathcal{B}|}\|v - \bar{v}\|_2^2, \quad \forall i \in [M].$$

**Remark.** Assumption 3 definitely provides a tighter bound for the patterns of variance reduction. In fact, solely with Assumption 2, the term $\mathbb{E}_{\mathcal{B} \sim \mathcal{D}_i}\left\|(\nabla f_i(v, \mathcal{B}) - \nabla f_i(\bar{v}, \mathcal{B})) - (\nabla F_i(v) - \nabla F_i(\bar{v}))\right\|_2^2$ can be bounded by a constant. Therefore, we can easily obtain the coefficient for $\|v - \bar{v}\|_2^2$, which could be with the same structure as the constant in RHS of Assumption 2. Furthermore, if the loss function is Lipschitz smooth, e.g., cross-entropy loss (Tewari & Chaudhuri, 2015), we can derive a similar structure as presented in Assumption 3.

## 4.1 SEQUENTIAL PROBABILITY SETTINGS

As mentioned in Section 3, a recursive pattern appeared in the probability sequence $\{p_t \in \{0, 1\}\}_{t \geq 0}$ can reduce FedAMD to the existing works. We assume that the caching gradient updates every $\tau(\geq 2)$ rounds, such that

$$p_t = \begin{cases} 1, & t \bmod \tau == 0 \\ 0, & \text{Otherwise} \end{cases}$$

We derive the following results under sequential probability settings. The corresponding proof is provided in Appendix D.

**Theorem 1.** *Suppose that Assumption 1, 2 and 3 hold. Let the local updates $K \geq 1$, the minibatch size $b = \min\left(\frac{\sigma^2}{M\epsilon}, n\right)$ and $b' < b$. Additionally, the settings for the local learning rate $\eta_l$ and the global learning rate $\eta_s$ satisfy the following two constraints: (1) $\eta_s \eta_l = \frac{1}{KL}\left(1 + \frac{2M\tau}{A}\right)^{-1}$; and (2) $\eta_l \leq \min\left(\frac{1}{2\sqrt{6}KL}, \frac{\sqrt{b'/K}}{4\sqrt{3}L_\sigma}\right)$. Then, to find an $\epsilon$-approximation of non-convex objectives, i.e., $\min_{t \in [T]}\|\nabla F(\tilde{x}_t)\|_2^2 \leq \epsilon$, the number of communication rounds $T$ performed by FedAMD is*

$$T = O\left(\left(1 + \frac{2M\tau}{A}\right) \cdot \frac{\tau}{\tau - 1} \cdot \frac{1}{\epsilon}\right)$$

*where we treat $\nabla F(\tilde{x}_0) - F_*$ and $L$ as constants.*

**Discussion on the selection of $\tau$.** According to Theorem 1, we notice that $\tau = 2$ achieves $\min_{t \in [T]}\|\nabla F(\tilde{x}_t)\|_2^2 \leq \epsilon$ with the fewest communication rounds. The following corollary discloses the relation between computation overhead and the value of $\tau$.

**Corollary 1.** *Under the setting of Theorem 1, FedAMD computes $O\left(\frac{Mb}{\epsilon} + \frac{\tau MKb'}{\epsilon}\right)$ gradients and consumes a communication overhead of $O\left(\frac{M\tau}{A\epsilon}\right)$ during the model training.*

**Remark.** FedAMD requires an increasing computation and communication cost as $\tau$ gets larger. Therefore, $\tau = 2$ possesses the most outstanding performance in the sequential probability settings. In this case, FedAMD requires the communication rounds of $O\left(\frac{M}{A\epsilon}\right)$, the communication overhead of $O\left(\frac{M}{A\epsilon}\right)$, and the computation cost of $O\left(\frac{\sigma^2}{\epsilon^2} + \frac{MK}{\epsilon}\right)$ while optimizing an online scenario where the size of local dataset is infinity large.

**Comparison with BVR-L-SGD.** As discussed in Section 3, FedAMD reduces to BVR-L-SGD (Murata & Suzuki, 2021) when $\tau = 2$ and all clients participate in the training. In this case, Theorem 1 shows a total of $T = O(1/\epsilon)$ communication rounds is needed. This result coincides with the complexity of BVR-L-SGD in Table 1 by the setting that (1) $nM \leq \frac{1}{\epsilon}$, and (2) $K \geq \sqrt{n/M}$. In other words, we theoretically prove that BVR-L-SGD still achieves $\min_{t \in [T]} \|\nabla F(\tilde{x}_t)\|_2^2 \leq \epsilon$ with $T = O(1/\epsilon)$ in a looser constraint. As for computation overhead, our proposed method requires $O\left(\frac{\sigma^2}{\epsilon^2} + \frac{MK}{\epsilon}\right)$, which is less than BVR-L-SGD (i.e., $O\left(\frac{\sigma^2}{\epsilon^2} + \frac{\sigma^2 + MK}{\epsilon} + MK\right)$).

## 4.2 Constant Probability Settings

Apparently, when we set the constant probability as 1, all participants are in the anchor group such that the model cannot be updated. Likewise, when the constant probability is 0, all participants are in the miner group such that the global target cannot be updated, leading to degraded performance. Therefore, we manually define a constant $p \in (0, 1)$ such that $\{p_t = p\}_{t \geq 0}$. In this section, we derive the following results with partial client participation. Detailed proof is provided in Appendix E. Specifically, Appendix E.2 and Appendix E.3 proves the convergence rate for Theorem 2 and Theorem 3, respectively.

**Theorem 2.** *Suppose that Assumption 1, 2 and 3 hold. Let the local updates $K \geq \max\left(1, \frac{2L_\sigma^2}{b'L^2}\right)$, the minibatch size $b = \min\left(\frac{\sigma^2}{M\epsilon}, n\right)$ and $b' < b$, the local learning rate $\eta_l = \frac{1}{2\sqrt{6}KL}$, and the global learning rate $\eta_s = \frac{2\sqrt{6}}{1 + \frac{2M}{Ap}\sqrt{1-p^A}}$. Then, to find an $\epsilon$-approximation of non-convex objectives, i.e., $\min_{t \in [T]} \|\nabla F(\tilde{x}_t)\|_2^2 \leq \epsilon$, the number of communication rounds $T$ performed by FedAMD is*

$$T = O\left(\frac{1}{\epsilon}\left(\frac{1}{1-p^A} + \frac{M}{Ap\sqrt{1-p^A}}\right)\right)$$

*where we treat $\nabla F(\tilde{x}_0) - F_*$ and $L$ as constants.*

With the constant probability $p$ approaching 0 or 1, Theorem 2 shows that FedAMD requires a significant number of communication rounds. Hence, there is an optimal $p$ such that FedAMD achieves convergence with the fewest communication rounds. In view that $M \geq A$, the number of communication rounds is dominated by $O\left(\frac{M}{Ap\sqrt{1-p^A}} \cdot \frac{1}{\epsilon}\right)$. Based on this observation, the following corollary provides the settings for the constant probability $p$ that leads to the optimal convergence result. Based on the value of $p$, we further refine the settings for other parameters. The following corollary takes $b' = 1$ into consideration, i.e., the small batch size is 1.

**Corollary 2.** *Suppose that Assumption 1, 2 and 3 hold. Let the constant probability $p = \frac{1}{c}\left(\frac{2}{A+2}\right)^{1/A}$, where $c$ is a constant greater than or equal to 1, the local updates $K \geq \max\left(1, \frac{2L_\sigma^2}{L^2}\right)$, the minibatch size $b = \min\left(\frac{\sigma^2}{M\epsilon}, n\right)$ and $b' = 1$, the local learning rate $\eta_l = \frac{1}{2\sqrt{6}KL}$, and the global learning rate $\eta_s = \frac{2\sqrt{6}A}{A+3Mc}$. Then, after the communication rounds of $T = O\left(\frac{M}{A\epsilon}\right)$, we have $\min_{t \in [T]} \|\nabla F(\tilde{x}_t)\|_2^2 \leq \epsilon$. Therefore, the number of total samples called by all clients (i.e., cumulative gradient complexity) is $O\left(\frac{\sigma^2}{\epsilon^2} + \frac{MK}{\epsilon}\right)$ when it optimizes an online scenario.*

**Discussion on the effectiveness of $c$.** When $c = 1$, we can obtain the minimum value for the term $\left(p\sqrt{1-p^A}\right)^{-1}$ with the constant $p$ devised in Corollary 2. As the number of participants (i.e., $A$) gets larger, the optimal $p$ increases as well and tends to be 1, indicating that most participants are in the anchor group. When we optimize an online scenario, the anchors compute a gradient with massive samples. As a result, the computation overhead of a single round is not acceptable. By deducting the anchor sampling probability to its $1/c$, FedAMD consumes up to $(c-1)/c$ less computation overhead, while its convergence performance remains.

**Comparison with FedAvg.** As a classical algorithm, FedAvg (Yang et al., 2020) requires $O\left(\frac{K}{A\epsilon^2} + \frac{1}{\epsilon}\right)$ communication rounds to achieve $\min_{t \in [T]} \|\nabla F(\tilde{x}_t)\|_2^2 \leq \epsilon$ with the total computation

consumption of $O\left(\frac{K^2}{\epsilon^2} + \frac{AK}{\epsilon}\right)$. Apparently, FedAMD needs $O(\frac{1}{\epsilon})$ fewer communication rounds. As mentioned in Section 3, FedAMD consumes $(1-p)/2$ more communication overhead than FedAvg. As $\epsilon$ is close to 0, the total communication overhead for FedAMD is far less than the cost of FedAvg. As for computation overhead, (Yang et al., 2020) implicitly assumes that $K \geq \sigma^2$, FedAMD is more communication friendly than FedAvg.

In addition to the generalized non-convex objectives, we investigate the convergence rate of the PL condition, a special case under non-convex objectives. The following assumption describes this case:

**Assumption 4** (PL Condition (Karimi et al., 2016))**.** *The objective function $F$ satisfies the PL condition when there exists a scalar $\mu > 0$ such that*

$$\|\nabla F(v)\|_2^2 \geq 2\mu \left(F(v) - F_*\right), \quad \forall v \in \mathbb{R}^d.$$

Under PL condition, the rest of the section draws the convergence performance of FedAMD with partial client participation.

**Theorem 3.** *Suppose that Assumption 1, 2, 3 and 4 hold. Let the local updates $K \geq \max\left(1, \frac{2L_\sigma^2}{b'L^2}\right)$, the minibatch size $b = \min\left(\frac{\sigma^2}{M\mu\epsilon}, n\right)$ and $b' < b$, the local learning rate $\eta_l = \frac{1}{2\sqrt{6}KL}$, and the global learning rate $\eta_s = \min\left(\frac{2\sqrt{6}LAp}{M\mu(1-p^A)}, \frac{2\sqrt{6}}{1+\frac{16ML}{\mu Ap}}\right)$. Then, to find an $\epsilon$-approximation of PL condition, i.e., $F(\tilde{x}_T) - F(x_*) \leq \epsilon$, the number of communication rounds $T$ performed by FedAMD is*

$$T = O\left(\frac{1}{\mu(1-p^A)}\left(1 + \frac{M}{\mu Ap} + \frac{M\mu(1-p^A)}{Ap}\right)\log\frac{1}{\epsilon}\right)$$

*where we treat $\nabla F(\tilde{x}_0) - F_*$ and $L$ as constants.*

Similar to Theorem 2, the number of communication rounds of FedAMD is mainly occupied by $O\left(\frac{M}{\mu^2 Ap(1-p^A)} + \frac{M}{Ap}\right)$. According to such an approximation, we provide a mathematical expression for the setting of $p$ in Corollary 3. Subsequently, we adjust the value of the hyper-parameters such that we can obtain the best result for Theorem 3.

**Corollary 3.** *Suppose that Assumption 1, 2, 3 and 4 hold. Let the constant probability $p = \frac{1}{c}\left(\left(1 + \frac{A+1}{2\mu^2}\right) - \sqrt{\left(\frac{A+1}{2\mu^2}\right)^2 + \frac{A}{\mu^2}}\right)^{1/A}$, where $c$ is a constant greater than or equal to 1, the local updates $K \geq \max\left(1, \frac{2L_\sigma^2}{L^2}\right)$, the minibatch size $b = \min\left(\frac{\sigma^2}{M\mu\epsilon}, n\right)$ and $b' = 1$, the local learning rate $\eta_l = \frac{1}{2\sqrt{6}KL}$, and the global learning rate $\eta_s = \min\left(\frac{\sqrt{6}AL}{M\mu c}, 2\sqrt{6}\left(1 + \frac{32Mc}{\mu A}\right)^{-1}\right)$. Then, after the communication rounds of $T = O\left(\left(\frac{1}{\mu} + \frac{M}{\mu^2 A} + \frac{M}{A}\right)\log\frac{1}{\epsilon}\right)$, we have $F(\tilde{x}_T) - F(x_*) \leq \epsilon$. Therefore, the number of total samples called by all clients (i.e., cumulative gradient complexity) is $O\left(\left(\frac{A}{\mu} + \frac{M}{\mu^2} + M\right)\left(\frac{\sigma^2}{M\mu\epsilon} + K\right)\log\frac{1}{\epsilon}\right)$ when it optimizes an online scenario.*

**Remark.** When $c = 1$, the probability $p$ for anchor sampling approaches 100% as the number of participants is increasing. Likewise, it is necessary to use $c \geq 1$ to reduce the computation consumption of each round. Besides, FedAMD achieves a linear convergence under PL conditions. In view that strongly-convex objectives possess a looser setting than PL conditions, FedAMD can also achieve linear convergence under strongly-convex objectives.

## 5 EXPERIMENTS

This section presents the experiments of our proposed approach and other existing baselines that are most relative to this work. We also investigate the effectiveness of probability $\{p_t\}_{t\geq0}$. Account for the limited space, numerical analysis on other factors like the number of local updates are presented in the supplementary materials.

| | | | | |
|---|---|---|---|---|
| (a) 20 clients | | (b) 40 clients | | (c) 100 clients |

Figure 1: Comparison of different probability settings using test accuracy against the communication rounds for FedAMD.

| Method | 20 clients | | | | 40 clients | | | | 100 clients | | | |
|---|---|---|---|---|---|---|---|---|---|---|---|---|
| | Grad. | Comm. | Round | Acc. | Grad. | Comm. | Round | Acc. | Grad. | Comm. | Round | Acc. |
| BVR-L-SGD | 101.9 | 853.9 | 310 | 77.0 | 177.4 | 1492.3 | 275 | 78.6 | 463.8 | 3908.7 | 291 | 78.0 |
| FedAvg | 40.8 | 566.9 | 318 | 76.4 | 78.3 | 1087.5 | 305 | 76.7 | 186.9 | 2594.5 | 291 | 77.0 |
| FedPAGE | 148.6 | 1670.4 | 271 | 79.2 | 164.3 | 1622.4 | 203 | 80.6 | 525.5 | 4540.3 | 339 | 77.1 |
| SCAFFOLD | 47.5 | 1318.6 | 370 | 75.9 | 91.4 | 2537.8 | 356 | 76.0 | 250.9 | 6966.0 | 391 | 75.1 |
| FedAMD (constant) | 35.5 | 489.8 | 259 | 80.6 | 55.0 | 776.3 | 209 | 82.3 | 153.9 | 2147.8 | 229 | 83.4 |
| FedAMD (sequential) | 40.2 | 475.3 | 213 | 79.5 | 80.4 | 904.5 | 190 | 80.8 | 253.8 | 2998.8 | 269 | 78.7 |

Table 2: Comparison among baselines in terms of cumulative gradient complexity ($\times 10^5$ samples), communication costs ($\times 32$ Mbits), and rounds reaching the accuracy of $75\%$, and the final accuracy (%) after 400 rounds. **Bold**: The best result in each column; underline: The best result of the baselines in each column; *Italy*: The results that FedAMD outperforms all baselines in each column.

**Experimental setup.** We train a convolutional neural network LeNet-5 (LeCun et al., 2015; 1989) (non-convex) using Fashion MNIST (Xiao et al., 2017) which consists of 10 classes that equally disjoint the training set of 60K samples and the test set of 10K samples. We conduct the experiments with a total of 100 clients. To simulate the non-i.i.d. features, each client holds the data from 2 classes with a total of 600 samples, and each label is held by 20 clients. Let the number of local updates $K$ be 10, the mini-batch size $b'$ be 64 and $b$ be 600. For different experiments, unless some hyper-parameters have been defined, we leverage the best setting (e.g., learning rate) to obtain the best results. All the numerical results in this section represent the average performance of three experiments using different random seeds. More empirical results are put in Appendix F.

**Effectiveness of probability** $\{p_t\}_{t \geq 0}$**.** Figure 1 demonstrates the performance of various probability settings under the scenarios of different participants. In Figure 1a with 20 clients, both sequential probability setting and constant probability setting achieve the best performance. In Figure 1b and 1c, where 40 and 100 clients are selected in each round, constant probability setting outperforms sequential probability setting. In all three scenarios, with sequential probability settings, the pattern of $\{0, 0, 1\}$ has a much worse performance than the pattern of $\{0, 1\}$. This empirically validates Theorem 1 for the best setting $\tau = 2$ in terms of communication complexity. Similarly, with constant probability settings, the best performance is achieved when $p$ approximates or equals optimal, which validates the statement in Corollary 2.

**Comparison with the state-of-the-art works.** Table 2 compares FedAMD with the existing works under partial/full client participation. At first glance, FedAMD outperforms other baselines because the texts in bold are all appeared in FedAMD. With 20-client participation, our proposed method surpasses four baselines all aroundness. As for 40-client participation, FedAMD with constant probability saves at least 30% computation and communication cost, and the final accuracy realizes up to 6% improvement. In terms of the training with full client participation, FedAMD requires 10%–20% fewer communication rounds, and its final accuracy has significant improvement, i.e., within the range of 0.7%–8.3%. Also, it is well noted that BVR-L-SGD has a similar performance as FedAMD using sequential probability in terms of communication rounds and test accuracy, but the former needs more computation and communication overhead. This is because BVR-L-SGD computes the bullseye using multiple $b'$-size batches rather than a large batch.

## 6 CONCLUSION

In this work, we investigate a federated learning framework FedAMD that disjoints the partial participants into anchor and miner groups. We provide the convergence analysis of our proposed algorithm for constant and sequential probability settings. Under the partial-client scenario, FedAMD achieves sublinear speedup under non-convex objectives and linear speedup under the PL condition. To the best of our knowledge, this is the first work to analyze the effectiveness of large batches under partial client participation. Experimental results demonstrate that FedAMD is superior to state-of-the-art works. It is interesting to explore anchor sampling in the other scenarios of FL, e.g., arbitrary device unavailability.

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

# A    RELATED WORK

**Mini-batch SGD vs. Local SGD.**    Distributed optimization is required to train large-scale deep learning systems. Local SGD (also known as FedAvg) (Stich, 2018; Dieuleveut & Patel, 2019; Haddadpour et al., 2019; Haddadpour & Mahdavi, 2019) performs multiple (i.e., $K \geq 1$) local updates with $K$ small batches, while mini-batch SGD computes the gradients averaged by $K$ small batches (Woodworth et al., 2020b;a) (or a large batches (Shallue et al., 2019; You et al., 2018; Goyal et al., 2017)) on a given model. There has been a long discussion on which one is better (Lin et al., 2019; Woodworth et al., 2020a;b; Yun et al., 2021), but no existing work considers how to disjoint the nodes such that both can be trained at the same time.

**Federated Learning.**    FL was proposed to ensure data privacy and security (Kairouz et al., 2019), and now it has become a hot field in the distributed system (Yuan & Ma, 2020; Shamsian et al., 2021; Zhang et al., 2021; Avdiukhin & Kasiviswanathan, 2021; Yuan et al., 2021; Diao et al., 2020; Blum et al., 2021). The FL training methods in the past few years usually require all trainers to participate in each training session (Kairouz et al., 2019), but this is obviously impractical when facing the increase in FL clients. To enhance the systems' feasibility, this work assumes that a fixed number of clients are sampled at each round, which is widely adopted in (Li et al., 2019b; Philippenko & Dieuleveut, 2020; Gorbunov et al., 2021a; Karimireddy et al., 2020b; Yang et al., 2020; Li et al., 2020; Eichner et al., 2019; Ruan et al., 2021). Therefore, the server collects the data from this participation every synchronization to update the model parameters (Li et al., 2019b; Philippenko & Dieuleveut, 2020; Gorbunov et al., 2021a; Karimireddy et al., 2020b; Yang et al., 2020; Li et al., 2020; Eichner et al., 2019; Yan et al., 2020; Ruan et al., 2021; Lai et al., 2021; Gu et al., 2021).

**Variance Reduction in Finite-sum Problems.**    Variance reduction techniques (Johnson & Zhang, 2013; Defazio et al., 2014; Nguyen et al., 2017; Li et al., 2021a; Lan & Zhou, 2018a;b; Allen-Zhu & Hazan, 2016; Reddi et al., 2016; Lei et al., 2017; Zhou et al., 2018; Horváth & Richtárik, 2019; Horváth et al., 2020; Fang et al., 2018; Wang et al., 2018; Li, 2019; Roux et al., 2012; Lian et al., 2017; Zhang et al., 2016) was once proposed for traditional centralized machine learning to optimize finite-sum problems (Bietti & Mairal, 2017; Bottou & Cun, 2003; Robbins & Monro, 1951) by mitigating the estimation gap between small-batch (Bottou, 2012; Ghadimi et al., 2016; Khaled & Richtárik, 2020) and large-batch (Nesterov, 2003; Ruder, 2016; Mason et al., 1999). SGD randomly samples a small-batch and computes the gradient in order to approach the optimal solution. Since the data are generally noisy, an insufficiently large batch results in convergence rate degradation. By utilizing all data in every update, GD can remove the noise affecting the training process. However, it is time-consuming because the period for a single GD step can implement multiple SGD updates. Based on the trade-off, variance-reduced methods periodically perform GD steps while correcting SGD updates with reference to the most recent GD steps.

**Variance Reduction in FL.**    The variance-reduced techniques have critically driven the advent of FL algorithms (Karimireddy et al., 2020b; Wu et al., 2021; Liang et al., 2019; Karimireddy et al., 2020a; Murata & Suzuki, 2021; Mitra et al., 2021) by correcting each local computed gradient with respect to the estimated global orientation. However, a concern is addressed on how to attain an accurate global orientation to mitigate the update drift from the global model. Roughly, the estimation lies in two types, namely, precalculated and cached. The former methods (Murata & Suzuki, 2021; Mitra et al., 2021) required precalculation typically require full worker participation, which is infeasible for federated learning settings. As for the global orientation estimated by cached information, existing approaches (Karimireddy et al., 2020b; Wu et al., 2021; Liang et al., 2019; Karimireddy et al., 2020a) utilize small batches, which derives a biased estimation and misleads the training. This work explores the effectiveness of large-batch estimation for the global orientation under partial client participation.

# B    USEFUL LEMMAS

Prior to giving detailed proofs of the theorems, we cover some technical lemmas in this section, and all of them are valid in general cases.

**Lemma 1.** *Let $\varepsilon = \{\varepsilon_1, \ldots, \varepsilon_a\}$ be the set of random variables in $\mathbb{R}^{a \times d}$. Every element in $\varepsilon$ is independent with others. For $i \in \{1, \ldots, a\}$, the value for $\varepsilon_i$ follows the setting below:*

$$\varepsilon_i = \begin{cases} e_i, & probability = q \\ \mathbf{0}, & otherwise \end{cases} \tag{2}$$

*where $q$ is a constant real number between 0 and 1, i.e., $q \in [0, 1]$. Let $|\cdot|$ indicate the length of a set, $\varepsilon \setminus \{\mathbf{0}\}$ represent a set in which an element is in $\varepsilon$ but not $\mathbf{0}$. Then, there is a probability of $(1 - q)^a$ for $|\varepsilon \setminus \{\mathbf{0}\}| = 0$, let $avg(\varepsilon)$ be the averaged result with the exception of zero vectors, i.e.,*

$$avg(\varepsilon) = \begin{cases} \frac{1}{|\varepsilon \setminus \{\mathbf{0}\}|} \sum_{i=1}^{a} \varepsilon_i, & |\varepsilon \setminus \{\mathbf{0}\}| \neq 0 \\ 0, & |\varepsilon \setminus \{\mathbf{0}\}| = 0 \end{cases} \tag{3}$$

*Then, the following formulas hold for $\mathbb{E}\left(avg(\varepsilon)\right)$ and its second norm $\mathbb{E}\left\|avg(\varepsilon)\right\|_2^2$:*

$$\mathbb{E}\left(avg(\varepsilon)\right) = (1 - (1-q)^a) \cdot \frac{1}{a} \sum_{i=1}^{a} e_i; \quad \mathbb{E}\left\|avg(\varepsilon)\right\|_2^2 \leq (1 - (1-q)^a) \cdot \frac{1}{a} \sum_{i=1}^{a} \|e_i\|_2^2 \tag{4}$$

*Proof.* When $q = 0$, the formulas in Equation 4 obviously hold because $\mathbb{E}\left(avg(\varepsilon)\right) = 0$ and $\mathbb{E}\left\|avg(\varepsilon)\right\|_2^2 = 0$. As for $q = 1$, since $avg(\varepsilon) = \frac{1}{a} \sum_{i=1}^{a} e_i$, we leverage Cauchy–Schwarz inequality and get $\mathbb{E}\left\|avg(\varepsilon)\right\|_2^2 = \left\|\frac{1}{a} \sum_{i=1}^{a} e_i\right\|_2^2 \leq \frac{1}{a} \sum_{i=1}^{a} \|e_i\|_2^2$, which is consistent with the formulas in Equation 4. In addition to the preceding cases, we consider some general cases for the probability $q$ within 0 and 1, i.e., $q \in (0, 1)$.

Firstly, we show the proof details for $\mathbb{E}\left(avg(\varepsilon)\right)$. For all $i$ in $\{1, \ldots, a\}$, given that $\varepsilon_i$ is not a zero vector, the coefficient of $e_i$ is based on the binomial distribution on how many non-zero elements in the set $\{\varepsilon_1, \ldots, \varepsilon_{i-1}\} \cup \{\varepsilon_{i+1}, \ldots, \varepsilon_a\}$. Therefore, with the probability $q$ that $\varepsilon_i$ is equal to $e_i$, the coefficient of $e_i$ in the expected form is

$$q \left( \frac{1}{a} \cdot \underbrace{\binom{a-1}{a-1} q^{a-1}}_{(a-1) \text{ non-zero elements}} + \cdots + \frac{1}{1} \cdot \underbrace{\binom{a-1}{0}(1-q)^{a-1}}_{0 \text{ non-zero element}} \right)$$

Then, the coefficient of $\frac{1}{a} e_i$ can be expressed and simplified for

$$q \left( \frac{a}{a} \cdot \binom{a-1}{a-1} q^{a-1} + \cdots + \frac{a}{1} \cdot \binom{a-1}{0}(1-q)^{a-1} \right) \tag{5}$$

$$= q \left( \binom{a}{a} q^{a-1} + \cdots + \binom{a}{1}(1-q)^{a-1} \right) \tag{6}$$

$$= \binom{a}{a} q^a + \cdots + \binom{a}{1} q(1-q)^{a-1} \tag{7}$$

$$= 1 - (1-q)^a \tag{8}$$

where Equation (7) follows

$$\binom{\alpha}{\beta} = \frac{\alpha}{\beta} \cdot \frac{(\alpha-1) \times \cdots \times (\alpha-\beta+1)}{1 \times \cdots \times (\beta-1)} = \frac{\alpha}{\beta} \binom{\alpha-1}{\beta-1}, \quad \forall \alpha \geq \beta > 0$$

and Equation (8) follows

$$(q + (1-q))^a = \binom{a}{a} q^a + \cdots + \binom{a}{0}(1-q)^a.$$

Thus, the equation $\mathbb{E}\left(avg(\varepsilon)\right) = (1 - (1-q)^a) \cdot \frac{1}{a} \sum_{i=1}^{a} e_i$ holds.

Secondly, we provide the analysis for $\mathbb{E} \left\| avg(\varepsilon) \right\|_2^2$. Based on the definition for $avg(\varepsilon)$ in Equation (3), we discuss the case $|\varepsilon \setminus \{\mathbf{0}\}| \neq 0$. By means of Cauchy-Schwarz inequality, we can obtain the following inequality:

$$\left\| \frac{1}{|\varepsilon \setminus \{\mathbf{0}\}|} \sum_{i=1}^a \varepsilon_i \right\|_2^2 = \left\| \frac{1}{|\varepsilon \setminus \{\mathbf{0}\}|} \sum_{i, \varepsilon_i \neq \mathbf{0}} \varepsilon_i \right\|_2^2 \leq \frac{1}{|\varepsilon \setminus \{\mathbf{0}\}|} \sum_{i, \varepsilon_i \neq \mathbf{0}} \|\varepsilon_i\|_2^2 = \frac{1}{|\varepsilon \setminus \{\mathbf{0}\}|} \sum_{i=1}^a \|\varepsilon_i\|_2^2 \quad (9)$$

Therefore,

$$\|avg(\varepsilon)\|_2^2 \leq \begin{cases} \frac{1}{|\varepsilon \setminus \{\mathbf{0}\}|} \sum_{i=1}^a \|\varepsilon_i\|_2^2, & |\varepsilon \setminus \{\mathbf{0}\}| \neq 0 \\ 0, & |\varepsilon \setminus \{\mathbf{0}\}| = 0 \end{cases} \quad (10)$$

Apparently, Equation (10) is very similar to Equation (3) in terms of the expression. As a result, we can adopt the same proof framework in the analysis of $\mathbb{E}\left(avg(\varepsilon)\right)$. Then, we can directly draw a conclusion $\mathbb{E} \left\| avg(\varepsilon) \right\|_2^2 \leq (1 - (1-q)^a) \cdot \frac{1}{a} \sum_{i=1}^a \|e_i\|_2^2$. $\qquad \square$

**Lemma 2.** *Let $\varepsilon = \{\varepsilon_1, \ldots, \varepsilon_a\}$ be the set of random variables in $\mathbb{R}^d$ with the number of $a$. These random variables are not necessarily independent. We can suppose that $\mathbb{E}\left[\varepsilon_i\right] = e_i$, and the variance is bounded as $\mathbb{E}\left[\left\|\varepsilon_i - e_i\right\|_2^2\right] \leq \sigma^2$. After that we can get*

$$\mathbb{E}\left[\left\|\sum_{i=1}^a \varepsilon_i\right\|_2^2\right] \leq \left\|\sum_{i=1}^a e_i\right\|_2^2 + a^2 \sigma^2 \quad (11)$$

*If we make another suppose that the conditional mean of these random variables is $\mathbb{E}\left[\varepsilon_i | \varepsilon_{i-1}, \ldots, \varepsilon_1\right] = e_i$, and the variables $\{\varepsilon_i - e_i\}$ form a martingale difference sequence, and the bound of the variance is $\mathbb{E}\left[\left\|\varepsilon_i - e_i\right\|_2^2\right] \leq \sigma^2$. So we can make a much tighter bound*

$$\mathbb{E}\left[\left\|\sum_{i=1}^a \varepsilon_i\right\|_2^2\right] \leq 2 \left\|\sum_{i=1}^a e_i\right\|_2^2 + 2a\sigma^2 \quad (12)$$

*Proof.* For any random variable $X$, $\mathbb{E}\left[X^2\right] = \left(\mathbb{E}\left[X - \mathbb{E}\left[X\right]\right]\right)^2 + \left(\mathbb{E}\left[X\right]\right)^2$ implying

$$\mathbb{E}\left[\left\|\sum_{i=1}^a \varepsilon_i\right\|_2^2\right] = \left\|\sum_{i=1}^a e_i\right\|_2^2 + \mathbb{E}\left[\left\|\sum_{i=1}^a \varepsilon_i - e_i\right\|_2^2\right] \quad (13)$$

Expanding above expression using relaxed triangle inequality:

$$\mathbb{E}\left[\left\|\sum_{i=1}^a \varepsilon_i - e_i\right\|_2^2\right] \leq a \sum_{i=1}^a \mathbb{E}\left[\|\varepsilon_i - e_i\|_2^2\right] \leq a^2 \sigma^2 \quad (14)$$

For the second statement, $e_i$ depends on $[\varepsilon_{i-1}, \ldots, \varepsilon_1]$. Thus we choose to use a relaxed triangle inequality

$$\mathbb{E}\left[\left\|\sum_{i=1}^a \varepsilon_i\right\|_2^2\right] \leq 2 \left\|\sum_{i=1}^a e_i\right\|_2^2 + 2\mathbb{E}\left[\left\|\sum_{i=1}^a \varepsilon_i - e_i\right\|_2^2\right] \quad (15)$$

then we use a much tighter expansion and we can get:

$$\mathbb{E}\left[\left\|\sum_{i=1}^a \varepsilon_i - e_i\right\|_2^2\right] = \sum_{i,j} \mathbb{E}\left[(\varepsilon_i - e_i)^\top (\varepsilon_j - e_j)\right] = \sum_i \mathbb{E}\left[\left\|\sum_{i=1}^a \varepsilon_i - e_i\right\|_2^2\right] \leq a\sigma^2 \quad (16)$$

When $\{\varepsilon_i - e_i\}$ form a martingale difference sequence, the cross terms will have zero means. $\qquad \square$

**Lemma 3.** *Suppose there is a sequence $\{y_t \in \mathbb{R}^d\}_{t \geq 0}$ satisfying a recursive function $y_{t+1} = y_t - \eta \Delta y_t$, where $\eta > 0$ is a constant and $\Delta y_t \in \mathbb{R}^d$ is a vector. Given a L-smooth function $G$, the following inequality holds for any $\eta$ and $\Delta y_t$:*

$$G(y_{t+1}) \leq G(y_t) - \frac{\eta \eta'}{2} \|\nabla G(y_t)\|_2^2 - \left(\frac{1}{2\eta\eta'} - \frac{L}{2}\right) \|y_{t+1} - y_t\|_2^2 + \frac{\eta}{2\eta'} \|\Delta y_t - \eta' \nabla G(y_t)\|_2^2 \tag{17}$$

*where $\eta' > 0$ can be any constant.*

*Proof.* Since $G$ is a L-smooth function, for any $v, \bar{v} \in \mathbb{R}^d$, the following inequality holds:

$$G(\bar{v}) = G(v) + \int_0^1 \frac{\partial G(v + t(\bar{v} - v))}{\partial t} dt \tag{18}$$

$$= G(v) + \int_0^1 \nabla G(v + t(\bar{v} - v)) \cdot (\bar{v} - v) dt \tag{19}$$

$$= G(v) + \nabla G(v)(\bar{v} - v) + \int_0^1 (\nabla G(v + t(\bar{v} - v)) - G(v)) \cdot (\bar{v} - v) dt \tag{20}$$

$$\leq G(v) + \nabla G(v)(\bar{v} - v) + \int_0^1 L\|t(\bar{v} - v)\|_2 \|\bar{v} - v\|_2 dt \tag{21}$$

$$\leq G(v) + \nabla G(v)(\bar{v} - v) + \frac{L}{2}\|\bar{v} - v\|_2^2. \tag{22}$$

Based on the conclusion on L-smooth drawn from Equation (22), we derive Equation (17) step by step:

$$G(y_{t+1}) \leq G(y_t) + \langle \nabla G(y_t), y_{t+1} - y_t \rangle + \frac{L}{2} \|y_{t+1} - y_t\|_2^2 \tag{23}$$

$$= G(y_t) + \langle \nabla G(y_t), -\eta \Delta y_t \rangle + \frac{L}{2} \|y_{t+1} - y_t\|_2^2 \tag{24}$$

$$= G(y_t) - \frac{\eta}{\eta'} \langle \eta' \nabla G(y_t), \Delta y_t \rangle + \frac{L}{2} \|y_{t+1} - y_t\|_2^2 \tag{25}$$

$$= G(y_t) - \frac{\eta}{2\eta'} \left(\eta'^2 \|\nabla G(y_t)\|_2^2 + \|\Delta y_t\|_2^2 - \|\Delta y_t - \eta' \nabla G(y_t)\|_2^2\right) + \frac{L}{2} \|y_{t+1} - y_t\|_2^2 \tag{26}$$

$$= G(y_t) - \frac{\eta \eta'}{2} \|\nabla G(y_t)\|_2^2 - \left(\frac{1}{2\eta\eta'} - \frac{L}{2}\right) \|y_{t+1} - y_t\|_2^2 + \frac{\eta}{2\eta'} \|\Delta y_t - \eta' \nabla G(y_t)\|_2^2 \tag{27}$$

where Equation (26) is in accordance with $\langle \alpha, \beta \rangle = \frac{1}{2}\left(\alpha^2 + \beta^2 - (\alpha - \beta)^2\right)$, and Equation (27) follows $\|\Delta y_t\|_2^2 = \frac{1}{\eta^2} \|y_{t+1} - y_t\|_2^2$. $\qquad\square$

## C    PRELIMINARY FOR FEDAMD

Algorithm 1 describes FedAMD in details. The objective in this part is to find the recursive function for the sequence of models, i.e., $\{\tilde{\boldsymbol{x}}_t\}_{t\geq 0}$. As mentioned in Line 26 in Algorithm 1, let $\Delta\boldsymbol{x}_t$ aggregate $\Delta\boldsymbol{x}_t^{(i)}$ where client $i$ updates model, then the difference between $\tilde{\boldsymbol{x}}_{t+1}$ and $\tilde{\boldsymbol{x}}_t$ follows the recursive function written as

$$\tilde{\boldsymbol{x}}_{t+1} = \tilde{\boldsymbol{x}}_t - \eta_s \cdot avg(\Delta\boldsymbol{x}_t) \tag{28}$$

where $avg()$ is same as defined in Lemma 1. As we know, the length of $\Delta\boldsymbol{x}_t$ changes over rounds but does not exceed the number of participants, i.e., $|\Delta\boldsymbol{x}_t| \leq A$. Then, suppose that $\Delta\boldsymbol{x}_t^{(m)}$ is in $\Delta\boldsymbol{x}_t$, $\Delta\boldsymbol{x}_t^{(m)}$ can be expressed as

$$\Delta\boldsymbol{x}_t^{(m)} = -\left(\boldsymbol{x}_{t,K}^{(m)} - \tilde{\boldsymbol{x}}_t\right) = -\sum_{k=0}^{K-1}\left(\boldsymbol{x}_{t,k+1}^{(m)} - \boldsymbol{x}_{t,k+1}^{(m)}\right) = \sum_{k=0}^{K-1}\eta_l g_{t,k+1}^{(m)} \tag{29}$$

where the last equal sign is according to Line 20 in Algorithm 1. Next, with the recursive formula in Line 19, we have

$$g_{t,k+1}^{(m)} = g_{t,k}^{(m)} - \nabla f_m\left(\boldsymbol{x}_{t,k-1}^{(m)}, \mathcal{B}_{m,k}'\right) + \nabla f_m\left(\boldsymbol{x}_{t,k}^{(m)}, \mathcal{B}_{m,k}'\right) \tag{30}$$

$$= \tilde{g}_t - \sum_{\kappa=0}^{k}\nabla f_m\left(\boldsymbol{x}_{t,\kappa-1}^{(m)}, \mathcal{B}_{m,\kappa}'\right) + \sum_{\kappa=0}^{k}\nabla f_m\left(\boldsymbol{x}_{t,\kappa}^{(m)}, \mathcal{B}_{m,\kappa}'\right). \tag{31}$$

Then, Equation (29) can be rewritten as

$$\Delta\boldsymbol{x}_t^{(m)} = \eta_l K \tilde{g}_t - \eta_l \sum_{k=0}^{K-1}\sum_{\kappa=0}^{k}\nabla f_m\left(\boldsymbol{x}_{t,\kappa-1}^{(m)}, \mathcal{B}_{m,\kappa}'\right) + \eta_l \sum_{k=0}^{K-1}\sum_{\kappa=0}^{k}\nabla f_m\left(\boldsymbol{x}_{t,\kappa}^{(m)}, \mathcal{B}_{m,\kappa}'\right) \tag{32}$$

## D    PROOFS UNDER SEQUENTIAL PROBABILISTIC SETTINGS

### D.1    PRELIMINARY

**Lemma 4.** *Suppose that Assumption 1, 2 and 3 hold. Let the local learning rate satisfy $\eta_l \leq \min\left(\frac{1}{2\sqrt{3}KL}, \frac{1}{2\sqrt{3}L_\sigma^2}\sqrt{\frac{b'}{K}}\right)$. With FedAMD, $\sum_{k=0}^{K-1}\left\|\boldsymbol{x}_{t,k}^{(m)} - \boldsymbol{x}_{t,k-1}^{(m)}\right\|_2^2$ represents the sum of the second norm of every iteration's difference. Therefore, the bound for such a summation in the expected form should be*

$$\sum_{k=0}^{K-1}\mathbb{E}\left\|\boldsymbol{x}_{t,k}^{(m)} - \boldsymbol{x}_{t,k-1}^{(m)}\right\|_2^2 \leq 6\eta_l^2 K \|\tilde{g}_t - \nabla F(\tilde{\boldsymbol{x}}_t)\|_2^2 + 6\eta_l^2 K \|\nabla F(\tilde{\boldsymbol{x}}_t)\|_2^2 \tag{33}$$

*Proof.* According to Equation (31), the update at $(k-1)$-th iteration is

$$\boldsymbol{x}_{t,k}^{(m)} - \boldsymbol{x}_{t,k-1}^{(m)} = -\eta_l g_{t,k}^{(m)} = -\eta_l\left(\tilde{g}_t - \sum_{\kappa=0}^{k-1}\nabla f_m\left(\boldsymbol{x}_{t,\kappa-1}^{(m)}, \mathcal{B}_{m,\kappa}'\right) + \sum_{\kappa=0}^{k-1}\nabla f_m\left(\boldsymbol{x}_{t,\kappa}^{(m)}, \mathcal{B}_{m,\kappa}'\right)\right). \tag{34}$$

To find the bound for the expected value of its second norm, the analysis is presented as follows:

$$\mathbb{E}\left\|\boldsymbol{x}_{t,k}^{(m)} - \boldsymbol{x}_{t,k-1}^{(m)}\right\|_2^2 \tag{35}$$

$$= \eta_l^2 \mathbb{E}\left\|\tilde{g}_t - \sum_{\kappa=0}^{k-1}\nabla f_m\left(\boldsymbol{x}_{t,\kappa-1}^{(m)}, \mathcal{B}_{m,\kappa}'\right) + \sum_{\kappa=0}^{k-1}\nabla f_m\left(\boldsymbol{x}_{t,\kappa}^{(m)}, \mathcal{B}_{m,\kappa}'\right)\right\|_2^2 \tag{36}$$

$$\leq 3\eta_l^2 \|\tilde{g}_t - \nabla F(\tilde{\boldsymbol{x}}_t)\|_2^2 + 3\eta_l^2 \|\nabla F(\tilde{\boldsymbol{x}}_t)\|_2^2$$

$$+ 3\eta_l^2 \mathbb{E}\left\|\sum_{\kappa=0}^{k-1}\nabla f_m\left(\boldsymbol{x}_{t,\kappa-1}^{(m)}, \mathcal{B}_{m,\kappa}'\right) - \sum_{\kappa=0}^{k-1}\nabla f_m\left(\boldsymbol{x}_{t,\kappa}^{(m)}, \mathcal{B}_{m,\kappa}'\right)\right\|_2^2 \tag{37}$$

$$= 3\eta_l^2 \left\| \tilde{g}_t - \nabla F\left(\tilde{\boldsymbol{x}}_t\right) \right\|_2^2 + 3\eta_l^2 \left\| \nabla F\left(\tilde{\boldsymbol{x}}_t\right) \right\|_2^2 + 3\eta_l^2 \mathbb{E} \left\| \sum_{\kappa=0}^{k-1} \left( \nabla F_m\left(\boldsymbol{x}_{t,\kappa-1}^{(m)}\right) - \nabla F_m\left(\boldsymbol{x}_{t,\kappa}^{(m)}\right) \right) \right\|_2^2$$

$$+ 3\eta_l^2 \mathbb{E} \left\| \sum_{\kappa=0}^{k-1} \left( \nabla f_m\left(\boldsymbol{x}_{t,\kappa-1}^{(m)}, \mathcal{B}_{m,\kappa}'\right) - \nabla f_m\left(\boldsymbol{x}_{t,\kappa}^{(m)}, \mathcal{B}_{m,\kappa}'\right) - \nabla F_m\left(\boldsymbol{x}_{t,\kappa-1}^{(m)}\right) + \nabla F_m\left(\boldsymbol{x}_{t,\kappa}^{(m)}\right) \right) \right\|_2^2 \tag{38}$$

$$\leq 3\eta_l^2 \left\| \tilde{g}_t - \nabla F\left(\tilde{\boldsymbol{x}}_t\right) \right\|_2^2 + 3\eta_l^2 \left\| \nabla F\left(\tilde{\boldsymbol{x}}_t\right) \right\|_2^2 + 3\eta_l^2 K L^2 \sum_{\kappa=0}^{k-1} \mathbb{E} \left\| \boldsymbol{x}_{t,\kappa-1}^{(m)} - \boldsymbol{x}_{t,\kappa}^{(m)} \right\|_2^2$$

$$+ 3\eta_l^2 \mathbb{E} \left\| \sum_{\kappa=0}^{k-1} \left( \nabla f_m\left(\boldsymbol{x}_{t,\kappa-1}^{(m)}, \mathcal{B}_{m,\kappa}'\right) - \nabla f_m\left(\boldsymbol{x}_{t,\kappa}^{(m)}, \mathcal{B}_{m,\kappa}'\right) - \nabla F_m\left(\boldsymbol{x}_{t,\kappa-1}^{(m)}\right) + \nabla F_m\left(\boldsymbol{x}_{t,\kappa}^{(m)}\right) \right) \right\|_2^2 \tag{39}$$

$$\leq 3\eta_l^2 \left\| \tilde{g}_t - \nabla F\left(\tilde{\boldsymbol{x}}_t\right) \right\|_2^2 + 3\eta_l^2 \left\| \nabla F\left(\tilde{\boldsymbol{x}}_t\right) \right\|_2^2 + 3\eta_l^2 K L^2 \sum_{\kappa=0}^{k-1} \mathbb{E} \left\| \boldsymbol{x}_{t,\kappa-1}^{(m)} - \boldsymbol{x}_{t,\kappa}^{(m)} \right\|_2^2$$

$$+ 3\eta_l^2 \frac{L_\sigma^2}{b'} \sum_{\kappa=0}^{k-1} \mathbb{E} \left\| \boldsymbol{x}_{t,\kappa-1}^{(m)} - \boldsymbol{x}_{t,\kappa}^{(m)} \right\|_2^2 \tag{40}$$

where Equation (37) is based on Cauchy-Schwarz inequality; Equation (38) is based on the variance expansion on the third term of Equation (37); Equation (39) is based on Cauchy-Schwarz inequality and Assumption 1 on the third term of Equation (38); Equation (40) is based on Lemma 2 and Assumption 3 on the fourth term of Equation (39).

Therefore, by summing Equation (40) for $k = 1, \ldots, K$, we have

$$\sum_{k=0}^{K-1} \left\| \boldsymbol{x}_{t,k}^{(m)} - \boldsymbol{x}_{t,k-1}^{(m)} \right\|_2^2 \leq \sum_{k=0}^{K} \left\| \boldsymbol{x}_{t,k}^{(m)} - \boldsymbol{x}_{t,k-1}^{(m)} \right\|_2^2 \tag{41}$$

$$\leq 3\eta_l^2 K \left\| \tilde{g}_t - \nabla F\left(\tilde{\boldsymbol{x}}_t\right) \right\|_2^2 + 3\eta_l^2 K \left\| \nabla F\left(\tilde{\boldsymbol{x}}_t\right) \right\|_2^2 + 3\eta_l^2 K \left( K L^2 + \frac{L_\sigma^2}{b'} \right) \sum_{k=0}^{K-1} \mathbb{E} \left\| \boldsymbol{x}_{t,\kappa-1}^{(m)} - \boldsymbol{x}_{t,\kappa}^{(m)} \right\|_2^2 \tag{42}$$

Obviously, according to the setting of the local learning rate in the description above, the inequality $3\eta_l^2 K \left( K L^2 + \frac{L_\sigma^2}{b'} \right) \leq \frac{1}{2}$ holds. Therefore, we can easily obtain the bound for the sum of the second norm of every iteration's difference, which is consistent with Equation (33). $\qquad\square$

### D.2 FULL CLIENT PARTICIPATION

**Theorem 4.** *Suppose that Assumption 1, 2 and 3 hold, and all clients participate in the training, i.e., $A = M$. Let the local updates $K \geq 1$, and the local learning rate $\eta_l$ and the global learning rate $\eta_s$ be $\eta_s \eta_l = \frac{1}{KL(1+2\tau)}$, where $\eta_l \leq \min\left( \frac{1}{2\sqrt{6}KL}, \frac{\sqrt{b'/K}}{4\sqrt{3}L_\sigma} \right)$. Therefore, the convergence rate of FedAMD for non-convex objectives should be*

$$\min_{t \in [T]} \left\| \nabla F\left(\tilde{\boldsymbol{x}}_t\right) \right\|_2^2 \leq O\left( \frac{1+2\tau}{T - \lfloor T/\tau \rfloor} \right) + O\left( \mathbf{1}_{\{b<n\}} \frac{\sigma^2}{Mb} \right) \tag{43}$$

*where we treat $F(\tilde{\boldsymbol{x}}_0) - F_*$ and $L$ as constants.*

*Proof.* When $p_t = 1$, according to Algorithm 1, there is no model update between two consecutive rounds, i.e., $\tilde{\boldsymbol{x}}_{t+1} = \tilde{\boldsymbol{x}}_t$.

Next, we consider the case when $p_t = 0$. Based on Lemma 3, we have

$$\mathbb{E}F(\tilde{\boldsymbol{x}}_{t+1}) - F(\tilde{\boldsymbol{x}}_t) \leq -\frac{\eta_s \eta_l K}{2} \left\| \nabla F\left(\tilde{\boldsymbol{x}}_t\right) \right\|_2^2 - \left( \frac{1}{2\eta_s \eta_l K} - \frac{L}{2} \right) \mathbb{E} \left\| \tilde{\boldsymbol{x}}_{t+1} - \tilde{\boldsymbol{x}}_t \right\|_2^2$$

$$+ \frac{\eta_s}{2\eta_l K} \mathbb{E} \left\| avg(\Delta \boldsymbol{x}_t) - \eta_l K \nabla F (\tilde{\boldsymbol{x}}_t) \right\|_2^2 \tag{44}$$

Knowing that when $p_t = 0$ and all clients involve in the training,

$$avg(\Delta \boldsymbol{x}_t) = \eta_l K \tilde{g}_t - \frac{\eta_l}{M} \sum_{m \in [M]} \sum_{k=0}^{K-1} \sum_{\kappa=0}^{k} \nabla f_m \left( \boldsymbol{x}_{t,\kappa-1}^{(m)}, \mathcal{B}_{m,\kappa}' \right)$$

$$+ \frac{\eta_l}{M} \sum_{m \in [M]} \sum_{k=0}^{K-1} \sum_{\kappa=0}^{k} \nabla f_m \left( \boldsymbol{x}_{t,\kappa}^{(m)}, \mathcal{B}_{m,\kappa}' \right), \tag{45}$$

we have the bound for $\mathbb{E} \left\| avg(\Delta \boldsymbol{x}_t) - \eta_l K \nabla F (\tilde{\boldsymbol{x}}_t) \right\|_2^2$ according to the following derivation:

$$\mathbb{E} \left\| avg(\Delta \boldsymbol{x}_t) - \eta_l K \nabla F (\tilde{\boldsymbol{x}}_t) \right\|_2^2 \tag{46}$$

$$= \mathbb{E} \left\| \eta_l K \left( \tilde{g}_t - \nabla F (\tilde{\boldsymbol{x}}_t) \right) + \frac{\eta_l}{M} \sum_{m \in [M]} \sum_{k=0}^{K-1} \sum_{\kappa=0}^{k} \left( \nabla f_m \left( \boldsymbol{x}_{t,\kappa}^{(m)}, \mathcal{B}_{m,\kappa}' \right) - \nabla f_m \left( \boldsymbol{x}_{t,\kappa-1}^{(m)}, \mathcal{B}_{m,\kappa}' \right) \right) \right\|_2^2 \tag{47}$$

$$\leq 2\eta_l^2 K^2 \left\| \tilde{g}_t - \nabla F (\tilde{\boldsymbol{x}}_t) \right\|_2^2$$

$$+ 2\mathbb{E} \left\| \frac{\eta_l}{M} \sum_{m \in [M]} \sum_{k=0}^{K-1} \sum_{\kappa=0}^{k} \left( \nabla f_m \left( \boldsymbol{x}_{t,\kappa}^{(m)}, \mathcal{B}_{m,\kappa}' \right) - \nabla f_m \left( \boldsymbol{x}_{t,\kappa-1}^{(m)}, \mathcal{B}_{m,\kappa}' \right) \right) \right\|_2^2 \tag{48}$$

$$= 2\eta_l^2 K^2 \left\| \tilde{g}_t - \nabla F (\tilde{\boldsymbol{x}}_t) \right\|_2^2 + 2\mathbb{E} \left\| \frac{\eta_l}{M} \sum_{m \in [M]} \sum_{k=0}^{K-1} \sum_{\kappa=0}^{k} \left( \nabla F_m \left( \boldsymbol{x}_{t,\kappa}^{(m)} \right) - \nabla F_m \left( \boldsymbol{x}_{t,\kappa-1}^{(m)} \right) \right) \right\|_2^2$$

$$+ 2\mathbb{E} \left\| \frac{\eta_l}{M} \sum_{m \in [M]} \sum_{k=0}^{K-1} \sum_{\kappa=0}^{k} \left( \nabla f_m \left( \boldsymbol{x}_{t,\kappa}^{(m)}, \mathcal{B}_{m,\kappa}' \right) - \nabla f_m \left( \boldsymbol{x}_{t,\kappa-1}^{(m)}, \mathcal{B}_{m,\kappa}' \right) - \nabla F_m \left( \boldsymbol{x}_{t,\kappa}^{(m)} \right) + \nabla F_m \left( \boldsymbol{x}_{t,\kappa-1}^{(m)} \right) \right) \right\|_2^2 \tag{49}$$

$$= 2\eta_l^2 K^2 \left\| \tilde{g}_t - \nabla F (\tilde{\boldsymbol{x}}_t) \right\|_2^2 + \frac{2\eta_l^2 K L^2}{M} \sum_{m \in [M]} \sum_{k=0}^{K-1} \sum_{\kappa=0}^{k} k \cdot \mathbb{E} \left\| \boldsymbol{x}_{t,\kappa}^{(m)} - \boldsymbol{x}_{t,\kappa-1}^{(m)} \right\|_2^2$$

$$+ 2\mathbb{E} \left\| \frac{\eta_l}{M} \sum_{m \in [M]} \sum_{k=0}^{K-1} \sum_{\kappa=0}^{k} \left( \nabla f_m \left( \boldsymbol{x}_{t,\kappa}^{(m)}, \mathcal{B}_{m,\kappa}' \right) - \nabla f_m \left( \boldsymbol{x}_{t,\kappa-1}^{(m)}, \mathcal{B}_{m,\kappa}' \right) - \nabla F_m \left( \boldsymbol{x}_{t,\kappa}^{(m)} \right) + \nabla F_m \left( \boldsymbol{x}_{t,\kappa-1}^{(m)} \right) \right) \right\|_2^2 \tag{50}$$

$$\leq 2\eta_l^2 K^2 \left\| \tilde{g}_t - \nabla F (\tilde{\boldsymbol{x}}_t) \right\|_2^2 + \frac{2\eta_l^2 K L^2}{M} \sum_{m \in [M]} \sum_{k=0}^{K-1} \sum_{\kappa=0}^{k} k \cdot \mathbb{E} \left\| \boldsymbol{x}_{t,\kappa}^{(m)} - \boldsymbol{x}_{t,\kappa-1}^{(m)} \right\|_2^2$$

$$+ \frac{2\eta_l^2}{M^2} \sum_{m \in [M]} \sum_{k=0}^{K-1} \sum_{\kappa=0}^{k} \frac{L_\sigma^2}{b'} \mathbb{E} \left\| \boldsymbol{x}_{t,\kappa}^{(m)} - \boldsymbol{x}_{t,\kappa-1}^{(m)} \right\|_2^2 \tag{51}$$

$$\leq 2\eta_l^2 K^2 \left\| \tilde{g}_t - \nabla F (\tilde{\boldsymbol{x}}_t) \right\|_2^2 + \frac{2\eta_l^2 K^3 L^2}{M} \sum_{m \in [M]} \sum_{k=0}^{K-1} \mathbb{E} \left\| \boldsymbol{x}_{t,k}^{(m)} - \boldsymbol{x}_{t,k-1}^{(m)} \right\|_2^2$$

$$+ \frac{2\eta_l^2 K L_\sigma^2}{M^2 b'} \sum_{m \in [M]} \sum_{k=0}^{K-1} \mathbb{E} \left\| \boldsymbol{x}_{t,k}^{(m)} - \boldsymbol{x}_{t,k-1}^{(m)} \right\|_2^2 \tag{52}$$

$$\leq 2\eta_l^2 K^2 \left\| \tilde{g}_t - \nabla F (\tilde{\boldsymbol{x}}_t) \right\|_2^2 + 12\eta_l^4 K^2 \left( \frac{L_\sigma^2}{Mb'} + K^2 L^2 \right) \left( \left\| \tilde{g}_t - \nabla F (\tilde{\boldsymbol{x}}_t) \right\|_2^2 + \left\| \nabla F (\tilde{\boldsymbol{x}}_t) \right\|_2^2 \right) \tag{53}$$

where Equation (48) follows $(\alpha + \beta)^2 \le 2\alpha^2 + 2\beta^2$; Equation (49) is based on variance expansion; Equation (50) is based on Cauchy-Schwarz inequality and Assumption 1; Equation (51) is based on Lemma 2 and Assumption 3; Equation (53) is based on Lemma 4. According to the constraints on the local learning rate, we can further simplify Equation (53) as

$$\mathbb{E}\left\|avg(\Delta \boldsymbol{x}_t) - \eta_l K \nabla F\left(\tilde{\boldsymbol{x}}_t\right)\right\|_2^2 \le 4\eta_l^2 K^2 \left\|\tilde{g}_t - \nabla F\left(\tilde{\boldsymbol{x}}_t\right)\right\|_2^2 + \frac{\eta_l^2 K^2}{2}\left\|\nabla F\left(\tilde{\boldsymbol{x}}_t\right)\right\|_2^2. \tag{54}$$

Plugging Equation (54) into Equation (44), we have

$$\begin{aligned}
\mathbb{E}F(\tilde{\boldsymbol{x}}_{t+1}) - F(\tilde{\boldsymbol{x}}_t) \le &- \frac{\eta_s \eta_l K}{4}\left\|\nabla F\left(\tilde{\boldsymbol{x}}_t\right)\right\|_2^2 - \left(\frac{1}{2\eta_s \eta_l K} - \frac{L}{2}\right)\mathbb{E}\left\|\tilde{\boldsymbol{x}}_{t+1} - \tilde{\boldsymbol{x}}_t\right\|_2^2 \\
&+ 2\eta_s \eta_l K \left\|\tilde{g}_t - \nabla F\left(\tilde{\boldsymbol{x}}_t\right)\right\|_2^2.
\end{aligned} \tag{55}$$

Let $\Lambda(t)$ indicate the most recent round where $p_{\Lambda(t)} = 1$ and $\Lambda(t) \ne t$. It is noted that recursively using $\Lambda(\cdot)$ can achieve the value of 0, i.e., $\underbrace{\Lambda(\Lambda(...\Lambda(t)))}_{\text{multiple } \Lambda} = 0$ By summing Equation (55) from $\Lambda(t)$ to $t-1$, we have

$$\mathbb{E}F(\tilde{\boldsymbol{x}}_t) - F\left(\tilde{\boldsymbol{x}}_{\Lambda(t)}\right) = \sum_{\theta=\Lambda(t)}^{t-1}\left(\mathbb{E}F(\tilde{\boldsymbol{x}}_{\theta+1}) - F(\tilde{\boldsymbol{x}}_\theta)\right) \tag{56}$$

$$\begin{aligned}
\le &- \frac{\eta_s \eta_l K}{4}\sum_{\theta=\Lambda(t)+1}^{t-1}\left\|\nabla F\left(\tilde{\boldsymbol{x}}_\theta\right)\right\|_2^2 - \left(\frac{1}{2\eta_s \eta_l K} - \frac{L}{2}\right)\sum_{\theta=\Lambda(t)+1}^{t-1}\mathbb{E}\left\|\tilde{\boldsymbol{x}}_{\theta+1} - \tilde{\boldsymbol{x}}_\theta\right\|_2^2 \\
&+ 2\eta_s \eta_l K \sum_{\theta=\Lambda(t)+1}^{t-1}\mathbb{E}\left\|\tilde{g}_\theta - \nabla F\left(\tilde{\boldsymbol{x}}_\theta\right)\right\|_2^2.
\end{aligned} \tag{57}$$

The bound for the last term of Equation (57) is

$$\mathbb{E}\left\|\tilde{g}_\theta - \nabla F\left(\tilde{\boldsymbol{x}}_\theta\right)\right\|_2^2 = \mathbb{E}\left\|\tilde{g}_{\Lambda(\theta)} - \nabla F\left(\tilde{\boldsymbol{x}}_\theta\right)\right\|_2^2 \tag{58}$$

$$= \mathbb{E}\left\|\tilde{g}_{\Lambda(\theta)} - \nabla F\left(\tilde{\boldsymbol{x}}_{\Lambda(\theta)}\right)\right\|_2^2 + \mathbb{E}\left\|\nabla F\left(\tilde{\boldsymbol{x}}_{\Lambda(\theta)}\right) - \nabla F\left(\tilde{\boldsymbol{x}}_\theta\right)\right\|_2^2 \tag{59}$$

$$\le \mathbb{E}\left\|\tilde{g}_{\Lambda(\theta)} - \nabla F\left(\tilde{\boldsymbol{x}}_{\Lambda(\theta)}\right)\right\|_2^2 + L^2 \mathbb{E}\left\|\tilde{\boldsymbol{x}}_\theta - \tilde{\boldsymbol{x}}_{\Lambda(\theta)}\right\|_2^2 \tag{60}$$

$$\le \mathbb{E}\left\|\tilde{g}_{\Lambda(\theta)} - \nabla F\left(\tilde{\boldsymbol{x}}_{\Lambda(\theta)}\right)\right\|_2^2 + L^2 \tau \sum_{\Xi=\Lambda(\theta)}^{\theta-1}\mathbb{E}\left\|\tilde{\boldsymbol{x}}_{\Xi+1} - \tilde{\boldsymbol{x}}_\Xi\right\|_2^2 \tag{61}$$

$$\le \mathbf{1}_{\{b<n\}}\frac{\sigma^2}{Mb} + L^2 \tau \sum_{\Xi=\Lambda(\theta)}^{\theta-1}\mathbb{E}\left\|\tilde{\boldsymbol{x}}_{\Xi+1} - \tilde{\boldsymbol{x}}_\Xi\right\|_2^2 \tag{62}$$

where Equation (59) is based on the variance expansion; Equation (60) is based on Assumption 1; Equation (61) is according to Cauchy-Schwarz inequality and $\theta - \Lambda(\theta) \le \tau$; Equation (62) follows Assumption 2. Based on the definition of $\Lambda(\cdot)$, for all $\theta \in \{\Lambda(t)+1, \dots, t-1\}$, $\Lambda(\theta) = \Lambda(t)$. Therefore, with Equation (62), Equation (57) can be further simplified as:

$$\mathbb{E}F(\tilde{\boldsymbol{x}}_t) - F\left(\tilde{\boldsymbol{x}}_{\Lambda(t)}\right) \tag{63}$$

$$\begin{aligned}
\le &- \frac{\eta_s \eta_l K}{4}\sum_{\theta=\Lambda(t)+1}^{t-1}\left\|\nabla F\left(\tilde{\boldsymbol{x}}_\theta\right)\right\|_2^2 - \left(\frac{1}{2\eta_s \eta_l K} - \frac{L}{2}\right)\sum_{\theta=\Lambda(t)+1}^{t-1}\mathbb{E}\left\|\tilde{\boldsymbol{x}}_{\theta+1} - \tilde{\boldsymbol{x}}_\theta\right\|_2^2 \\
&+ 2\eta_s \eta_l K (t - \Lambda(t) - 1) \cdot \mathbf{1}_{\{b<n\}}\frac{\sigma^2}{Mb} + 2\eta_s \eta_l K L^2 \tau \sum_{\theta=\Lambda(t)+1}^{t-1}\sum_{\Xi=\Lambda(\theta)}^{\theta-1}\mathbb{E}\left\|\tilde{\boldsymbol{x}}_{\Xi+1} - \tilde{\boldsymbol{x}}_\Xi\right\|_2^2
\end{aligned} \tag{64}$$

$$\le - \frac{\eta_s \eta_l K}{4}\sum_{\theta=\Lambda(t)+1}^{t-1}\left\|\nabla F\left(\tilde{\boldsymbol{x}}_\theta\right)\right\|_2^2 - \left(\frac{1}{2\eta_s \eta_l K} - \frac{L}{2} - 2\eta_s \eta_l K L^2 \tau^2\right)\sum_{\theta=\Lambda(t)+1}^{t-1}\mathbb{E}\left\|\tilde{\boldsymbol{x}}_{\theta+1} - \tilde{\boldsymbol{x}}_\theta\right\|_2^2$$

$$+ 2\eta_s\eta_l K \left(t - \Lambda(t) - 1\right) \cdot \mathbf{1}_{\{b<n\}} \frac{\sigma^2}{Mb} \tag{65}$$

Since $\eta_s\eta_l = \frac{1}{KL(1+2\tau)}$, Equation (65) can be further simplified as

$$\mathbb{E}F(\tilde{\boldsymbol{x}}_t) - F\left(\tilde{\boldsymbol{x}}_{\Lambda(t)}\right) \leq -\frac{\eta_s\eta_l K}{4} \sum_{\theta=\Lambda(t)+1}^{t-1} \|\nabla F(\tilde{\boldsymbol{x}}_\theta)\|_2^2 + 2\eta_s\eta_l K \left(t - \Lambda(t) - 1\right) \cdot \mathbf{1}_{\{b<n\}} \frac{\sigma^2}{Mb} \tag{66}$$

Therefore, based on the equation above, by summing up all $t \in \{T+1, \Lambda(T+1), \ldots, \tau\}$, we can obtain the following inequality:

$$F_* - F(\tilde{\boldsymbol{x}}_0) \leq \mathbb{E}F(\tilde{\boldsymbol{x}}_{T+1}) - F(\tilde{\boldsymbol{x}}_0) \tag{67}$$

$$\leq -\frac{\eta_s\eta_l K}{4} \sum_{t=0; t \bmod \tau = 0}^{T} \|\nabla F(\tilde{\boldsymbol{x}}_t)\|_2^2 + 2\eta_s\eta_l K \left(T - \lfloor T/\tau\rfloor\right) \cdot \mathbf{1}_{\{b<n\}} \frac{\sigma^2}{Mb} \tag{68}$$

Thus, we have

$$\frac{1}{T - \lfloor T/\tau\rfloor} \sum_{t=0; t \bmod \tau = 0}^{T} \|\nabla F(\tilde{\boldsymbol{x}}_t)\|_2^2 \leq \frac{4\left(F(\tilde{\boldsymbol{x}}_0) - F_*\right)}{\eta_s\eta_l K \left(T - \lfloor T/\tau\rfloor\right)} + 8 \cdot \mathbf{1}_{\{b<n\}} \frac{\sigma^2}{Mb} \tag{69}$$

By using the settings of the local learning rate and the global learning rate in the description, we can obtain the desired result. $\qquad\square$

### D.3 PARTIAL CLIENT PARTICIPATION

**Theorem 5.** *Suppose that Assumption 1, 2 and 3 hold. Let the local updates $K \geq 1$, and the local learning rate $\eta_l$ and the global learning rate $\eta_s$ be $\eta_s\eta_l = \frac{1}{KL}\left(1 + \frac{2M\tau}{A}\right)^{-1}$, where $\eta_l \leq \min\left(\frac{1}{2\sqrt{6}KL}, \frac{\sqrt{b'/K}}{4\sqrt{3}L_\sigma}\right)$. Therefore, the convergence rate of FedAMD for non-convex objectives should be*

$$\min_{t\in[T]} \|\nabla F(\tilde{\boldsymbol{x}}_t)\|_2^2 \leq O\left(\frac{1}{T - \lfloor T/\tau\rfloor}\left(1 + \frac{2M\tau}{A}\right)\right) + O\left(\mathbf{1}_{\{b<n\}} \frac{\sigma^2}{Mb}\right) \tag{70}$$

*where we treat $F(\tilde{\boldsymbol{x}}_0) - F_*$ and $L$ as constants.*

*Proof.* When $p_t = 1$, according to Algorithm 1, there is no model update between two consecutive rounds, i.e., $\tilde{\boldsymbol{x}}_{t+1} = \tilde{\boldsymbol{x}}_t$.

Next, we consider the case when $p_t = 0$. Based on Lemma 3, we have

$$\mathbb{E}F(\tilde{\boldsymbol{x}}_{t+1}) - F(\tilde{\boldsymbol{x}}_t) \leq -\frac{\eta_s\eta_l K}{2} \|\nabla F(\tilde{\boldsymbol{x}}_t)\|_2^2 - \left(\frac{1}{2\eta_s\eta_l K} - \frac{L}{2}\right) \mathbb{E}\|\tilde{\boldsymbol{x}}_{t+1} - \tilde{\boldsymbol{x}}_t\|_2^2$$
$$+ \frac{\eta_s}{2\eta_l K}\mathbb{E}\|avg(\Delta\boldsymbol{x}_t) - \eta_l K\nabla F(\tilde{\boldsymbol{x}}_t)\|_2^2 \tag{71}$$

Knowing that when $p_t = 0$ and a set of clients $\mathcal{A}$ involve in the training,

$$avg(\Delta\boldsymbol{x}_t) = \eta_l K\tilde{g}_t - \frac{\eta_l}{A}\sum_{i\in\mathcal{A}}\sum_{k=0}^{K-1}\sum_{\kappa=0}^{k}\nabla f_m\left(\boldsymbol{x}_{t,\kappa-1}^{(m)}, \mathcal{B}'_{m,\kappa}\right)$$
$$+ \frac{\eta_l}{A}\sum_{i\in\mathcal{A}}\sum_{k=0}^{K-1}\sum_{\kappa=0}^{k}\nabla f_m\left(\boldsymbol{x}_{t,\kappa}^{(m)}, \mathcal{B}'_{m,\kappa}\right), \tag{72}$$

we have the bound for $\mathbb{E}\|avg(\Delta\boldsymbol{x}_t) - \eta_l K\nabla F(\tilde{\boldsymbol{x}}_t)\|_2^2$ according to the following derivation:

$$\mathbb{E}\|avg(\Delta\boldsymbol{x}_t) - \eta_l K\nabla F(\tilde{\boldsymbol{x}}_t)\|_2^2 \tag{73}$$

$$= \mathbb{E} \left\| \eta_l K \left( \tilde{g}_t - \nabla F \left( \tilde{x}_t \right) \right) + \frac{\eta_l}{A} \sum_{i \in \mathcal{A}} \sum_{k=0}^{K-1} \sum_{\kappa=0}^{k} \left( \nabla f_i \left( x_{t,\kappa}^{(i)}, \mathcal{B}_{i,\kappa}' \right) - \nabla f_i \left( x_{t,\kappa-1}^{(i)}, \mathcal{B}_{i,\kappa}' \right) \right) \right\|_2^2 \quad (74)$$

$$\leq 2\eta_l^2 K^2 \left\| \tilde{g}_t - \nabla F \left( \tilde{x}_t \right) \right\|_2^2$$
$$+ 2\mathbb{E} \left\| \frac{\eta_l}{A} \sum_{i \in \mathcal{A}} \sum_{k=0}^{K-1} \sum_{\kappa=0}^{k} \left( \nabla f_i \left( x_{t,\kappa}^{(i)}, \mathcal{B}_{i,\kappa}' \right) - \nabla f_i \left( x_{t,\kappa-1}^{(i)}, \mathcal{B}_{i,\kappa}' \right) \right) \right\|_2^2 \quad (75)$$

$$= 2\eta_l^2 K^2 \left\| \tilde{g}_t - \nabla F \left( \tilde{x}_t \right) \right\|_2^2 + 2\mathbb{E} \left\| \frac{\eta_l}{A} \sum_{i \in \mathcal{A}} \sum_{k=0}^{K-1} \sum_{\kappa=0}^{k} \left( \nabla F_i \left( x_{t,\kappa}^{(i)} \right) - \nabla F_i \left( x_{t,\kappa-1}^{(i)} \right) \right) \right\|_2^2$$
$$+ 2\mathbb{E} \left\| \frac{\eta_l}{A} \sum_{i \in \mathcal{A}} \sum_{k=0}^{K-1} \sum_{\kappa=0}^{k} \left( \nabla f_i \left( x_{t,\kappa}^{(i)}, \mathcal{B}_{i,\kappa}' \right) - \nabla f_i \left( x_{t,\kappa-1}^{(i)}, \mathcal{B}_{i,\kappa}' \right) - \nabla F_i \left( x_{t,\kappa}^{(i)} \right) + \nabla F_i \left( x_{t,\kappa-1}^{(i)} \right) \right) \right\|_2^2$$
$$(76)$$

$$= 2\eta_l^2 K^2 \left\| \tilde{g}_t - \nabla F \left( \tilde{x}_t \right) \right\|_2^2 + \frac{2\eta_l^2 K L^2}{A} \sum_{i \in \mathcal{A}} \sum_{k=0}^{K-1} \sum_{\kappa=0}^{k} k \cdot \mathbb{E} \left\| x_{t,\kappa}^{(i)} - x_{t,\kappa-1}^{(i)} \right\|_2^2$$
$$+ 2\mathbb{E} \left\| \frac{\eta_l}{A} \sum_{i \in \mathcal{A}} \sum_{k=0}^{K-1} \sum_{\kappa=0}^{k} \left( \nabla f_i \left( x_{t,\kappa}^{(i)}, \mathcal{B}_{i,\kappa}' \right) - \nabla f_i \left( x_{t,\kappa-1}^{(i)}, \mathcal{B}_{i,\kappa}' \right) - \nabla F_i \left( x_{t,\kappa}^{(i)} \right) + \nabla F_i \left( x_{t,\kappa-1}^{(i)} \right) \right) \right\|_2^2$$
$$(77)$$

$$\leq 2\eta_l^2 K^2 \left\| \tilde{g}_t - \nabla F \left( \tilde{x}_t \right) \right\|_2^2 + \frac{2\eta_l^2 K L^2}{A} \sum_{i \in \mathcal{A}} \sum_{k=0}^{K-1} \sum_{\kappa=0}^{k} k \cdot \mathbb{E} \left\| x_{t,\kappa}^{(i)} - x_{t,\kappa-1}^{(i)} \right\|_2^2$$
$$+ \frac{2\eta_l^2}{A^2} \sum_{i \in \mathcal{A}} \sum_{k=0}^{K-1} \sum_{\kappa=0}^{k} \frac{L_\sigma^2}{b'} \mathbb{E} \left\| x_{t,\kappa}^{(i)} - x_{t,\kappa-1}^{(i)} \right\|_2^2 \quad (78)$$

$$= 2\eta_l^2 K^2 \left\| \tilde{g}_t - \nabla F \left( \tilde{x}_t \right) \right\|_2^2 + \frac{2\eta_l^2 K L^2}{M} \sum_{m \in [M]} \sum_{k=0}^{K-1} \sum_{\kappa=0}^{k} k \cdot \mathbb{E} \left\| x_{t,\kappa}^{(i)} - x_{t,\kappa-1}^{(i)} \right\|_2^2$$
$$+ \frac{2\eta_l^2}{AM} \sum_{m \in [M]} \sum_{k=0}^{K-1} \sum_{\kappa=0}^{k} \frac{L_\sigma^2}{b'} \mathbb{E} \left\| x_{t,\kappa}^{(i)} - x_{t,\kappa-1}^{(i)} \right\|_2^2 \quad (79)$$

$$\leq 2\eta_l^2 K^2 \left\| \tilde{g}_t - \nabla F \left( \tilde{x}_t \right) \right\|_2^2 + \frac{2\eta_l^2 K^3 L^2}{M} \sum_{m \in [M]} \sum_{k=0}^{K-1} \mathbb{E} \left\| x_{t,k}^{(m)} - x_{t,k-1}^{(m)} \right\|_2^2$$
$$+ \frac{2\eta_l^2 K L_\sigma^2}{AM b'} \sum_{m \in [M]} \sum_{k=0}^{K-1} \mathbb{E} \left\| x_{t,k}^{(m)} - x_{t,k-1}^{(m)} \right\|_2^2 \quad (80)$$

$$\leq 2\eta_l^2 K^2 \left\| \tilde{g}_t - \nabla F \left( \tilde{x}_t \right) \right\|_2^2 + 12\eta_l^4 K^2 \left( \frac{L_\sigma^2}{A b'} + K^2 L^2 \right) \left( \left\| \tilde{g}_t - \nabla F \left( \tilde{x}_t \right) \right\|_2^2 + \left\| \nabla F \left( \tilde{x}_t \right) \right\|_2^2 \right)$$
$$(81)$$

where Equation (75) follows $(\alpha + \beta)^2 \leq 2\alpha^2 + 2\beta^2$; Equation (76) is based on variance expansion; Equation (77) is based on Cauchy-Schwarz inequality and Assumption 1; Equation (78) is based on Lemma 2 and Assumption 3; Equation (79) is based on the setting of client selection, where each client is selected with a probability of $A/M$; Equation (81) is based on Lemma 4. According to the constraints on the local learning rate, we can further simplify Equation (81) as

$$\mathbb{E} \left\| avg(\Delta x_t) - \eta_l K \nabla F \left( \tilde{x}_t \right) \right\|_2^2 \leq 4\eta_l^2 K^2 \left\| \tilde{g}_t - \nabla F \left( \tilde{x}_t \right) \right\|_2^2 + \frac{\eta_l^2 K^2}{2} \left\| \nabla F \left( \tilde{x}_t \right) \right\|_2^2. \quad (82)$$

Plugging Equation (82) into Equation (71), we have

$$\mathbb{E} F(\tilde{x}_{t+1}) - F(\tilde{x}_t) \quad (83)$$

$$\leq -\frac{\eta_s \eta_l K}{4} \left\| \nabla F\left(\tilde{\boldsymbol{x}}_t\right) \right\|_2^2 - \left(\frac{1}{2\eta_s \eta_l K} - \frac{L}{2}\right) \mathbb{E} \left\| \tilde{\boldsymbol{x}}_{t+1} - \tilde{\boldsymbol{x}}_t \right\|_2^2 + 2\eta_s \eta_l K \mathbb{E} \left\| \tilde{g}_t - \nabla F\left(\tilde{\boldsymbol{x}}_t\right) \right\|_2^2 \tag{84}$$

$$= -\frac{\eta_s \eta_l K}{4} \left\| \nabla F\left(\tilde{\boldsymbol{x}}_t\right) \right\|_2^2 - \left(\frac{1}{2\eta_s \eta_l K} - \frac{L}{2}\right) \mathbb{E} \left\| \tilde{\boldsymbol{x}}_{t+1} - \tilde{\boldsymbol{x}}_t \right\|_2^2$$
$$+ 2\eta_s \eta_l K \left( \mathbb{E} \left\| \tilde{g}_t - \mathbb{E}\tilde{g}_t \right\|_2^2 + \mathbb{E} \left\| \mathbb{E}\tilde{g}_t - \nabla F\left(\tilde{\boldsymbol{x}}_t\right) \right\|_2^2 \right) \tag{85}$$

$$\leq -\frac{\eta_s \eta_l K}{4} \left\| \nabla F\left(\tilde{\boldsymbol{x}}_t\right) \right\|_2^2 - \left(\frac{1}{2\eta_s \eta_l K} - \frac{L}{2}\right) \mathbb{E} \left\| \tilde{\boldsymbol{x}}_{t+1} - \tilde{\boldsymbol{x}}_t \right\|_2^2 + 2\eta_s \eta_l K \cdot \mathbf{1}_{\{b<n\}} \frac{\sigma^2}{Mb}$$
$$+ 2\eta_s \eta_l K \mathbb{E} \left\| \mathbb{E}\tilde{g}_t - \nabla F\left(\tilde{\boldsymbol{x}}_t\right) \right\|_2^2 \tag{86}$$

where Equation (85) is based on variance expansion, and Equation (86) is based on Assumption 2.
By summing Equation (86) for all $t \in \{0, \ldots, T\}$, we have

$$F_* - F(\tilde{\boldsymbol{x}}_0) \leq \mathbb{E} F(\tilde{\boldsymbol{x}}_{T+1}) - F(\tilde{\boldsymbol{x}}_0) = \sum_{t=0}^{T} \left( \mathbb{E} F(\tilde{\boldsymbol{x}}_{t+1}) - F(\tilde{\boldsymbol{x}}_t) \right) \tag{87}$$

$$\leq -\frac{\eta_s \eta_l K}{4} \sum_{t=0; t \bmod \tau = 0}^{T} \left\| \nabla F\left(\tilde{\boldsymbol{x}}_t\right) \right\|_2^2 - \left(\frac{1}{2\eta_s \eta_l K} - \frac{L}{2}\right) \sum_{t=0; t \bmod \tau = 0}^{T} \mathbb{E} \left\| \tilde{\boldsymbol{x}}_{t+1} - \tilde{\boldsymbol{x}}_t \right\|_2^2$$
$$+ 2\eta_s \eta_l K \sum_{t=0; t \bmod \tau = 0}^{T} \mathbb{E} \left\| \mathbb{E}\tilde{g}_t - \nabla F\left(\tilde{\boldsymbol{x}}_t\right) \right\|_2^2 + 2\eta_s \eta_l K \left(T - \lfloor T/\tau \rfloor\right) \cdot \mathbf{1}_{\{b<n\}} \frac{\sigma^2}{Mb} \tag{88}$$

Let $\Lambda(t)$ indicate the most recent round where $p_{\Lambda(t)} = 1$ and $\Lambda(t) \neq t$. It is noted that recursively using $\Lambda(\cdot)$ can achieve the value of 0, i.e., $\underbrace{\Lambda(\Lambda(...\Lambda(t)))}_{\text{multiple } \Lambda} = 0$.

To find the bound for $\sum_{t=0; t \bmod \tau = 0}^{T} \mathbb{E} \left\| \mathbb{E}\tilde{g}_t - \nabla F\left(\tilde{\boldsymbol{x}}_t\right) \right\|_2^2$, the first step is to provide the bound for $\mathbb{E} \left\| \mathbb{E}\tilde{g}_t - \nabla F\left(\tilde{\boldsymbol{x}}_t\right) \right\|_2^2$. When $p_t = 1$, a client updates the caching gradient with a probability of $A/M$, and therefore, $\mathbb{E}\tilde{g}_t = \left(1 - \frac{A}{M}\right) \mathbb{E}\tilde{g}_{\Lambda(t)} + \frac{A}{M} \nabla F(\tilde{\boldsymbol{x}}_t)$. Based on this fact, the bound for $\mathbb{E} \left\| \mathbb{E}\tilde{g}_t - \nabla F\left(\tilde{\boldsymbol{x}}_t\right) \right\|_2^2$ can be derived as follows:

$$\mathbb{E} \left\| \mathbb{E}\tilde{g}_t - \nabla F\left(\tilde{\boldsymbol{x}}_t\right) \right\|_2^2 = \mathbb{E} \left\| \mathbb{E}\tilde{g}_{\Lambda(t)} - \nabla F\left(\tilde{\boldsymbol{x}}_t\right) \right\|_2^2 \tag{89}$$

$$= \mathbb{E} \left\| \left(1 - \frac{A}{M}\right) \mathbb{E} \left(\tilde{g}_{\Lambda(\Lambda(t))} - \nabla F\left(\tilde{\boldsymbol{x}}_{\Lambda(t)}\right)\right) + \left(\nabla F\left(\tilde{\boldsymbol{x}}_{\Lambda(t)}\right) - \nabla F\left(\tilde{\boldsymbol{x}}_t\right)\right) \right\|_2^2 \tag{90}$$

$$\leq \left(1 - \frac{A}{M}\right) \mathbb{E} \left\| \mathbb{E}\tilde{g}_{\Lambda(\Lambda(t))} - \nabla F\left(\tilde{\boldsymbol{x}}_{\Lambda(t)}\right) \right\|_2^2 + \frac{M}{A} \mathbb{E} \left\| \nabla F\left(\tilde{\boldsymbol{x}}_{\Lambda(t)}\right) - \nabla F\left(\tilde{\boldsymbol{x}}_t\right) \right\|_2^2 \tag{91}$$

$$\leq \sum_{\theta=0}^{\lfloor t/\tau \rfloor - 1} \left(1 - \frac{A}{M}\right)^{\lfloor t/\tau \rfloor - \theta} \cdot \frac{M}{A} L^2 \mathbb{E} \left\| \tilde{\boldsymbol{x}}_{\theta\tau} - \tilde{\boldsymbol{x}}_{(\theta+1)\tau} \right\|_2^2 + \frac{M}{A} L^2 \mathbb{E} \left\| \tilde{\boldsymbol{x}}_{\Lambda(t)} - \tilde{\boldsymbol{x}}_t \right\|_2^2 \tag{92}$$

where Equation (91) follows $(\alpha + \beta)^2 \leq \left(1 + \frac{1}{\gamma}\right) \alpha^2 + (1 + \gamma) \beta^2 - \left(\frac{1}{\sqrt{\gamma}}\alpha + \sqrt{\gamma}\beta\right)^2 \leq \left(1 + \frac{1}{\gamma}\right) \alpha^2 + (1 + \gamma) \beta^2$ and $\gamma = \frac{M-A}{A}$. With Equation (92), we sum up all $t \in \{1, \ldots, T\}$ and obtain the following result:

$$\sum_{t=0}^{T} \mathbb{E} \left\| \mathbb{E}\tilde{g}_t - \nabla F\left(\tilde{\boldsymbol{x}}_t\right) \right\|_2^2 \tag{93}$$

$$\leq \sum_{t=0}^{T} \left( \sum_{\theta=0}^{\lfloor t/\tau \rfloor - 1} \left(1 - \frac{A}{M}\right)^{\lfloor t/\tau \rfloor - \theta} \cdot \frac{M}{A} L^2 \mathbb{E} \left\| \tilde{\boldsymbol{x}}_{\theta\tau} - \tilde{\boldsymbol{x}}_{(\theta+1)\tau} \right\|_2^2 + \frac{M}{A} L^2 \mathbb{E} \left\| \tilde{\boldsymbol{x}}_{\Lambda(t)} - \tilde{\boldsymbol{x}}_t \right\|_2^2 \right) \tag{94}$$

$$\leq \sum_{\theta=0}^{\lfloor T/\tau \rfloor -1} \frac{M(M-A)}{A^2} L^2 \tau \mathbb{E} \left\| \tilde{\boldsymbol{x}}_{\theta\tau} - \tilde{\boldsymbol{x}}_{(\theta+1)\tau} \right\|_2^2 + \frac{M}{A} L^2 \sum_{t=0}^{T} \left\| \tilde{\boldsymbol{x}}_t - \tilde{\boldsymbol{x}}_{\Lambda(t)} \right\|_2^2 \tag{95}$$

$$\leq \sum_{\theta=0}^{\lfloor T/\tau \rfloor -1} \frac{M(M-A)}{A^2} L^2 \tau^2 \sum_{\Xi=\theta\tau+1}^{(\theta+1)\tau-1} \mathbb{E} \left\| \tilde{\boldsymbol{x}}_{\Xi+1} - \tilde{\boldsymbol{x}}_{\Xi} \right\|_2^2$$

$$+ \frac{M}{A} L^2 \sum_{t=0}^{T} (t - \Lambda(t) - 1) \sum_{\Xi=\Lambda(t)+1}^{t-1} \left\| \tilde{\boldsymbol{x}}_{\Xi+1} - \tilde{\boldsymbol{x}}_{\Xi} \right\|_2^2 \tag{96}$$

$$\leq \frac{M(M-A)}{A^2} L^2 \tau^2 \sum_{t=0; t \bmod \tau=0}^{T-1} \mathbb{E} \left\| \tilde{\boldsymbol{x}}_{t+1} - \tilde{\boldsymbol{x}}_t \right\|_2^2 + \frac{M}{A} L^2 \tau^2 \sum_{t=0; t \bmod \tau=0}^{T-1} \mathbb{E} \left\| \tilde{\boldsymbol{x}}_{t+1} - \tilde{\boldsymbol{x}}_t \right\|_2^2 \tag{97}$$

where Equation (95) follows that, for all $\theta \in \{0, \ldots, \lfloor T/\tau \rfloor - 1\}$, the coefficient for $\frac{M}{A} L^2$ includes $\left(1 - \frac{A}{M}\right), \ldots, \left(1 - \frac{A}{M}\right)^{\lfloor T/\tau \rfloor - \theta}$, and each of them has a maximum of $\tau$ $t$s, meaning that the upper bound of the coefficient should be

$$\tau \left( \left(1 - \frac{A}{M}\right) + \cdots + \left(1 - \frac{A}{M}\right)^{\lfloor T/\tau \rfloor - \theta} \right) \leq \tau \cdot \frac{M}{2A} \left(1 - \frac{A}{M}\right); \tag{98}$$

Equation (96) follows Cauchy-Schwarz inequality.

Plugging Equation (97) back to Equation (88), we have:

$$F_* - F(\tilde{\boldsymbol{x}}_0) \leq -\frac{\eta_s \eta_l K}{4} \sum_{t=0; t \bmod \tau=0}^{T} \left\| \nabla F(\tilde{\boldsymbol{x}}_t) \right\|_2^2$$

$$- \left( \frac{1}{2\eta_s \eta_l K} - \frac{L}{2} - \eta_s \eta_l K L^2 \tau^2 \frac{M^2}{A^2} \right) \sum_{t=0; t \bmod \tau=0}^{T} \mathbb{E} \left\| \tilde{\boldsymbol{x}}_{t+1} - \tilde{\boldsymbol{x}}_t \right\|_2^2$$

$$+ 2\eta_s \eta_l K \left( T - \lfloor T/\tau \rfloor \right) \cdot \mathbf{1}_{\{b<n\}} \frac{\sigma^2}{Mb} \tag{99}$$

Since $\eta_s \eta_l = \frac{1}{KL} \left(1 + \frac{2M\tau}{A}\right)^{-1}$, $\frac{1}{2\eta_s \eta_l K} - \frac{L}{2} - \eta_s \eta_l K L^2 \tau^2 \frac{M^2}{A^2} \geq 0$ such that the term $\sum_{t=0; t \bmod \tau=0}^{T} \mathbb{E} \left\| \tilde{\boldsymbol{x}}_{t+1} - \tilde{\boldsymbol{x}}_t \right\|_2^2$ can be omitted in Equation (99). Hence, we can easily obtain the following inequality:

$$\frac{1}{T - \lfloor T/\tau \rfloor} \sum_{t=0; t \bmod \tau=0}^{T} \left\| \nabla F(\tilde{\boldsymbol{x}}_t) \right\|_2^2 \leq \frac{4 \left( F(\tilde{\boldsymbol{x}}_0) - F_* \right)}{\eta_s \eta_l K \left( T - \lfloor T/\tau \rfloor \right)} + 8 \cdot \mathbf{1}_{\{b<n\}} \frac{\sigma^2}{Mb} \tag{100}$$

By using the settings of the local learning rate and the global learning rate in the description, we can obtain the desired result. $\qquad \square$

# E  PROOFS UNDER CONSTANT PROBABILISTIC SETTINGS

## E.1  PRELIMINARY

**Lemma 5.** *Suppose that Assumption 1 holds, and $p_t \in (0,1)$. Let $\tilde{g}_t$ be the definition of Line 9 of Algorithm 1, i.e., the average of the caching gradients. Therefore, the recursive expression for $\{\tilde{g}_t\}_{t \geq 0}$ in the expected form is*

$$\mathbb{E}\tilde{g}_t = \begin{cases} \left(1 - \frac{A}{M}p_{t-1}\right) \cdot \mathbb{E}\tilde{g}_{t-1} + \frac{A}{M}p_{t-1} \cdot \nabla F\left(\tilde{\boldsymbol{x}}_{t-1}\right), & t > 0 \\ \nabla F\left(\tilde{\boldsymbol{x}}_0\right), & t = 0 \end{cases} \tag{101}$$

*Furthermore, when $t > 0$ we can obtain the following inequality:*

$$\mathbb{E}\left\|\mathbb{E}\tilde{g}_t - \nabla F\left(\tilde{\boldsymbol{x}}_t\right)\right\|_2^2 \leq \left(1 - \frac{A}{M}p_{t-1}\right) \cdot \mathbb{E}\left\|\mathbb{E}\tilde{g}_{t-1} - \nabla F\left(\tilde{\boldsymbol{x}}_{t-1}\right)\right\|_2^2 + \frac{M}{Ap_{t-1}} \cdot L^2 \cdot \mathbb{E}\left\|\tilde{\boldsymbol{x}}_t - \tilde{\boldsymbol{x}}_{t-1}\right\|_2^2. \tag{102}$$

*As for $t = 0$, we have $\mathbb{E}\left\|\mathbb{E}\tilde{g}_t - \nabla F\left(\tilde{\boldsymbol{x}}_t\right)\right\|_2^2 = 0$.*

*Proof.* According to the definition of Line 9 of Algorithm 1, $\tilde{g}_{t+1} = avg\left(v_{t+1}\right) = \frac{1}{M}\sum_{m \in [M]} v_{t+1}^{(m)}$. Hence, for each element in $v_{t+1}$, i.e., $v_{t+1}^{(m)}$, where $m \in [M]$, they have a probability of $\left(1 - \frac{A}{M}p_t\right)$ to retain the previous value, or otherwise update as anchor clients using large batches. Thus, the expected value for $\mathbb{E}v_{t+1}^{(m)}$ is:

$$\mathbb{E}v_{t+1}^{(m)} = \frac{A}{M}p_t \cdot \mathbb{E}\nabla f_m\left(\tilde{\boldsymbol{x}}_t, \mathcal{B}_{m,t}\right)] + \left(1 - \frac{A}{M}p_t\right) \cdot \mathbb{E}v_t^{(m)} \tag{103}$$

$$= \frac{A}{M}p_t \cdot \nabla F_m\left(\tilde{\boldsymbol{x}}_t\right) + \left(1 - \frac{A}{M}p_t\right) \cdot \mathbb{E}v_t^{(m)} \tag{104}$$

Therefore, we have

$$\mathbb{E}\tilde{g}_{t+1} = \frac{1}{M}\sum_{m=1}^{M}\mathbb{E}v_{t+1}^{(m)} = \frac{A}{M}p_t \cdot \nabla F\left(\tilde{\boldsymbol{x}}_t\right) + \left(1 - \frac{A}{M}p_t\right) \cdot \mathbb{E}\tilde{g}_t \tag{105}$$

It is worth noting that $\mathbb{E}\tilde{g}_0 = \nabla F\left(\tilde{\boldsymbol{x}}_0\right)$ as it is initialized at the beginning of the training, i.e., Line 2 – 4 in Algorithm 1. Therefore, $\mathbb{E}\left\|\mathbb{E}\tilde{g}_t - \nabla F\left(\tilde{\boldsymbol{x}}_t\right)\right\|_2^2 = 0$.

Next, we find the recursive bound for $\mathbb{E}\left\|\mathbb{E}\tilde{g}_{t+1} - \nabla F\left(\tilde{\boldsymbol{x}}_{t+1}\right)\right\|_2^2$:

$$\mathbb{E}\left\|\mathbb{E}\tilde{g}_{t+1} - \nabla F\left(\tilde{\boldsymbol{x}}_{t+1}\right)\right\|_2^2 \tag{106}$$

$$= \mathbb{E}\left\|\left(1 - \frac{A}{M}p_t\right) \cdot \left(\mathbb{E}\tilde{g}_t - \nabla F\left(\tilde{\boldsymbol{x}}_t\right)\right) + \nabla F\left(\tilde{\boldsymbol{x}}_t\right) - \nabla F\left(\tilde{\boldsymbol{x}}_{t+1}\right)\right\|_2^2 \tag{107}$$

$$\leq \left(1 + \frac{Ap_t}{M - Ap_t}\right)\left(1 - \frac{A}{M}p_t\right)^2 \mathbb{E}\left\|\mathbb{E}\tilde{g}_t - \nabla F\left(\tilde{\boldsymbol{x}}_t\right)\right\|_2^2$$
$$+ \left(1 + \frac{M - Ap_t}{Ap_t}\right)\mathbb{E}\left\|\nabla F\left(\tilde{\boldsymbol{x}}_t\right) - \nabla F\left(\tilde{\boldsymbol{x}}_{t+1}\right)\right\|_2^2 \tag{108}$$

$$\leq \left(1 - \frac{A}{M}p_t\right)\mathbb{E}\left\|\mathbb{E}\tilde{g}_t - \nabla F\left(\tilde{\boldsymbol{x}}_t\right)\right\|_2^2 + \frac{M}{Ap_t}L^2\mathbb{E}\left\|\tilde{\boldsymbol{x}}_{t+1} - \tilde{\boldsymbol{x}}_t\right\|_2^2 \tag{109}$$

where Equation (108) follows $(\alpha + \beta)^2 \leq \left(1 + \frac{1}{\gamma}\right)\alpha^2 + (1 + \gamma)\beta^2 - \left(\frac{1}{\sqrt{\gamma}}\alpha + \sqrt{\gamma}\beta\right)^2 \leq \left(1 + \frac{1}{\gamma}\right)\alpha^2 + (1 + \gamma)\beta^2$, and Equation (109) follows Assumption 1. $\square$

**Lemma 6.** *Suppose that Assumption 1, 2 and 3 hold. Let the local learning rate satisfy $\eta_l \leq \min\left(\frac{1}{2\sqrt{3}KL}, \frac{1}{2\sqrt{3}L_\sigma^2}\sqrt{\frac{b'}{K}}\right)$. With FedAMD, $\sum_{k=0}^{K-1}\left\|\boldsymbol{x}_{t,k}^{(m)} - \boldsymbol{x}_{t,k-1}^{(m)}\right\|_2^2$ represents the sum of the*

*second norm of every iteration's difference. Therefore, the bound for such a summation in the expected form should be*

$$\sum_{k=0}^{K-1} \mathbb{E} \left\| \boldsymbol{x}_{t,k}^{(m)} - \boldsymbol{x}_{t,k-1}^{(m)} \right\|_2^2 \leq 2\eta_l^2(K+1)\frac{\sigma^2}{Mb} + 6\eta_l^2(K+1) \left\| \mathbb{E}\tilde{g}_t - \nabla F(\tilde{\boldsymbol{x}}_t) \right\|_2^2 + 6\eta_l^2(K+1) \left\| \nabla F(\tilde{\boldsymbol{x}}_t) \right\|_2^2 \tag{110}$$

*Proof.* According to Equation (31), the update at $(k-1)$-th iteration is

$$\boldsymbol{x}_{t,k}^{(m)} - \boldsymbol{x}_{t,k-1}^{(m)} = -\eta_l g_{t,k}^{(m)} = -\eta_l \left( g_{t,0}^{(m)} - \sum_{\kappa=0}^{k-1} \nabla f_m \left( \boldsymbol{x}_{t,\kappa-1}^{(m)}, \mathcal{B}_{m,\kappa}' \right) + \sum_{\kappa=0}^{k-1} \nabla f_m \left( \boldsymbol{x}_{t,\kappa}^{(m)}, \mathcal{B}_{m,\kappa}' \right) \right). \tag{111}$$

To find the bound for the expected value of its second norm, the analysis is presented as follows:

$$\mathbb{E} \left\| \boldsymbol{x}_{t,k}^{(m)} - \boldsymbol{x}_{t,k-1}^{(m)} \right\|_2^2 \tag{112}$$

$$= \eta_l^2 \mathbb{E} \left\| \tilde{g}_t - \sum_{\kappa=0}^{k-1} \nabla f_m \left( \boldsymbol{x}_{t,\kappa-1}^{(m)}, \mathcal{B}_{m,\kappa}' \right) + \sum_{\kappa=0}^{k-1} \nabla f_m \left( \boldsymbol{x}_{t,\kappa}^{(m)}, \mathcal{B}_{m,\kappa}' \right) \right\|_2^2 \tag{113}$$

$$= \eta_l^2 \mathbb{E} \left\| \tilde{g}_t - \mathbb{E}\tilde{g}_t - \sum_{\kappa=0}^{k-1} \left( \nabla f_m \left( \boldsymbol{x}_{t,\kappa-1}^{(m)}, \mathcal{B}_{m,\kappa}' \right) - \nabla f_m \left( \boldsymbol{x}_{t,\kappa}^{(m)}, \mathcal{B}_{m,\kappa}' \right) - \nabla F_m \left( \boldsymbol{x}_{t,\kappa-1}^{(m)} \right) + \nabla F_m \left( \boldsymbol{x}_{t,\kappa}^{(m)} \right) \right) \right\|_2^2$$
$$+ \eta_l^2 \mathbb{E} \left\| \mathbb{E}\tilde{g}_t - \sum_{\kappa=0}^{k-1} \nabla F_m \left( \boldsymbol{x}_{t,\kappa-1}^{(m)} \right) + \sum_{\kappa=0}^{k-1} \nabla F_m \left( \boldsymbol{x}_{t,\kappa}^{(m)} \right) \right\|_2^2 \tag{114}$$

$$= \eta_l^2 \left( \mathbf{1}_{\{b<n\}} \frac{\sigma^2}{Mb} + \sum_{\kappa=0}^{k-1} \frac{L_\sigma^2}{b'} \mathbb{E} \left\| \boldsymbol{x}_{t,\kappa}^{(m)} - \boldsymbol{x}_{t,\kappa-1}^{(m)} \right\|_2^2 \right)$$
$$+ \eta_l^2 \mathbb{E} \left\| \mathbb{E}\tilde{g}_t - \sum_{\kappa=0}^{k-1} \left( \nabla F_m \left( \boldsymbol{x}_{t,\kappa-1}^{(m)} \right) - \nabla F_m \left( \boldsymbol{x}_{t,\kappa}^{(m)} \right) \right) \right\|_2^2 \tag{115}$$

$$= \eta_l^2 \left( \mathbf{1}_{\{b<n\}} \frac{\sigma^2}{Mb} + \sum_{\kappa=0}^{k-1} \frac{L_\sigma^2}{b'} \left\| \boldsymbol{x}_{t,\kappa}^{(m)} - \boldsymbol{x}_{t,\kappa-1}^{(m)} \right\|_2^2 \right)$$
$$+ \eta_l^2 \mathbb{E} \left\| \mathbb{E}\tilde{g}_t - \nabla F(\tilde{\boldsymbol{x}}_t) + \nabla F(\tilde{\boldsymbol{x}}_t) + \sum_{\kappa=0}^{k-1} \left( \nabla F_m \left( \boldsymbol{x}_{t,\kappa}^{(m)} \right) + \nabla F_m \left( \boldsymbol{x}_{t,\kappa-1}^{(m)} \right) \right) \right\|_2^2 \tag{116}$$

$$\leq \eta_l^2 \left( \mathbf{1}_{\{b<n\}} \frac{\sigma^2}{Mb} + \sum_{\kappa=0}^{k-1} \frac{L_\sigma^2}{b'} \mathbb{E} \left\| \boldsymbol{x}_{t,\kappa}^{(m)} - \boldsymbol{x}_{t,\kappa-1}^{(m)} \right\|_2^2 \right)$$
$$+ 3\eta_l^2 \cdot \left\| \mathbb{E}\tilde{g}_t - \nabla F(\tilde{\boldsymbol{x}}_t) \right\|_2^2 + 3\eta_l^2 \left\| \nabla F(\tilde{\boldsymbol{x}}_t) \right\|_2^2 + 3\eta_l^2 K \sum_{\kappa=0}^{k-1} L^2 \left\| \boldsymbol{x}_{t,\kappa}^{(m)} - \boldsymbol{x}_{t,\kappa-1}^{(m)} \right\|_2^2 \tag{117}$$

$$= \eta_l^2 \mathbf{1}_{\{b<n\}} \frac{\sigma^2}{Mb} + 3\eta_l^2 \cdot \mathbb{E} \left\| \mathbb{E}\tilde{g}_t - \nabla F(\tilde{\boldsymbol{x}}_t) \right\|_2^2 + 3\eta_l^2 \left\| \nabla F(\tilde{\boldsymbol{x}}_t) \right\|_2^2$$
$$+ 3\eta_l^2 \left( KL^2 + \frac{L_\sigma^2}{b'} \right) \sum_{\kappa=0}^{k-1} \mathbb{E} \left\| \boldsymbol{x}_{t,\kappa}^{(m)} - \boldsymbol{x}_{t,\kappa-1}^{(m)} \right\|_2^2 \tag{118}$$

where Equation (114) is based on the variance expansion on the first term of Equation (113); Equation (117) is based on Cauchy-Schwarz inequality.

Therefore, by summing Equation (118) for $k = 1, \ldots, K$, we have

$$\mathbb{E} \sum_{k=0}^{K-1} \left\| \boldsymbol{x}_{t,k}^{(m)} - \boldsymbol{x}_{t,k-1}^{(m)} \right\|_2^2 \leq \sum_{k=0}^{K} \mathbb{E} \left\| \boldsymbol{x}_{t,k}^{(m)} - \boldsymbol{x}_{t,k-1}^{(m)} \right\|_2^2 \tag{119}$$

$$\leq \eta_l^2 (K+1) \mathbf{1}_{\{b<n\}} \frac{\sigma^2}{Mb} + 3\eta_l^2 (K+1) \|\mathbb{E}\tilde{g}_t - \nabla F(\tilde{\boldsymbol{x}}_t)\|_2^2 + 3\eta_l^2 (K+1) \|\nabla F(\tilde{\boldsymbol{x}}_t)\|_2^2$$

$$+ 3\eta_l^2 K \left( KL^2 + \frac{L_\sigma^2}{b'} \right) \sum_{k=0}^{K-1} \mathbb{E} \left\| \boldsymbol{x}_{t,k}^{(m)} - \boldsymbol{x}_{t,k-1}^{(m)} \right\|_2^2 \tag{120}$$

Obviously, according to the setting of the local learning rate in the description above, the inequality $3\eta_l^2 K \left( KL^2 + \frac{L_\sigma^2}{b'} \right) \leq \frac{1}{2}$ holds. Therefore, we can easily obtain the bound for the sum of the second norm of every iteration's difference, which is consistent with Equation (110). $\qquad\square$

## E.2 Proofs for Non-convex Objectives

The following lemma provides a recursive expression on $\mathbb{E}F(\tilde{\boldsymbol{x}}_{t+1}) - F(\tilde{\boldsymbol{x}}_t)$ for time-varying probability settings.

**Lemma 7.** *Suppose that Assumption 1, 2 and 3 hold, and the time-varying probability sequence $\{p_t \in (0,1)\}_{t\geq 0}$. Let the local updates $K \geq 1$, and the local learning rate $\eta_l \leq \min\left(\frac{1}{2\sqrt{6}KL}, \frac{\sqrt{b'/K}}{2\sqrt{3}L_\sigma}\right)$. With the model training using FedAMD, the recursive function between $F(\tilde{\boldsymbol{x}}_{t+1})$ and $F(\tilde{\boldsymbol{x}}_t)$ in expected form is*

$$\mathbb{E}F(\tilde{\boldsymbol{x}}_{t+1}) - F(\tilde{\boldsymbol{x}}_t) \leq -\frac{\eta_s \eta_l K}{4}\left(1 - (p_t)^A\right)\|\nabla F(\tilde{\boldsymbol{x}}_t)\|_2^2 - \left(\frac{1}{2\eta_s \eta_l K} - \frac{L}{2}\right)\|\tilde{\boldsymbol{x}}_{t+1} - \tilde{\boldsymbol{x}}_t\|_2^2$$

$$+ 4\eta_s \eta_l K \left(1 - (p_t)^A\right)\|\mathbb{E}\tilde{g}_t - \nabla F(\tilde{\boldsymbol{x}}_t)\|_2^2$$

$$+ 3\eta_s \eta_l K \left(1 - (p_t)^A\right)\mathbf{1}_{\{b<n\}}\frac{\sigma^2}{Mb} \tag{121}$$

*Proof.* According to Lemma 4, we have:

$$\mathbb{E}F(\tilde{\boldsymbol{x}}_{t+1}) - F(\tilde{\boldsymbol{x}}_t) \tag{122}$$

$$\leq -\frac{\eta_s \eta_l K}{2}\|\nabla F(\tilde{\boldsymbol{x}}_t)\|_2^2 - \left(\frac{1}{2\eta_s \eta_l K} - \frac{L}{2}\right)\|\tilde{\boldsymbol{x}}_{t+1} - \tilde{\boldsymbol{x}}_t\|_2^2 + \frac{\eta_s}{2\eta_l K}\mathbb{E}\|\Delta\boldsymbol{x}_t - \eta_l K\nabla F(\tilde{\boldsymbol{x}}_t)\|_2^2 \tag{123}$$

When $|\Delta\boldsymbol{x}_t| = 0$, the probability will be $(1-q)^a$, and when $|\Delta\boldsymbol{x}_t| \neq 0$, the probability will be $1 - (1-q)^a$. Next, we find the bound for the third term of Equation (123), i.e., $\mathbb{E}\|\Delta\boldsymbol{x}_t - \eta_l K\nabla F(\tilde{\boldsymbol{x}}_t)\|_2^2$. By Lemma 1, we have the following derivation:

$$\mathbb{E}\|\Delta\boldsymbol{x}_t - \eta_l K\nabla F(\tilde{\boldsymbol{x}}_t)\|_2^2 \tag{124}$$

$$\leq \left(1 - (p_t)^A\right)\frac{1}{M}\sum_{m=1}^M \mathbb{E}\left\|\Delta\boldsymbol{x}_t^{(m)} - \eta_l K\nabla F(\tilde{\boldsymbol{x}}_t)\right\|_2^2 + (p_t)^A\|\eta_l K\nabla F(\tilde{\boldsymbol{x}}_t)\|_2^2 \tag{125}$$

$$= \left(1 - (p_t)^A\right)\frac{1}{M}\sum_{m=1}^M \left(\mathbb{E}\left\|\Delta\boldsymbol{x}_t^{(m)} - \mathbb{E}\Delta\boldsymbol{x}_t^{(m)}\right\|_2^2 + \mathbb{E}\left\|\mathbb{E}\Delta\boldsymbol{x}_t^{(m)} - \eta_l K\nabla F(\tilde{\boldsymbol{x}}_t)\right\|_2^2\right)$$

$$+ (p_t)^A \eta_l^2 K^2 \|\nabla F(\tilde{\boldsymbol{x}}_t)\|_2^2 \tag{126}$$

where Equation (126) follows variance equation. To find the bound for Equation (126), we first analyze its first term, i.e., $\mathbb{E}\left\|\Delta\boldsymbol{x}_t^{(m)} - \mathbb{E}\Delta\boldsymbol{x}_t^{(m)}\right\|_2^2$. According to Section C, we have:

$$\mathbb{E}\left\|\Delta\boldsymbol{x}_t^{(m)} - \mathbb{E}\Delta\boldsymbol{x}_t^{(m)}\right\|_2^2 \tag{127}$$

$$\leq 2\eta_l^2 K^2 \mathbb{E}\|\tilde{g}_t - \mathbb{E}\tilde{g}_t\|_2^2 + 2\eta_l^2 \sum_{k=0}^{K-1}(K-1)\frac{L_\sigma^2}{b'}\left\|\boldsymbol{x}_{t,k}^{(m)} - \boldsymbol{x}_{t,k-1}^{(m)}\right\|_2^2 \tag{128}$$

$$\leq 2\eta_l^2 K^2 \left(1 + 2\eta_l^2 \frac{L_\sigma^2}{b'}\right)\mathbf{1}_{\{b<n\}}\frac{\sigma^2}{Mb} + 12\eta_l^4 K^2 \frac{L_\sigma^2}{b'}\left(\|\mathbb{E}\tilde{g}_t - \nabla F(\tilde{\boldsymbol{x}}_t)\|_2^2 + \|\nabla F(\tilde{\boldsymbol{x}}_t)\|_2^2\right) \tag{129}$$

where Equation (129) follows Lemma 6. According to the local learning rate setting in the description, we have

$$\mathbb{E} \left\| \Delta \boldsymbol{x}_t^{(m)} - \mathbb{E} \Delta \boldsymbol{x}_t^{(m)} \right\|_2^2 \leq 4 \eta_l^2 K^2 \cdot \mathbf{1}_{\{b<n\}} \frac{\sigma^2}{Mb} \tag{130}$$

$$+ 12 \eta_l^4 K^2 \frac{L_\sigma^2}{b'} \left( \left\| \mathbb{E} \tilde{g}_t - \nabla F\left(\tilde{\boldsymbol{x}}_t\right) \right\|_2^2 + \left\| \nabla F\left(\tilde{\boldsymbol{x}}_t\right) \right\|_2^2 \right) \tag{131}$$

After finding the bound for the first term of Equation (126), we now give the bound for its second term, i.e., $\mathbb{E} \left\| \mathbb{E} \Delta \boldsymbol{x}_t^{(m)} - \eta_l K \nabla F\left(\tilde{\boldsymbol{x}}_t\right) \right\|_2^2$.

$$\mathbb{E} \left\| \mathbb{E} \Delta \boldsymbol{x}_t^{(m)} - \eta_l K \nabla F\left(\tilde{\boldsymbol{x}}_t\right) \right\|_2^2 \tag{132}$$

$$= \mathbb{E} \left\| \eta_l K \left( \mathbb{E} \tilde{g}_t - \nabla F\left(\tilde{\boldsymbol{x}}_t\right) \right) + \eta_l \sum_{k=0}^{K-1} \sum_{\kappa=0}^{k} \left( \nabla F_m \left( \boldsymbol{x}_{t,\kappa}^{(m)} \right) - \nabla F_m \left( \boldsymbol{x}_{t,\kappa-1}^{(m)} \right) \right) \right\|_2^2 \tag{133}$$

$$\leq 2 \eta_l^2 K^2 \left\| \mathbb{E} \tilde{g}_t - \nabla F\left(\tilde{\boldsymbol{x}}_t\right) \right\|_2^2 + 2 \eta_l^2 \left\| \sum_{k=0}^{K-1} \sum_{\kappa=0}^{k} \left( \nabla F_m \left( \boldsymbol{x}_{t,\kappa}^{(m)} \right) - \nabla F_m \left( \boldsymbol{x}_{t,\kappa-1}^{(m)} \right) \right) \right\|_2^2 \tag{134}$$

$$\leq 2 \eta_l^2 K^2 \left\| \mathbb{E} \tilde{g}_t - \nabla F\left(\tilde{\boldsymbol{x}}_t\right) \right\|_2^2 + 2 \eta_l^2 K L^2 \sum_{k=0}^{K-1} \sum_{\kappa=0}^{k} k \left\| \boldsymbol{x}_{t,\kappa}^{(m)} - \boldsymbol{x}_{t,\kappa-1}^{(m)} \right\|_2^2 \tag{135}$$

$$\leq 2 \eta_l^2 K^2 \left\| \mathbb{E} \tilde{g}_t - \nabla F\left(\tilde{\boldsymbol{x}}_t\right) \right\|_2^2 + 2 \eta_l^2 K \frac{K(K-1)}{2} L^2 \sum_{k=0}^{K-1} \left\| \boldsymbol{x}_{t,k}^{(m)} - \boldsymbol{x}_{t,k-1}^{(m)} \right\|_2^2 \tag{136}$$

$$\leq 2 \eta_l^4 K^4 L^2 \cdot \mathbf{1}_{\{b<n\}} \frac{\sigma^2}{Mb} + 2 \eta_l^2 K^2 \left( 1 + 3 \eta_l^2 K^2 L^2 \right) \left\| \mathbb{E} \tilde{g}_t - \nabla F\left(\tilde{\boldsymbol{x}}_t\right) \right\|_2^2 + 6 \eta_l^4 K^4 L^2 \left\| \nabla F\left(\tilde{\boldsymbol{x}}_t\right) \right\|_2^2 \tag{137}$$

where Equation (134) follows $(\alpha + \beta)^2 \leq 2\alpha^2 + 2\beta^2$; Equation (135) follows Cauchy–Schwarz inequality and Assumption 1; Equation (137) is based on Lemma 6. Then, according to the setting for the local learning rate in the description above, we can further simplify Equation (137):

$$\mathbb{E} \left\| \mathbb{E} \Delta \boldsymbol{x}_t^{(m)} - \eta_l K \nabla F\left(\tilde{\boldsymbol{x}}_t\right) \right\|_2^2 \leq 2 \eta_l^4 K^4 L^2 \cdot \mathbf{1}_{\{b<n\}} \frac{\sigma^2}{Mb} + 4 \eta_l^2 K^2 \left\| \mathbb{E} \tilde{g}_t - \nabla F\left(\tilde{\boldsymbol{x}}_t\right) \right\|_2^2$$
$$+ 6 \eta_l^4 K^4 L^2 \left\| \nabla F\left(\tilde{\boldsymbol{x}}_t\right) \right\|_2^2 \tag{138}$$

Plugging Equation (131) and Equation (138) back to Equation (126), we can primarily obtain the inequality below:

$$\mathbb{E} \left\| \Delta \boldsymbol{x}_t - \eta_l K \nabla F\left(\tilde{\boldsymbol{x}}_t\right) \right\|_2^2 \leq 2 \eta_l^2 K^2 \left( 1 - \left(p_t\right)^A \right) \left( 2 + \eta_l^2 K^2 L^2 \right) \mathbf{1}_{\{b<n\}} \frac{\sigma^2}{Mb}$$
$$+ 4 \eta_l^2 K^2 \left( 1 - \left(p_t\right)^A \right) \left( 1 + 3 \eta_l^2 \frac{L_\sigma^2}{b'} \right) \left\| \mathbb{E} \tilde{g}_t - \nabla F\left(\tilde{\boldsymbol{x}}_t\right) \right\|_2^2$$
$$+ 6 \eta_l^4 K^2 \left( 1 - \left(p_t\right)^A \right) \left( \frac{2 L_\sigma^2}{b'} + K^2 L^2 \right) \left\| \nabla F\left(\tilde{\boldsymbol{x}}_t\right) \right\|_2^2 \tag{139}$$
$$+ \left(p_t\right)^A \eta_l^2 K^2 \left\| \nabla F\left(\tilde{\boldsymbol{x}}_t\right) \right\|_2^2 \tag{140}$$

With the setting described in the Lemma, we have:

$$\mathbb{E} \left\| \Delta \boldsymbol{x}_t - \eta_l K \nabla F\left(\tilde{\boldsymbol{x}}_t\right) \right\|_2^2 \leq 6 \eta_l^2 K^2 \left( 1 - \left(p_t\right)^A \right) \mathbf{1}_{\{b<n\}} \frac{\sigma^2}{Mb}$$
$$+ 8 \eta_l^2 K^2 \left( 1 - \left(p_t\right)^A \right) \left\| \mathbb{E} \tilde{g}_t - \nabla F\left(\tilde{\boldsymbol{x}}_t\right) \right\|_2^2$$
$$+ 6 \eta_l^4 K^2 \left( 1 - \left(p_t\right)^A \right) \left( \frac{2 L_\sigma^2}{b'} + K^2 L^2 \right) \left\| \nabla F\left(\tilde{\boldsymbol{x}}_t\right) \right\|_2^2 \tag{141}$$

$$+ (p_t)^A \eta_l^2 K^2 \|\nabla F(\tilde{\boldsymbol{x}}_t)\|_2^2 \tag{142}$$

Therefore, according to the upper bound analyzed in the previous inequalities, Equation (123) can be reformulated as

$$\mathbb{E} F(\tilde{\boldsymbol{x}}_{t+1}) - F(\tilde{\boldsymbol{x}}_t) \tag{143}$$

$$\leq -\frac{\eta_s \eta_l K}{2} \left(1 - (p_t)^A\right) \left(1 - 6\eta_l^2 \left(\frac{2L_\sigma^2}{b'} + K^2 L^2\right)\right) \|\nabla F(\tilde{\boldsymbol{x}}_t)\|_2^2 - \left(\frac{1}{2\eta_s \eta_l K} - \frac{L}{2}\right) \|\tilde{\boldsymbol{x}}_{t+1} - \tilde{\boldsymbol{x}}_t\|_2^2$$

$$+ 4\eta_s \eta_l K \left(1 - (p_t)^A\right) \|\mathbb{E}\tilde{g}_t - \nabla F(\tilde{\boldsymbol{x}}_t)\|_2^2 + 3\eta_s \eta_l K \left(1 - (p_t)^A\right) \mathbf{1}_{\{b<n\}} \frac{\sigma^2}{Mb} \tag{144}$$

By means of the setting in the description above, we can obtain the desired conclusion. $\square$

**Theorem 6.** *Suppose that Assumption 1, 2 and 3 hold. Let the local updates $K \geq 1$, and the local learning rate $\eta_l$ and the global learning rate $\eta_s$ be $\eta_s \eta_l = \frac{1}{KL} \left(1 + \frac{2M}{Ap}\sqrt{1-p^A}\right)^{-1}$, where $\eta_l \leq \min\left(\frac{1}{2\sqrt{6}KL}, \frac{\sqrt{b'/K}}{4\sqrt{3}L_\sigma}\right)$. Therefore, the convergence rate of FedAMD for non-convex objectives should be*

$$\min_{t\in[T]} \|\nabla F(\tilde{\boldsymbol{x}}_t)\|_2^2 \leq O\left(\frac{1}{T}\left(\frac{1}{1-p^A} + \frac{M}{Ap\sqrt{1-p^A}}\right)\right) + O\left(\mathbf{1}_{\{b<n\}} \frac{\sigma^2}{Mb}\right) \tag{145}$$

*where we treat $F(\tilde{\boldsymbol{x}}_0) - F_*$ and $L$ as constants.*

*Proof.* With Lemma 5 and Lemma 7, we can find the following recursive function under the constant probability settings:

$$\mathbb{E} F(\tilde{\boldsymbol{x}}_{t+1}) + \frac{4\eta_s \eta_l K \left(1 - p^A\right) M}{Ap} \mathbb{E}\|\mathbb{E}\tilde{g}_{t+1} - \nabla F(\tilde{\boldsymbol{x}}_{t+1})\|_2^2 \tag{146}$$

$$\leq \mathbb{E} F(\tilde{\boldsymbol{x}}_t) + \frac{4\eta_s \eta_l K \left(1 - p^A\right) M}{Ap} \mathbb{E}\|\mathbb{E}\tilde{g}_t - \nabla F(\tilde{\boldsymbol{x}}_t)\|_2^2 - \frac{\eta_s \eta_l K}{4}\left(1 - p^A\right) \|\nabla F(\tilde{\boldsymbol{x}}_t)\|_2^2$$

$$- \left(\frac{1}{2\eta_s \eta_l K} - \frac{L}{2} - \frac{4\eta_s \eta_l K \left(1 - p^A\right) M}{Ap} \frac{M}{Ap} L^2\right) \mathbb{E}\|\tilde{\boldsymbol{x}}_{t+1} - \tilde{\boldsymbol{x}}_t\|_2^2$$

$$+ 3\eta_s \eta_l K \left(1 - p^A\right) \mathbf{1}_{\{b<n\}} \frac{\sigma^2}{Mb} \tag{147}$$

Since $\eta_s \eta_l = \frac{1}{KL}\left(1 + \frac{2M}{Ap}\sqrt{1-p^A}\right)^{-1}$, we have:

$$F_* \leq \mathbb{E} F(\tilde{\boldsymbol{x}}_T) \leq \mathbb{E} F(\tilde{\boldsymbol{x}}_T) + \frac{4\eta_s \eta_l K \left(1 - p^A\right) M}{Ap} \mathbb{E}\|\mathbb{E}\tilde{g}_T - \nabla F(\tilde{\boldsymbol{x}}_T)\|_2^2 \tag{148}$$

$$\leq \mathbb{E} F(\tilde{\boldsymbol{x}}_{T-1}) + \frac{4\eta_s \eta_l K \left(1 - p^A\right) M}{Ap} \mathbb{E}\|\mathbb{E}\tilde{g}_{T-1} - \nabla F(\tilde{\boldsymbol{x}}_{T-1})\|_2^2$$

$$- \frac{\eta_s \eta_l K}{4}\left(1 - p^A\right) \|\nabla F(\tilde{\boldsymbol{x}}_{T-1})\|_2^2 + 3\eta_s \eta_l K \left(1 - p^A\right) \mathbf{1}_{\{b<n\}} \frac{\sigma^2}{Mb} \tag{149}$$

$$\leq F(\tilde{\boldsymbol{x}}_0) + \frac{4\eta_s \eta_l K \left(1 - p^A\right) M}{Ap} \|\mathbb{E}\tilde{g}_0 - \nabla F(\tilde{\boldsymbol{x}}_0)\|_2^2$$

$$- \frac{\eta_s \eta_l K}{4}\left(1 - p^A\right) \sum_{t=0}^{T-1} \|\nabla F(\tilde{\boldsymbol{x}}_t)\|_2^2 + 3\eta_s \eta_l KT \left(1 - p^A\right) \mathbf{1}_{\{b<n\}} \frac{\sigma^2}{Mb} \tag{150}$$

According to Lemma 5, $\|\mathbb{E}\tilde{g}_0 - \nabla F(\tilde{\boldsymbol{x}}_0)\|_2^2 = 0$. Therefore, based on the derivation above, we can attain the following inequality:

$$\frac{1}{T} \sum_{t=0}^{T-1} \|\nabla F(\tilde{\boldsymbol{x}}_t)\|_2^2 \leq \frac{4 \left(F(\tilde{\boldsymbol{x}}_0) - F_*\right)}{\eta_s \eta_l KT \left(1 - p^A\right)} + 3\mathbf{1}_{\{b<n\}} \frac{\sigma^2}{Mb} \tag{151}$$

By using the settings of the local learning rate and the global learning rate in the description, we can obtain the desired result. □

### E.3 PROOFS FOR PL CONDITION

**Theorem 7.** *Suppose that Assumption 1, 2, 3 and 4 hold. Let the local updates $K \geq 1$, and the local learning rate $\eta_l$ and the global learning rate $\eta_s$ be $\eta_s \eta_l = \min\left(\frac{Ap}{MK\mu(1-p^A)}, \frac{1}{KL\left(1+\frac{16M}{\mu Ap}L\right)}\right)$, where $\eta_l \leq \min\left(\frac{1}{2\sqrt{6}KL}, \frac{\sqrt{b'/K}}{4\sqrt{3}L_\sigma}\right)$. Therefore, the convergence rate of FedAMD for PL condition should be*

$$\mathbb{E}F\left(\tilde{\boldsymbol{x}}_T\right) - F_* \leq \left(1 - \frac{1}{2}\mu K\left(1-p^A\right)\min\left(\frac{Ap}{MK\mu(1-p^A)}, \frac{1}{KL\left(1+\frac{16M}{\mu Ap}L\right)}\right)\right)^T \left(F\left(\tilde{\boldsymbol{x}}_0\right) - F_*\right)$$

$$+ O\left(\frac{1}{\mu} \cdot \mathbf{1}_{\{b<n\}}\frac{\sigma^2}{Mb}\right) \tag{152}$$

*Proof.* With Lemma 7, we have the recursive function on the time-varying probability settings under PL condition:

$$\mathbb{E}F\left(\tilde{\boldsymbol{x}}_{t+1}\right) - F\left(\tilde{\boldsymbol{x}}_t\right) \tag{153}$$

$$\leq -\frac{\eta_s\eta_l K}{4}\left(1-\left(p_t\right)^A\right)\|\nabla F\left(\tilde{\boldsymbol{x}}_t\right)\|_2^2 - \left(\frac{1}{2\eta_s\eta_l K} - \frac{L}{2}\right)\|\tilde{\boldsymbol{x}}_{t+1} - \tilde{\boldsymbol{x}}_t\|_2^2$$

$$+ 4\eta_s\eta_l K\left(1-\left(p_t\right)^A\right)\|\mathbb{E}\tilde{g}_t - \nabla F\left(\tilde{\boldsymbol{x}}_t\right)\|_2^2 + 3\eta_s\eta_l K\left(1-\left(p_t\right)^A\right)\mathbf{1}_{\{b<n\}}\frac{\sigma^2}{Mb} \tag{154}$$

$$\leq -\frac{\mu\eta_s\eta_l K}{2}\left(1-\left(p_t\right)^A\right)\left(F\left(\tilde{\boldsymbol{x}}_t\right) - F_*\right) - \left(\frac{1}{2\eta_s\eta_l K} - \frac{L}{2}\right)\|\tilde{\boldsymbol{x}}_{t+1} - \tilde{\boldsymbol{x}}_t\|_2^2$$

$$+ 4\eta_s\eta_l K\left(1-\left(p_t\right)^A\right)\|\mathbb{E}\tilde{g}_t - \nabla F\left(\tilde{\boldsymbol{x}}_t\right)\|_2^2 + 3\eta_s\eta_l K\left(1-\left(p_t\right)^A\right)\mathbf{1}_{\{b<n\}}\frac{\sigma^2}{Mb} \tag{155}$$

According to the description, we consider the probability $p_t = p$ and have:

$$\mathbb{E}F\left(\tilde{\boldsymbol{x}}_{t+1}\right) - F_* \tag{156}$$

$$\leq \left(1 - \frac{\mu\eta_s\eta_l K}{2}\left(1-p^A\right)\right)\left(F\left(\tilde{\boldsymbol{x}}_t\right) - F_*\right) - \left(\frac{1}{2\eta_s\eta_l K} - \frac{L}{2}\right)\|\tilde{\boldsymbol{x}}_{t+1} - \tilde{\boldsymbol{x}}_t\|_2^2$$

$$+ 4\eta_s\eta_l K\left(1-p^A\right)\mathbb{E}\|\mathbb{E}\tilde{g}_t - \nabla F\left(\tilde{\boldsymbol{x}}_t\right)\|_2^2 + 3\eta_s\eta_l K\left(1-p^A\right)\mathbf{1}_{\{b<n\}}\frac{\sigma^2}{Mb} \tag{157}$$

Since $\eta_s\eta_l \leq \frac{Ap}{MK\mu(1-p^A)}$, we have:

$$\mathbb{E}F\left(\tilde{\boldsymbol{x}}_{t+1}\right) - F_* + \frac{8}{\mu}\mathbb{E}\|\mathbb{E}\tilde{g}_{t+1} - \nabla F\left(\tilde{\boldsymbol{x}}_{t+1}\right)\|_2^2 \tag{158}$$

$$\leq \left(1 - \frac{\mu\eta_s\eta_l K}{2}\left(1-p^A\right)\right)\left(F\left(\tilde{\boldsymbol{x}}_t\right) - F_* + \frac{8}{\mu}\mathbb{E}\|\mathbb{E}\tilde{g}_t - \nabla F\left(\tilde{\boldsymbol{x}}_t\right)\|_2^2\right)$$

$$- \left(\frac{1}{2\eta_s\eta_l K} - \frac{L}{2} - \frac{8}{\mu}\frac{M}{Ap}L^2\right)\|\tilde{\boldsymbol{x}}_{t+1} - \tilde{\boldsymbol{x}}_t\|_2^2 + 3\eta_s\eta_l K\left(1-p^A\right)\mathbf{1}_{\{b<n\}}\frac{\sigma^2}{Mb} \tag{159}$$

According to the description $\eta_s\eta_l \leq \frac{1}{KL\left(1+\frac{16M}{\mu Ap}L\right)}$, we have:

$$\mathbb{E}F\left(\tilde{\boldsymbol{x}}_{t+1}\right) - F_* \leq \mathbb{E}F\left(\tilde{\boldsymbol{x}}_{t+1}\right) - F_* + \frac{8}{\mu}\mathbb{E}\|\mathbb{E}\tilde{g}_{t+1} - \nabla F\left(\tilde{\boldsymbol{x}}_{t+1}\right)\|_2^2 \tag{160}$$

$$\leq \left(1 - \frac{\mu\eta_s\eta_l K}{2}\left(1-p^A\right)\right)\left(F\left(\tilde{\boldsymbol{x}}_t\right) - F_* + \frac{8}{\mu}\mathbb{E}\|\mathbb{E}\tilde{g}_t - \nabla F\left(\tilde{\boldsymbol{x}}_t\right)\|_2^2\right)$$

$$+ 3\eta_s\eta_l K\left(1-p^A\right)\mathbf{1}_{\{b<n\}}\frac{\sigma^2}{Mb} \tag{161}$$

$$\leq \left(1 - \frac{\mu \eta_s \eta_l K}{2} \left(1 - p^A\right)\right)^{t+1} \left(F\left(\tilde{x}_0\right) - F_*\right)$$

$$+ \left(1 + \cdots + \left(1 - \frac{\mu \eta_s \eta_l K}{2} \left(1 - p^A\right)\right)^t\right) 3\eta_s \eta_l K \left(1 - p^A\right) \mathbf{1}_{\{b<n\}} \frac{\sigma^2}{Mb} \qquad (162)$$

$$= \left(1 - \frac{\mu \eta_s \eta_l K}{2} \left(1 - p^A\right)\right)^{t+1} \left(F\left(\tilde{x}_0\right) - F_*\right) + \frac{6}{\mu} \mathbf{1}_{\{b<n\}} \frac{\sigma^2}{Mb} \qquad (163)$$

By using the settings of the local learning rate and the global learning rate in the description, we can obtain the desired result. □

## F  ADDITIONAL EXPERIMENTS

In the main text, we have analyzed some experimental results in Section 5. In this part, we conduct more thorough experiments by setting different numbers of local updates and different secondary mini-batch sizes.

### F.1  DETAILED EXPERIMENTAL SETUP

**Training on Fashion MNIST.**  In Section 5, the experiment conducts on Fashion MNIST (Xiao et al., 2017), an image classification task to categorize a 28×28 greyscale image into 10 labels (including T-shirt/top, Trouser, Pullover, Dress, Coat, Sandal, Shirt, Sneaker, Bag, Ankle boot). In the training dataset, each class owns 6K samples. Then, we follow the setting of (Konečný et al., 2016; Li et al., 2019b) and partition the dataset into 100 clients ($M = 100$) such that each client holds two classes with a total of 600 samples. By this means, we simulate the heterogeneous data setting. To obtain a recognizable model on the images in the test dataset, we utilize a convolutional neural network structure LeNet-5 (LeCun et al., 1989; 2015). Below comprehensively presents the structure of LeNet-5 on Fashion MNIST:

Table 3: Details for LeNet-5 on Fashion-MNIST.

| Layer | Output Shape | Trainable Parameters | Activation | Hyperparameters |
|---|---|---|---|---|
| Input | (1, 28, 28) | 0 | | |
| Conv2d | (6, 24, 24) | 156 | ReLU | kernel size=5 |
| MaxPool2d | (6, 12, 12) | 0 | | kernel size=2 |
| Conv2d | (16, 8, 8) | 2416 | ReLU | kernel size=5 |
| MaxPool2d | (16, 4, 4) | 0 | | kernel size=2 |
| Flatten | 256 | 0 | | |
| Dense | 120 | 30840 | ReLU | |
| Dense | 84 | 10164 | ReLU | |
| Dense | 10 | 850 | softmax | |

**Training on EMNIST digits.**  In addition to Fashion MNIST, we utilize one more dataset EMNIST (Cohen et al., 2017) digits to further assess our approach efficiency. This task is to recognize 10 handwritten digits with a total of 240K training samples and 40K test samples. Similar to Fashion MNIST, we equally disjoint the dataset into 100 clients ($M = 100$), and each client possesses two classes. The model is trained with a 2-layer MLP (Yue et al., 2022), i.e.,

Table 4: Details for 2-layer MLP on EMNIST digits.

| Layer | Output Shape | Trainable Parameters | Activation | Hyperparameters |
|---|---|---|---|---|
| Input | (1, 28, 28) | 0 | | |
| Flatten | 784 | 0 | | |
| Dense | 100 | 78500 | ReLU | |
| Dense | 10 | 1010 | softmax | |

**Validation metrics.**  The training loss is calculated by the clients who perform local SGD on the average loss of all iterations. As for the test accuracy, the server utilizes the entire test dataset after the global model updates. The gradient complexity is the sum of all samples used for gradient calculation by all clients throughout the training. The communication overhead is measured by the transmission between the server and the clients.

**Miscellaneous.**  Our simulation experiment runs on Ubuntu 18.04 with Intel(R) Xeon(R) Gold 6254 CPU, NVIDIA RTX A6000 GPU, and CUDA 11.2. Our code is implemented using Python and PyTorch v.1.12.1. Clients are picked randomly and uniformly, without replacement in one round but with replacement in subsequent rounds. For each baseline, the local learning ($\eta_l$) rate picks the best one from the set $\{0.1, 0.03, 0.02, 0.01, 0.008, 0.005\}$, while the global learning rate ($\eta_s$) is selected from the set $\{1.0, 0.8, 0.1\}$. Without the annotation, we implicitly assume Fashion MNIST follows these settings: small minibatch size $b' = 64$, large minibatch size $b = full$, and the number of local updates $K = 10$. As for EMNIST, we suppose the anchor nodes utilize the entire dataset for the caching gradient, i.e., $b = full$, and the number of participants in each round is 20, i.e., $A = 20$.

Besides, to make BVR-L-SGD (Murata & Suzuki, 2021) compatible with partial participation in FL training, we only use sampled clients to compute the full gradients of local objectives instead of using all clients.

### F.2 MORE NUMERICAL RESULTS ON FASHION MNIST

In addition to the empirical results in Section 5, we evaluate the performance of FedAMD by using different large minibatch $b$ settings. Then, considering the number of local updates $K$, we assess the performance of the algorithm under various probability settings and compare it with other baselines.

#### F.2.1 COMPARISON AMONG VARIOUS HYPER-PARAMETER SETTINGS

**The setting of large mini-batch $b$.** Figure 2 – 4 depict the performance of FedAMD under constant probability $p = 0.9$, optimal constant probability, and optimal sequential probability, respectively, when the algorithm uses different $b$s. Overall, $b = full$ always outperform $b = 256$ and $b = 64$. Although there is no distinct difference between $b = 256$ and $b = 64$ in terms of final test accuracy and training loss, $b = 256$ is easier to attain a lower training loss during the training.

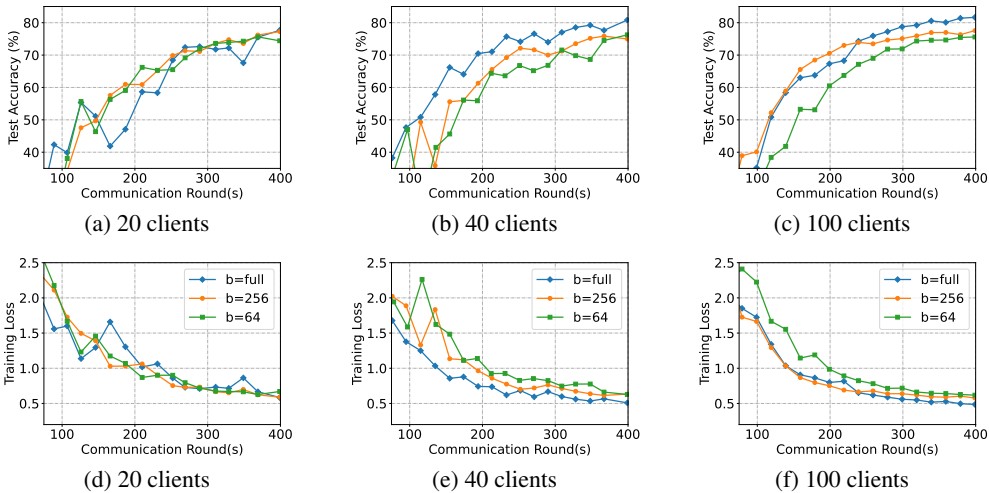

Figure 2: Comparison of test accuracy and training loss against the communication rounds for FedAMD with constant $p = 0.9$.

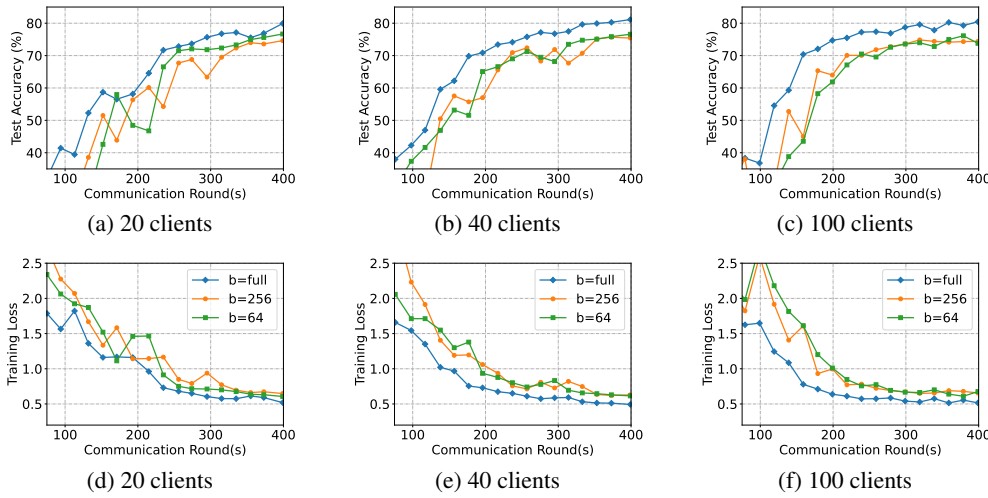

Figure 3: Comparison of test accuracy and training loss against the communication rounds for FedAMD with optimal constant probability $p$.

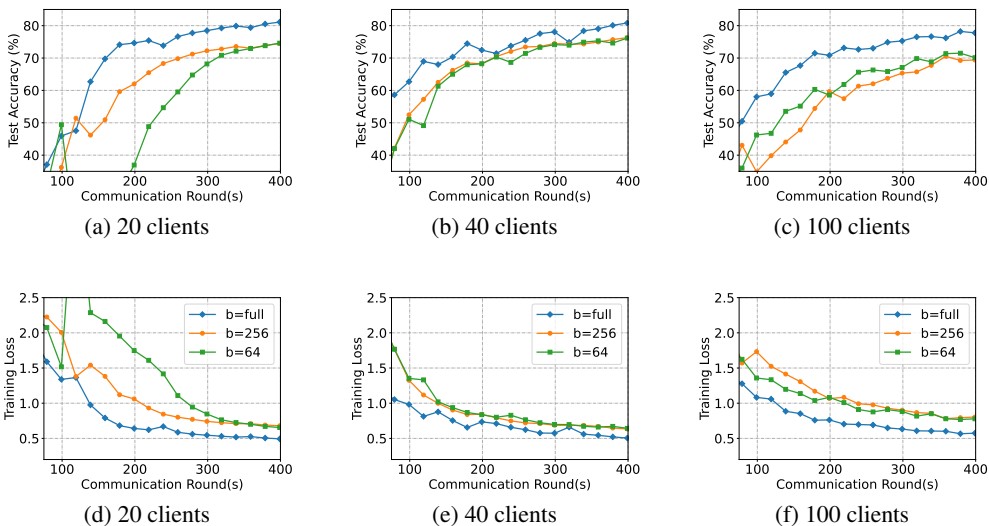

Figure 4: Comparison of test accuracy and training loss against the communication rounds for FedAMD with sequential $\{0, 1\}$.

**Number of local updates $K$.** Figure $5-7$ present the performance of FedAMD under the setting of $K = 10$, $K = 20$, and $K = 5$, respectively. In Section 5, we present the results under $K = 10$ (Figure 5), which manifests that: (I) The setting $\{0, 1\}$ is the most efficient performance under the sequential probability setting; (II) The setting near the optimal probability can attain the best result under the constant probability settings. In this part, we verify whether these two statements still hold in two more examples. As for $K = 20$ and $K = 5$, it can provide the best performance when the constant probability is set to be near the theoretical optimal one. However, statement (I) does not always hold in both settings. Specifically, when all clients participate in the training, $\{0, 0, 1\}$ even outperforms $\{0, 1\}$. A possible reason is that $\{0, 0, 1\}$ has more rounds to update the global model while the caching gradient does not significantly change compared to the situation running for one more round.

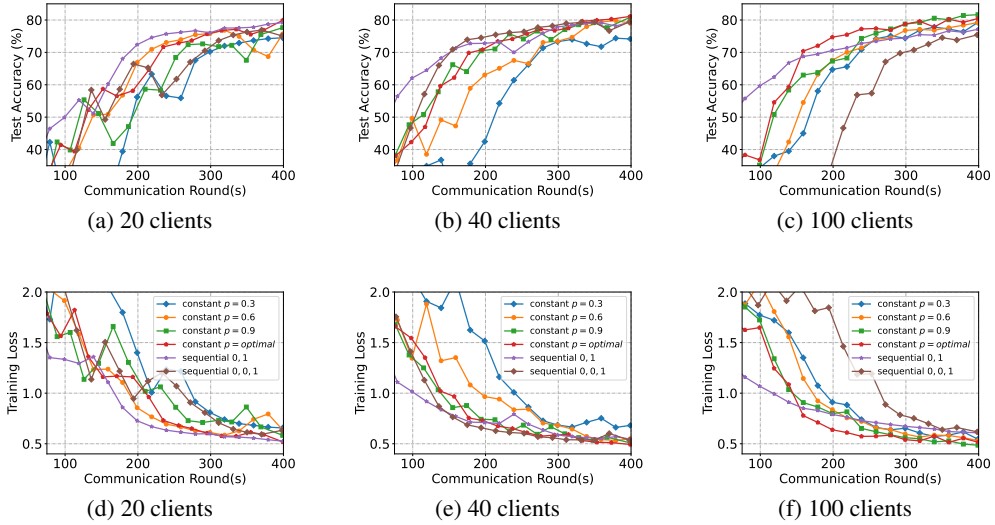

Figure 5: Comparison of different probability settings using training loss and test accuracy against the communication rounds for FedAMD by setting $K = 10$.

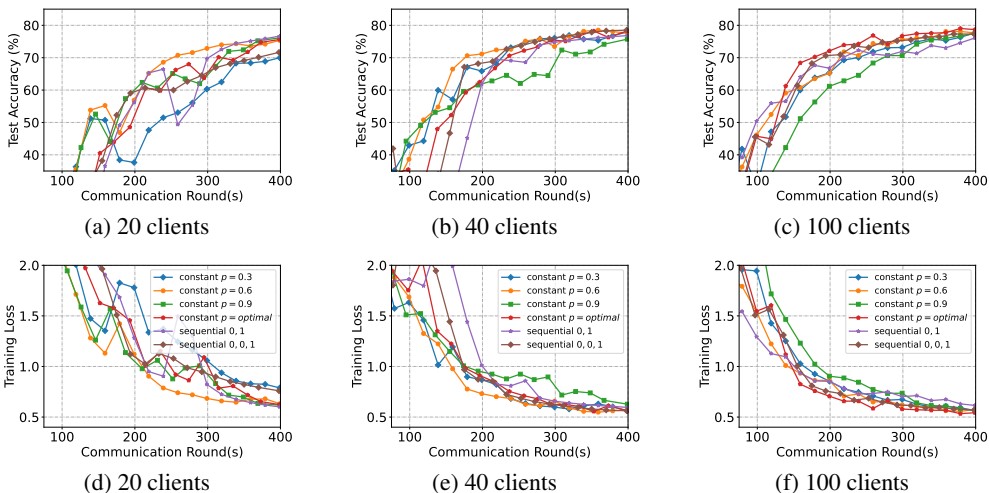

Figure 6: Comparison of different probability settings using training loss and test accuracy against the communication rounds for FedAMD by setting $K = 20$.

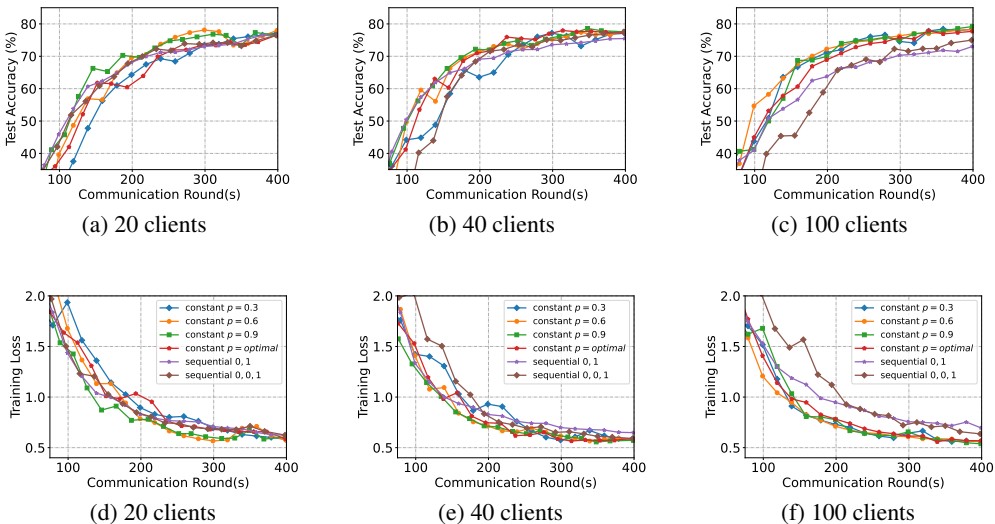

Figure 7: Comparison of different probability settings using training loss and test accuracy against the communication rounds for FedAMD by setting $K = 5$.

### F.2.2 COMPARISON AMONG VARIOUS BASELINES

In Section 5, we present the comparison in a tabular format. Then, in this part, we visualize the training progress as well as introduce more results under different $K$s with the help of Figure 8. In specific, Figure 8a – 8f are summarized into Table 2, while the rest explore the efficiency of FedAMD under more scenarios. As described in Table 2 and the first six figures, when $K = 10$, the conclusions we can draw include: (I) the final test accuracy of FedAMD exceeds that of the baselines; (II) FedAMD is able to attain an accurate model with less communication and computation consumption. Next, we evaluate the performance of FedAMD when $K = 20$ and $K = 5$.

- $K = 20$ **(Figure 8g – 8l):** In this case, the gradient computation of a miner is around twice as that of an anchor. Therefore, FedAMD may consume less computation overhead than FedAvg and SCAFFOLD. In terms of final test accuracy, these baselines achieve similar results in all cases, while FedAMD achieves the performance with less computation overhead.

- $K = 5$ **(Figure 8m – 8r):** FedAMD eventually achieves the best accuracy compared to the existing works. Additionally, we can obtain a well-performed model with less computational consumption. These two phenomena are in support of the statements mentioned above.

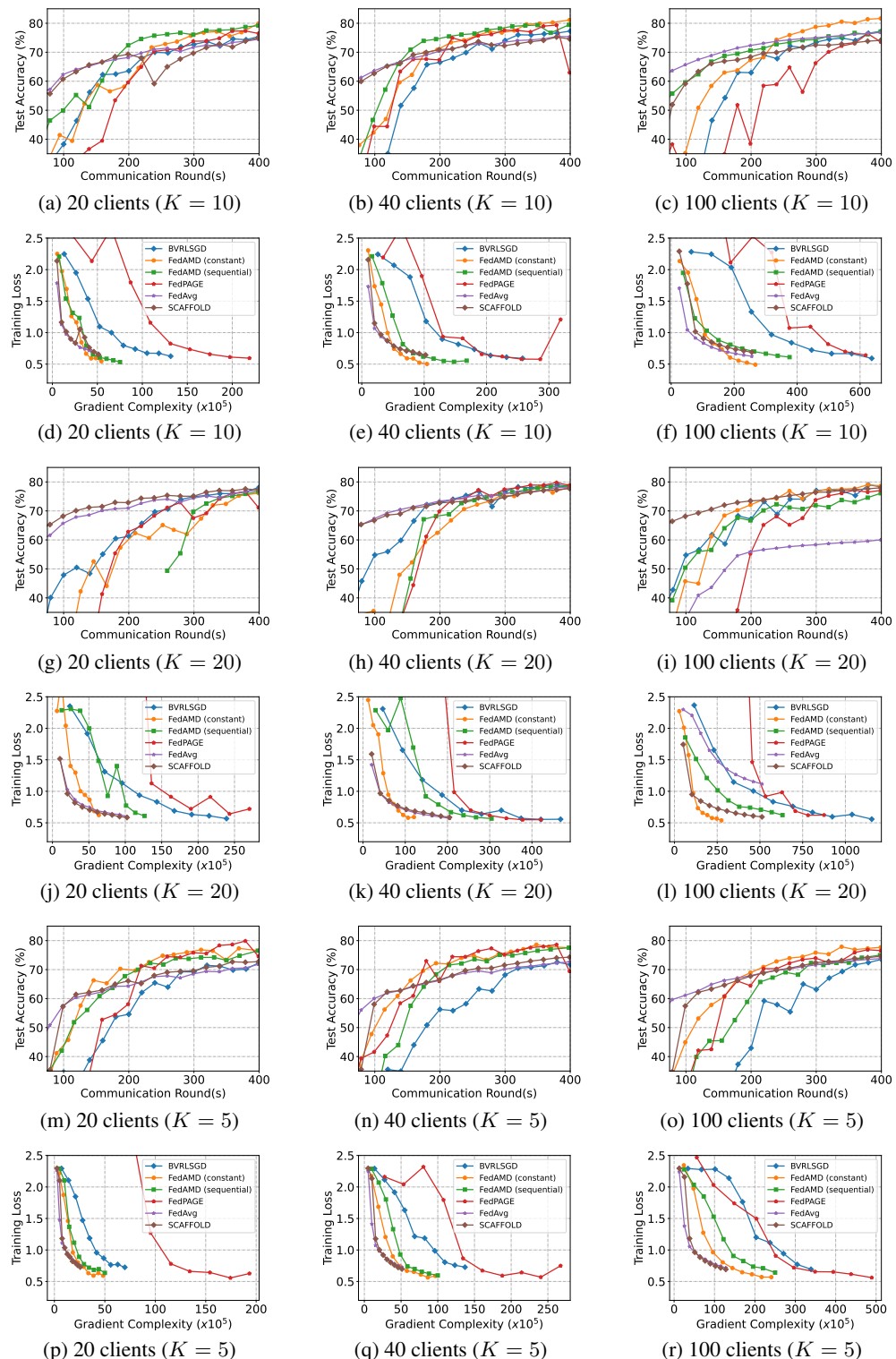

Figure 8: Comparison of different algorithms using test accuracy against the communication rounds and training loss against gradient complexity.

## F.3 Numerical Results on EMNIST digits

In this section, Figure 9 analyzes our algorithm with one more dataset, i.e., EMNIST. In the first three figures, we evaluate different probability settings. As for the rest of the figures, we compare FedAMD with other baselines.

With regards to different probability settings (Figure 9a – 9c), we are still able to draw two conclusions as stated in Appendix F.2.1. As for the comparison among different algorithms, our proposed algorithm is able to outperform the state-of-the-art works when we take the test accuracy and the computation overhead into joint consideration.

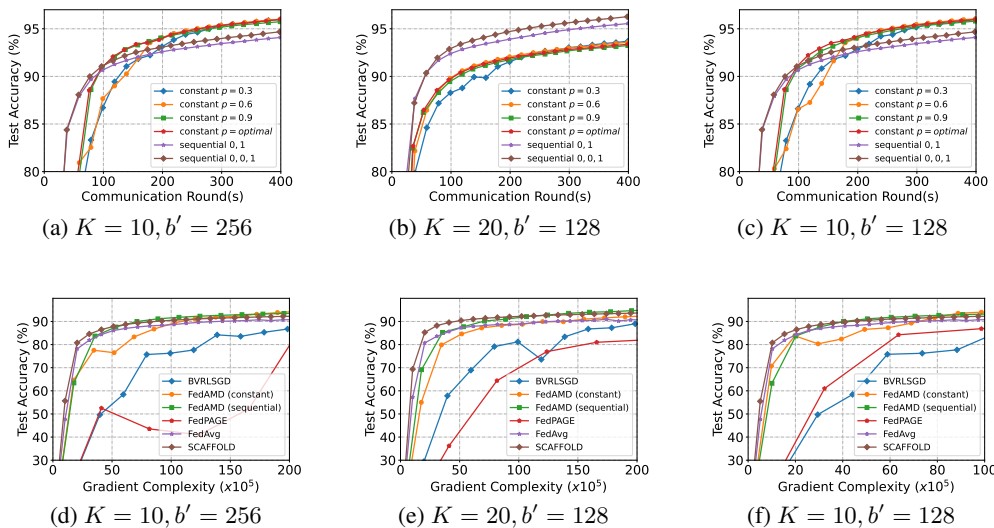

Figure 9: Comparison of different baselines and probability settings using test accuracy against communication rounds and gradient complexity, respectively.

