# OpenReview forum: "Anchor Sampling for Federated Learning with Partial Client Participation"
_ICLR.cc/2023/Conference — Submitted to ICLR 2023_

### Official Review · Reviewer_UWdf · 2022-10-24

**Confidence:** 2
**Correctness:** 2
**Technical Novelty And Significance:** 4
**Empirical Novelty And Significance:** 4
**Recommendation:** 6

**Clarity, Quality, Novelty And Reproducibility:**

Quality: The proposed FedAMD is superior to state-of-the-art works, and there are many theories to support the experimental results in this paper.
Clarity: The algorithm 1 is detailed and reasonable, but some lines(line 1 and line 19)need additional analysis and discussion.
Originality: This work may be the first work to analyze the effectiveness of large batches under partial client participation.


**Strength And Weaknesses:**

Strength:
1.The proposed FedAMD achieves sublinear speedup under non-convex objectives and linear speedup under the PL condition.
2.The proposed FedAMD is superior to SOTA works compared with BVR-L-SGD, FedAvg, FedPAGE, SCAFFOLD on Fashion MNIST.
3.The proofs of theories are detailed and correct, and they are consistent with the experimental results.

Weaknesses:
1.It seems that there is no analysis about the influence of initial model (line 1) on the experimental results in this paper. In fact, the influence of initial model exists (assuming that the best model is set at the beginning). It is suggested to add analysis about the influence of initial model.
2.The experiment is too simple. "We train a convolutional neural network LeNet-5 using Fashion MNIST". Such a specific experiment may affect the degree of confidence of the experiment results.
3.This paper proposes two strategies(constant and sequential) for selecting anchor and miner groups. But it would be better if there were a strategy based on the performance of each client.
4.The analysis of g(i)t,k (line 19)is reasonable, but it is worth discussing whether a factor α is needed for second and third items.
5.For the analysis of Figure 1, it seems that only Communication Rounds=400 is considered(why is 400?). Such analysis seems to be one-sided, and there are other considerations in fact, such as the change rate of the training loss.

**Summary Of The Paper:**

This paper proposes a framework FedAMD, that disjoints the partial participants into anchor and miner groups. Clients in anchor group target to discover the bullseyes based on their local data distribution, while clients in the miner group perform multiple local updates and finally drive the update of the global model guided by the global bullseye. Under the partial-client scenario, this paper gives many theoretical proofs that FedAMD achieves sublinear speedup under non-convex objectives and linear speedup under the PL condition. Experimental results demonstrate that FedAMD is superior to SOTA works compared with BVR-L-SGD, FedAvg, FedPAGE, SCAFFOLD on Fashion MNIST.

**Summary Of The Review:**

In general, it is a good paper. It has the SOTA results, many theoretical proofs and analysis. However, there are some weaknesses in the setting and analysis of the experiments section.

---

> ### Author Response · Authors · 2022-11-13
> **Response to Reviewer UWdf**
>
> Thanks a lot for providing us with valuable and constructive feedback. In view of your comments, we would like to address your concerns below.
> ***
> > **[P1]** It seems that there is no analysis about the influence of initial model (line 1) on the experimental results in this paper. In fact, the influence of initial model exists (assuming that the best model is set at the beginning). It is suggested to add analysis about the influence of initial model.
>
> Thanks for your question regarding the model initialization. We understand that the pretrained models can transfer some learned latent knowledge to a new task so as to improve the training efficiency. However, in federated optimization, such a pretrained model is likely unavailable. Therefore, we commonly assume the given model with **arbitrary parameters** and obtain a generalized bound [1, 2, 3] (p.s., this analysis covers the best model settings because they are the special cases of arbitrary models). Based on the random model settings, we can learn the performance of an algorithm in the worst case.
>
> **Update:** Equation (100), (151), and (163) in the appendix contain the initial function value gap to the optimal solution (i.e., $F(\tilde{x}\_0)-F\_*$). These three equations separately refer to three cases discussed in the main paper and reflect how the initial model affects the convergence rate. It is worth noting that this term is treated as a constant when we find a big-O expression of the convergence rate. This practice has been adopted in some well-known works [3, 4].
>
> ### **References:**
>
> [1] Li et al., On the Convergence of FedAvg on Non-IID Data, ICLR 2020
>
> [2] Karimireddy et al., SCAFFOLD: Stochastic Controlled Averaging for Federated Learning, ICML 2020
>
> [3] Yang et al., Achieving Linear Speedup with Partial Worker Participation in Non-IID Federated Learning, ICLR 2021
>
> [4] Guo et al., Hybrid Local SGD for Federated Learning with Heterogeneous Communications, ICLR 2022
> ***
> > **[P2]** This paper proposes two strategies (constant and sequential) for selecting anchor and miner groups. But it would be better if there were a strategy based on the performance of each client.
>
> Thanks for your insightful suggestion about the design of anchor sampling. In this paper, our main contribution is introducing anchor sampling and showing that this approach can generate a better convergence rate than the state-of-the-art works. We agree that a better anchor sampling design (e.g., client-specific probability setting) may exist to facilitate the training performance, but we believe it may not have significant improvement from the theoretical perspective, i.e., $O(1/\epsilon)$ and $O(\log(1/\epsilon))$ under non-convex and PL conditions, respectively.
>
> ***
>
> > **[P3]** The analysis of g(i)t,k (line 19) is reasonable, but it is worth discussing whether a factor $\alpha$ is needed for second and third items.
>
> Thanks for your suggestions on the fine-tuning of each local update. To the best of our knowledge, there are two existing works [1, 2] investigating the suggestions you mentioned. Xin et al. [1] show that the value of $\alpha$ approach one as the training rounds $T \rightarrow \infty$; a similar conclusion is also derived by [2] which utilizes $\alpha=1$ to develop the corollaries.
>
> ### **References:**
>
> [1] Xin et al., A Hybrid Variance-Reduced Method for Decentralized Stochastic Non-Convex Optimization, ICML 2021
>
> [2] Wu et al., From Deterioration to Acceleration: A Calibration Approach to Rehabilitating Step Asynchronism in Federated Optimization, 2021
>
> ***
>
> > **[P4]** For the analysis of Figure 1, it seems that only Communication Rounds=400 is considered(why is 400?). Such analysis seems to be one-sided, and there are other considerations in fact, such as the change rate of the training loss.
>
> Thanks for your question concerning our experiments. Firstly, as shown in Figure 1, various settings have reached a stable stage after 400-round training, although subtle fluctuation may occur. The process of 400-round training can clearly present the performance of different probability settings, including the training efficiency and the test accuracy.
>
> It is interesting to plot the change rate of the training loss against the communication rounds, but it is not a common practice. Most federated optimization works [1, 2, 3] solely focus on the training efficiency, e.g., the training loss and the test accuracy against the number of communication rounds. Besides, we provide the training loss against the gradient complexity in the supplementary materials.
>
> ### **References:**
>
> [1] Yang et al., Achieving Linear Speedup with Partial Worker Participation in Non-IID Federated Learning, ICLR 2021
>
> [2] Gu et al., Fast Federated Learning in the Presence of Arbitrary Device Unavailability, NeurIPS 2021
>
> [3] Guo et al., Hybrid Local SGD for Federated Learning with Heterogeneous Communications, ICLR 2022
>
> [4] Wang et al., Communication-Efficient Adaptive Federated Learning, ICML 2022

---

> > ### Author Response · Authors · 2022-11-20
> > **Second Response to Reviewer UWdf**
> >
> > First and foremost, thanks for your suggestion that we should conduct more experiments other than Fashion MNIST. In this response, we include a new empirical study on EMNIST digits. In this experiment, we train an MLP model (non-convex) and fix the number of clients and participants for 100 and 20, respectively. To simulate the non-i.i.d. features, we disjoint the training dataset such that each client owns two classes.
> >
> > The table below reports gradient complexity ($\times 10^5$ samples), communication costs ($\times 32$ Mbits), and rounds when the model achieves 88\% test accuracy for the first time, and the test accuracy (%) after 400 rounds. For more details, please refer to Appendix F.3.
> >
> > - $K=10$, $b'=128$:
> >
> >     |Method|Gradient Complexity |Communication Costs|Round|Accuracy|
> >     |------|:-----------------:|:-----------------:|:---:|:----------------:|
> >     |BVR-L-SGD|160.4|481.7|108|90.4
> >     |FedAvg|33.3|413.5|129|91.1
> >     |FedPAGE|104.5|512.0|63|93.5
> >     |SCAFFOLD|24.1|597.9|93|92.3
> >     |FedAMD (constant $p=0.3$)|45.7|388.9|90|96.0
> >     |FedAMD (constant $p=0.6$)|38.9|301.2|78|96.0
> >     |FedAMD (constant $p=0.9$)|30.0|207.8|54|95.9
> >     |FedAMD (constant $p=optimal$)|30.0|210.3|58|96.0
> >     |FedAMD (sequential $\{0, 1\}$)|25.8|206.7|25|94.1
> >     |FedAMD (sequential $\{0, 0, 1\}$)|25.1|211.5|32|94.7
> >
> > - $K=10$, $b'=256$:
> >
> >     |Method|Gradient Complexity |Communication Costs|Round|Accuracy|
> >     |------|:-----------------:|:-----------------:|:---:|:----------------:|
> >     |BVR-L-SGD|216.3|481.7|108|90.4
> >     |FedAvg|66.6|413.4|129|91.1
> >     |FedPAGE|223.5|896.9|107|93.9
> >     |SCAFFOLD|48.2|597.9|93|92.3
> >     |FedAMD (constant $p=0.3$)|77.7|388.9|90|96.1
> >     |FedAMD (constant $p=0.6$)|55.7|304.9|79|96.1
> >     |FedAMD (constant $p=0.9$)|33.4|207.8|54|95.8
> >     |FedAMD (constant $p=optimal$)|35.7|220.0|59|96.1
> >     |FedAMD (sequential $\{0, 1\}$)|39.1|206.7|25|94.1
> >     |FedAMD (sequential $\{0, 0, 1\}$)|42.0|211.5|32|94.7
> >
> > - $K=20$, $b'=128$:
> >
> >     |Method|Gradient Complexity |Communication Costs|Round|Accuracy|
> >     |------|:-----------------:|:-----------------:|:---:|:----------------:|
> >     |BVR-L-SGD|158.7|353.0| 79| 92.2
> >     |FedAvg|48.2|299.0|93|91.5
> >     |FedPAGE|661.6|2646.1|319|88.8
> >     |SCAFFOLD|24.6|305.3|47|93.8
> >     |FedAMD (constant $p=0.3$)|66.9|334.2|77|93.9
> >     |FedAMD (constant $p=0.6$)|46.2|252.4|65|93.8
> >     |FedAMD (constant $p=0.9$)|35.4|220.8|56|93.3
> >     |FedAMD (constant $p=optimal$)|33.2|203.5|57|93.7
> >     |FedAMD (sequential $\{0, 1\}$)|24.1|127.2|15|95.6
> >     |FedAMD (sequential $\{0, 0, 1\}$)|29.3|147.9|22|96.3

---

> ### Author Response · Authors · 2022-11-26
> **Thank you and look forward to your post-rebuttal updates**
>
> Dear Reviewer UWdf,
>
> First and foremost, we are grateful for your commitment to reviewing our paper.
>
> As you know, we have submitted two responses regarding your concerns in the first-phase review, but we haven't heard from you yet. We understand that you may not be available to read our response thoroughly, so we summarize each point of our response in one or two sentences:
>
> - **[P1]** Equation (100), (151), and (163) in the appendix contain the initial function value gap to the optimal solution (i.e., $F(\tilde{x}\_0) - F\_*$), which reflects how the initial model affects the convergence rate.
> - **[P2]** A sophisticated design of anchor sampling may facilitate the training performance, but it is unlikely to improve the order of the convergence rate, i.e., $O(1/T)$ under non-convex objectives and linear convergence under PL conditions.
> - **[P3]** Recent works have presented that the optimal $\alpha$ is 1.
> - **[P4]** In view of the trade-off between training progress and its cost, we stop the training at the 400th round because it can clearly show the differences among different baselines and hyper-parameter settings. Besides, it is common to plot a figure with a vertical axis showing training loss or test accuracy.
> - **[Second Response]** We conduct a new empirical study on EMNIST digits, and the results show that FedAMD can outperform other baselines when the anchor sampling is properly designed.
>
> This letter does not aim to force you to raise your rating and confidence; we want to know anything else we can do to improve our paper. Feel free to let us know if you have any comments. We are **eager** to receive your post-rebuttal updates.
>
> Best,
>
> Authors of Paper 3563 (**Anchor Sampling for Federated Learning with Partial Client Participation**)

---

> ### Author Response · Authors · 2022-11-28
> **A gentle reminder**
>
> Dear Reviewer UWdf,
>
> As the second-phase discussion will end in **two weeks**, we are yet to receive any updates from you. Could you please kindly take a look at our response? We are happy to discuss our work with you through OpenReview platform. Thanks a lot.
>
> Best,
>
> Authors of Paper 3563 (**Anchor Sampling for Federated Learning with Partial Client Participation**)

---

> ### Author Response · Authors · 2022-12-01
> **Your voice matters**
>
> Dear Reviewer UWdf,
>
> Thanks a lot for your insightful comments in the first-round review.
>
> We have posted our response for over two weeks, but we are still waiting to hear you. We would like to know if you have read our response. Does it fully address your concerns? Does our work deserve acceptance? Feel free to write something through the OpenReview platform. We will highly appreciate your prompt feedback.
>
> Best,
>
> Paper 3563 Authors

---

> ### Author Response · Authors · 2022-12-02
> **We hope you can read our rebuttal and give a response!**
>
> Dear Reviewer UWdf,
>
> It is the last ten days that we can discuss with each other through the OpenReview platform. Could you please kindly take a look at our response? It would be highly appreciated if you could provide follow-up comments. Thanks a lot.
>
> Lastly, we sincerely wish you a nice weekend.
>
> Best,
>
> Paper 3563 Authors

---

> ### Author Response · Authors · 2022-12-05
> **Look forward to your reponse as the discussion period ends in one week**
>
> Dear Reviewer UWdf,
>
> As the discussion will end in one week, we treasure the final chance to discuss our work with you. Could you feel free to go through our rebuttal and drop some notes? Thanks a lot.
>
> Best,
>
> Paper 3563 Authors

---

> ### Author Response · Authors · 2022-12-06
> **Do you have more comments?**
>
> Thanks again for your efforts and insightful suggestions in reviewing. As the discussion ends shortly, do you mind sparing some time to read our response and leave your comments (or acknowledge you have read it)? It won't take you too much time.
>
> Best,
> Authors

---

> ### Author Response · Authors · 2022-12-09
> **Your response is always welcome and highly appreciated**
>
> Dear Reviewer UWdf,
>
> We hope this kind reminder finds you well.
>
> Our response has been posted for over three weeks. Unfortunately, we are yet to receive your follow-up feedback. We understand everyone is very busy when the year comes to an end. To make your time more efficient, we are happy to summarize our response based on your first-phase comments: (*W refers to the weaknesses your comments listed at https://openreview.net/forum?id=VLnODGVVAsL&noteId=YKgmUEja4HP; P indicates the points of our response at [1: P1--P4] https://openreview.net/forum?id=VLnODGVVAsL&noteId=LdbnU2OaN6, [2: Second Response] https://openreview.net/forum?id=VLnODGVVAsL&noteId=f2nHSl6lJrn.*)
>
> - **[W1, see P1 for details]** Equation (100), (151), and (163) in the appendix contain the initial function value gap to the optimal solution (i.e., $F(\tilde{x}\_0) - F\_*$), which reflects how the initial model affects the convergence rate.
> - **[W2, see Second Response for details]** We conduct a new empirical study on EMNIST digits, and the results show that FedAMD can outperform other baselines when the anchor sampling is properly designed.
> - **[W3, see P2 for details]** A sophisticated design of anchor sampling may facilitate the training performance, but it is unlikely to improve the order of the convergence rate, i.e., $O(1/T)$ under non-convex objectives and linear convergence under PL conditions.
> - **[W4, see P3 for details]** Recent works have presented that the optimal $\alpha$ is 1.
> - **[W5, see P4 for details]** In view of the trade-off between training progress and its cost, we stop the training at the 400th round because it can clearly show the differences among different baselines and hyper-parameter settings. Besides, it is common to plot a figure with a vertical axis showing training loss or test accuracy.
>
> Should you have any concerns, please don't hesitate to let us know. If we cannot hear you in the next few days, we believe that our response has fully addressed your concerns. In other words, our work has properly settled all weaknesses you mentioned, and you CANNOT see any deficiencies in our work up to now.
>
> Thanks a lot for your time and efforts.

---

> ### Author Response · Authors · 2022-12-12
> **Final call for your feedback**
>
> Dear Reviewer UWdf,
>
> Thanks a lot for your engagement in reviewing our work.
>
> We hope our response has fully addressed your concerns. As your future comments on our work may not be visible to the authors and the public, we sincerely hope you can positively recommend our work to AC. If you have further comments/suggestions, don't hesitate to let us know. It seems that we still have some time to discuss the work.
>
> Best,
>
> Authors of Paper 3563 (**Anchor Sampling for Federated Learning with Partial Client Participation**)

---

### Official Review · Reviewer_Moju · 2022-10-24

**Confidence:** 4
**Correctness:** 3
**Technical Novelty And Significance:** 2
**Empirical Novelty And Significance:** 2
**Recommendation:** 5

**Clarity, Quality, Novelty And Reproducibility:**

The paper is well written, easy to understand in terms of clarity and is of quality. The paper modifies well known techniques used in the context of FL and carefully tunes them so as to attain better rates. Hence, the originality is moderate.

**Strength And Weaknesses:**

Strengths
+ The paper is very well written and the algorithm is well motivated.
+ The convergence results show an $O\left(\frac{1}{\epsilon}\right)$ for general non-convex losses and linear convergence with the extra PL assumptions which is an improvement over the state of the art.
+ The proposed algorithm FedAMP also shows the efficacy of schemes which resort to heterogeneous batch sizes and updating schemes while resorting to variance reduction.
+ The experimental results are comprehensive in terms of comparison with other baselines and tends to outperform other baselines.


Weaknesses

- Though the algorithm claims to use partial client participation; it requires a full client participation at the first step which in turn requires each gradient from each client to be individually available to the server and that too persisted over time instead of being available in an aggregated manner. This severely undermines the privacy-preserving nature of FL as it makes each client susceptible to gradient inversion attacks. It is also not clear how can this be alleviated without resorting to additional storage in each device.
- The convergence results while improving in terms of $\epsilon$-accuracy of the error, is substantially worse as compared to other baselines in terms of dependence on the total number of clients. While other algorithms scale inversely with respect to the total number of clients, FedAMP scales linearly with the number of clients in the setup. Unless, the number of selected clients is always a function of the total number of clients, it seems the performance of FedAMP gets worse with increasing number of clients, if the number of clients selected per round is constant. It is also worth noting the performance benefit comes at the expense of additional assumption 3.
- An inherent assumption made by the authors is that the number of samples across clients is the same. The method by which a client is assigned to being an anchor or minor is solely decided by a coin toss and not based on the number of samples. In the general case, some clients may have fewer samples, in the case of which the number of local epochs and the batch size can't be uniform. It is also not clear how the choice $b$ and $b^{'}$ affects the algorithm performance.
- The authors miss comparing and contrasting with FedNova by Wang et.al., which especially takes care of the case where different clients undergo different number of rounds along with the flexibility of running different local solvers. It would be good to add comparisons with FedDyn and FedOPT which are highly performant baselines.
- While a theoretically optimal $p$ is derived, it can't be derived in practice as quantities such as the number of samples on each device and $\sigma^{2}$ is unknown to the server.
- For the variance reduction phase, it requires additional storage of the past gradient which is the same size as that of the model. This can be particularly prohibitive with larger model and resource constrained devices.

**Summary Of The Paper:**

This paper studies the problem of improving convergence in Federated Learning especially under partial client participation settings. Differently from previous work, which either used large batch sizes or variance reduction techniques while resorting to full client participation, the authors propose a randomization of FedAvg and FedSGD. In that, the clients switch from being either anchor or minor groups. Convergence results are derived for non-convex losses under general conditions. Experimental results illustrate the efficacy of the approach.

**Summary Of The Review:**

The paper has weaknesses in terms of the convergence proofs with the dependence on the total number of clients being significantly worse as compared to other schemes.  The applicability, usability and originality of the paper is undermined by some assumptions which either don't hold or are difficult to verify in practice. Moreover, the randomization of a client either doing a local-SGD style update or no update, can lead to stretches with very few clients contributing to the update. Having said that, I would be more than happy to increase my score if my concerns are addressed.

---

> ### Author Response · Authors · 2022-11-13
> **Response to Reviewer Moju [1/4]**
>
> We are grateful for your careful review and constructive comments. We are glad to hear your appreciation of our presentation quality and the theoretical and empirical results. However, some misunderstandings of our work lead to your poor rating. We sincerely hope the following clarification can help you better understand our work as well as address your concerns.
>
> ***
>
> > **[P1]** Though the algorithm claims to use partial client participation; it requires a full client participation at the first step which in turn requires each gradient from each client to be individually available to the server and that too persisted over time instead of being available in an aggregated manner. This severely undermines the privacy-preserving nature of FL as it makes each client susceptible to gradient inversion attacks. It is also not clear how can this be alleviated without resorting to additional storage in each device.
>
> Thanks for your questions regarding the initialization of FedAMD. The initialization step (Line 1 - 6 in Algorithm 1) is **optional** in our algorithm, meaning that it does not conflict with partial client participation. Many well-known studies [1, 2, 3] implicitly or explicitly involve an initialization step with full-client participation. For example, SCAFFOLD [1] and FedDyn [2] require an accurate calculation on $\nabla F_i(w_0)$ for all workers $i$ at the initial model $w_0$, but these gradients act as an input. Motivated by these works, the initialization step is factually **regarded as an input** rather than part of the model training.
>
> Gradient inversion attack is not what we focus on in this work, although it is a significant problem in federated learning. To the best of our knowledge, when the batch size is sufficiently large, and the private labels are well protected, the existing works find it hard to recover the raw data of a batch [4]. In the initialization and the training of the anchor nodes, our algorithm utilizes a batch with the size of $b$, which is quite large (or full size) in practice. Therefore, it is **less likely** that the gradient inversion attack succeeds and weakens FedAMD's feasibility. As FedAMD is compatible with the defense approaches like perturbing gradients [4, 5], we can make use of these methods such that FedAMD can be robust against the gradient inversion attack.
>
> ### **References:**
>
> [1] Karimireddy et al., SCAFFOLD: Stochastic Controlled Averaging for Federated Learning, ICML 2020
>
> [2] Acar et al., Federated Learning Based on Dynamic Regularization, ICLR 2020
>
> [3] Zhao et al., Faster rates for compressed federated learning with client-variance reduction, 2022
>
> [4] Huang et al., Evaluating gradient inversion attacks and defenses in federated learning, NeurIPS 2021
>
> [5] Zhu et al., Deep leakage from gradients, NeurIPS 2019

---

> > ### Author Response · Authors · 2022-11-13
> > **Response to Reviewer Moju [2/4]**
> >
> > > **[P2]** An inherent assumption made by the authors is that the number of samples across clients is the same. The method by which a client is assigned to being an anchor or minor is solely decided by a coin toss and not based on the number of samples. In the general case, some clients may have fewer samples, in the case of which the number of local epochs and the batch size can't be uniform. It is also not clear how the choice $b$ and $b'$ affects the algorithm performance.
> >
> > Thanks for your question regarding the problem formulation and the design of anchor sampling. In our work, it is not mandatory that clients own the same size of training set or carry on the same weights. Our theoretical analysis exemplifies how we can analyze the convergence of FedAMD in a canonical federated learning setting, after which we can analyze more complicated cases with mild changes. Some previous works [1, 2, 3, 4, 5] adopt such a simplified way to analyze the convergence property as well. In specific, we would like to highlight the changes case-by-case:
> >
> > - **Clients hold the training sets with various sizes (i.e., $|\mathcal{D}\_m| = n\_m$ for all $m \in [M]$), but they carry on the equal weights (i.e., $F(x) = \frac{1}{M} \sum\_{m \in [M]} F\_m(x)$).** We have a **minor** change for the minibatch size $b$, i.e., for any client $m \in [M]$, the minibatch size $b$ is $\min(\cdot, n_m)$ in all theorems and corollaries (p.s., original writing is $\min(\cdot, n)$).
> >
> > - **Clients carry on the different weights (i.e., $F(x) = \sum\_{m \in [M]} q\_m F\_m(x)$ and $\sum_{m \in [M]} q_m = 1$).** We have a **minor** change for $avg(\cdot)$:
> >     - In Line 9 of Algorithm 1, $\tilde{g}\_t = avg(v\_t) = \sum\_{m \in [M]} q\_m v\_t^{(m)}$
> >     - Let $\Delta x_t = \{\Delta x_t^{(a_1)}, \dots, \Delta x_t^{(a_n)}\}$ in Line 26 where $0 \leq n \leq A$, $avg(\Delta x_t) = \frac{1}{n} \sum_{i} q_{a_i} M \Delta x_t^{(a_i)}$.
> >
> > Even if we make the changes as above, it is unnecessary to devise the anchor sampling probability to be client-specific. In this paper, we obtain the sequence {$p_t$}$_{t\geq0}$, which achieves a better convergence rate than the state-of-the-art works, although the sequence is pattern-wise (a.k.a. sequential) or constant.
> >
> > We have mentioned in the pseudocode that $b' < b$. Generally, the minibatch size in the miner group is the same as that used in the vanilla SGD algorithm, i.e., $b' = O(1)$. Besides, the clients in the anchor group make use of a large batch with the size of $b$. As we expect $\epsilon$ approaching 0, the setting for $b$ gets larger because it is at an order of $O(1/\epsilon)$ until it reaches the maximum capacity of a client's training set. As for how the setting of $b$ empirically affects the training, please refer to Figure 6 -- 11 in the supplementary materials.
> >
> > ### **References:**
> >
> > [1] Karimireddy et al., SCAFFOLD: Stochastic Controlled Averaging for Federated Learning, ICML 2020
> >
> > [2] Murata and Suzuki, Bias-variance reduced local SGD for less heterogeneous federated learning, ICML 2021
> >
> > [3] Mitra et al., Linear Convergence in Federated Learning: Tackling Client Heterogeneity and Sparse Gradients, NeurIPS 2021
> >
> > [4] Yun et al., Minibatch vs Local SGD with Shuffling: Tight Convergence Bounds And Beyond, ICLR 2022
> >
> > [5] Guo et al., Hybrid Local SGD for Federated Learning with Heterogeneous Communications, ICLR 2022

---

> > > ### Author Response · Authors · 2022-11-13
> > > **Response to Reviewer Moju [3/4]**
> > >
> > > > **[P3]** The authors miss comparing and contrasting with FedNova by Wang et.al., which especially takes care of the case where different clients undergo different number of rounds along with the flexibility of running different local solvers. It would be good to add comparisons with FedDyn and FedOPT which are highly performant baselines.
> > >
> > > Thanks for your question regarding the comparison of other baselines. As we mention in conclusion, it is interesting to explore anchor sampling in the other scenarios of FL. When all clients perform the same number of local updates, FedNova is equivalent to FedAvg. It is worth noting that in our work, clients in the miner group perform **the same number of iterations (a.k.a. local updates)** rather than the same number of epochs. As for various local updates among miners, objective inconsistency occurs and degrades a model's performance [1, 2]. We believe it is a promising direction to tackle the challenge with the help of anchor sampling. Since a recent study [1] points out that FedNova cannot settle objective inconsistency, it may not work when simply applying the normalization techniques to FedAMD.
> > >
> > > Here comes a comparison with FedDyn [3]: (Gradient complexity ($\times 10^5$ samples), Communication Costs ($\times 32$ Mbits), and Rounds measure when the model achieves 75\% test accuracy; Accuracy (%) shows the test accuracy after 400 rounds)
> > >
> > > - 20 clients:
> > >
> > >     |Method|Gradient Complexity |Communication Costs|Round|Accuracy|
> > >     |------|:-----------------:|:-----------------:|:---:|:----------------:|
> > >     |FedDyn|43.0|597.2|335|78.2|
> > >     |FedAMD (constant)|35.5|489.8|259|80.6|
> > >
> > > - 40 clients:
> > >
> > >     |Method|Gradient Complexity |Communication Costs|Round|Accuracy|
> > >     |------|:-----------------:|:-----------------:|:---:|:----------------:|
> > >     |FedDyn|67.1|930.6|261|80.4|
> > >     |FedAMD (constant)|55.0|776.3|209|82.3|
> > >
> > > - 100 clients:
> > >
> > >     |Method|Gradient Complexity |Communication Costs|Round|Accuracy|
> > >     |------|:-----------------:|:-----------------:|:---:|:----------------:|
> > >     |FedDyn|161.9|2246.8|252|81.8|
> > >     |FedAMD (constant)|153.9|2147.8|229|83.4|
> > >
> > > Based on multiple attempts, FedDyn is with the best hyper-parameter setting, i.e., learning rate = 0.01; weight decay = 1e-3; alpha_coef = 0.01. As we can see, our proposed algorithm always outperforms FedDyn. Furthermore, FedOPT [4] is an algorithm for privacy-preserving, which is irrelevant to our research. These two works are complementary, where utilizing the gradient encryption does not harm FedAMD's convergence property.
> > >
> > > ### **References:**
> > >
> > > [1] Mitra et al., Linear Convergence in Federated Learning: Tackling Client Heterogeneity and Sparse Gradients, NeurIPS 2021
> > >
> > > [2] Wang et al., Tackling the Objective Inconsistency Problem in Heterogeneous Federated Optimization, NeurIPS 2020
> > >
> > > [3] Acar et al., Federated Learning Based on Dynamic Regularization, ICLR 2021
> > >
> > > [4] Asad et al., FedOpt: Towards Communication Efficiency and Privacy Preservation in Federated Learning, Appl. Sci. 2020
> > >
> > > ***
> > >
> > > > **[P4]** While a theoretically optimal $p$ is derived, it can't be derived in practice as quantities such as the number of samples on each device and $\sigma^2$ is unknown to the server.
> > >
> > > Thanks for your question regarding the hyperparameters' settings. In Corollary 2, the optimal setting for $p$ is $\frac{1}{c} (\frac{2}{A+2})^{1/A}$, where $c$ is a constant greater than or equal to 1. In Corollary 3, $p = \frac{1}{c} ((1 + \frac{A+1}{2\mu^2}) - \sqrt{(\frac{A+1}{2\mu^2})^2 + \frac{A}{\mu^2}})^{1/A}$ can outcome the optimal solution. Both settings **do not** rely on the number of local samples and $\sigma^2$.
> > >
> > > We guess you mean the setting of $b$, which does depend on the number of local samples and $\sigma^2$. In fact, the server needless to know the number of local samples and $\sigma^2$. In finite-sum cases, clients in the anchor group are required to undergo all local samples. In online learning, the server can gradually tune up $b$ to obtain a model with the desired loss.

---

> > > > ### Author Response · Authors · 2022-11-13
> > > > **Response to Reviewer Moju [4/4]**
> > > >
> > > > > **[P5]** For the variance reduction phase, it requires additional storage of the past gradient which is the same size as that of the model. This can be particularly prohibitive with larger model and resource constrained devices.
> > > >
> > > > Thanks for your question regarding the memory consumption of the clients in the miner group. To the best of our knowledge, there is no algorithm simultaneously bearing training-efficient and memory-efficient. Compared to vanilla SGD, Adam [1] significantly improves the training efficiency, but it requires two additional model-like vectors for caching. Even if we use model compression techniques such as Octo [2], it more or less affects the training efficiency. In a nutshell, we can utilize FedAvg to obtain the desired model under the resource-constrained devices, but it requires more communication rounds. To get a feasible model faster, we should improve the devices' capacities so that we can utilize more space to save the training wall-clock time. For instance, SCAFFOLD [3], FedDyn [4], and BVR-L-SGD [5] achieve faster convergence rates than FedAvg, but they require all clients to utilize additional memory to store extra model-like vectors.
> > > >
> > > > ### **References:**
> > > >
> > > > [1] Kingma and Ba, Adam: A Method for Stochastic Optimization, ICLR 2015
> > > >
> > > > [2] Zhou et al., Octo: INT8 Training with Loss-aware Compensation and Backward Quantization for Tiny On-device Learning, ATC 2021
> > > >
> > > > [3] Karimireddy et al., SCAFFOLD: Stochastic Controlled Averaging for Federated Learning, ICML 2020
> > > >
> > > > [4] Acar et al., Federated Learning Based on Dynamic Regularization, ICLR 2021
> > > >
> > > > [5] Murata and Suzuki, Bias-variance reduced local SGD for less heterogeneous federated learning, ICML 2021
> > > >
> > > > ***
> > > >
> > > > > **[P6]** The convergence results while improving in terms of $\epsilon$-accuracy of the error, is substantially worse as compared to other baselines in terms of dependence on the total number of clients. While other algorithms scale inversely with respect to the total number of clients, FedAMD scales linearly with the number of clients in the setup. Unless, the number of selected clients is always a function of the total number of clients, it seems the performance of FedAMD gets worse with increasing number of clients, if the number of clients selected per round is constant. It is also worth noting the performance benefit comes at the expense of additional assumption 3.
> > > >
> > > > Thanks for your question regarding the theoretical results. As the number of workers $M$ is in the numerator of the convergence rate, we agree that FedAMD requires more communication rounds when the number of participants $A$ is fixed and $M$ gets larger. However, we **do not** agree that the result is worse than other baselines supporting partial clients (see Table 1). In optimization, it is widely acknowledged that $\epsilon \rightarrow 0$, meaning that we mainly focus on **the order of $\epsilon$**. In other words, it is meaningless to discuss the impact of $M$ when we have achieved the improvement in terms of $\epsilon$, i.e., from $\frac{1}{\epsilon^2}$ (FedAvg and SCAFFOLD) and $\frac{1}{\epsilon^{3/2}}$ (BVR-L-SGD [3] and VR-MARINA [2]) to $\frac{1}{\epsilon}$ (our proposed algorithm).
> > > >
> > > > It is worth noting that Assumption 3 only benefits variance-reduced algorithms to generate a tighter bound. Therefore, the existence of Assumption 3 does not affect the convergence property of FedAvg, i.e., $O(\frac{1}{\epsilon^2})$. Also, here is an example that supports Assumption 3: Tewari and Chaudhuri [1] show that some common-used loss functions (e.g., cross-entropy loss) are Lipschitz smooth. With a widely-adopted assumption on the smoothness of local objective functions (i.e., Assumption 1), we can obtain a form of Assumption 3. As mentioned in the paper, MARINA [2] adopts Assumption 3 as well, while some recent works [3, 4, 5] make a Lipschitz-smooth assumption on the loss functions such that we can derive Assumption 3 (For the reason why a Lipschitz-smooth assumption on a loss function is equivalent to Assumption 3, you can refer to **Variance Reduction in Section 3.1 in [3]**).
> > > >
> > > > ### **References:**
> > > >
> > > > [1] Tewari and Chaudhuri, Generalization error bounds for learning to rank: Does the length of document lists matter?, ICML 2015
> > > >
> > > > [2] Gorbunov et al., MARINA: Faster non-convex distributed learning with compression, ICML 2021
> > > >
> > > > [3] Murata and Suzuki, Bias-variance reduced local SGD for less heterogeneous federated learning, ICML 2021
> > > >
> > > > [4] Das et al., Faster Non-Convex Federated Learning via Global and Local Momentum, UAI 2022
> > > >
> > > > [5] Elgabli et al., FedNew: A Communication-Efficient and Privacy-Preserving Newton-Type Method for Federated Learning, ICML 2022
> > > >
> > > > ***
> > > >
> > > > We sincerely hope our explanation helps you better understand our work. We look forward to your re-evaluations of our work. Feel free to let me know if you have any questions. Thanks a lot.

---

> > > > > ### Comment · Reviewer_Moju · 2022-12-09
> > > > > **Thanks for your rebuttal!**
> > > > >
> > > > > "In optimization, it is widely acknowledged that ϵ→0, meaning that we mainly focus on the order of ϵ."
> > > > >
> > > > > Bounds, where M is of the order or larger than that of $O(1/\epsilon)$, M will dominate instead of $\epsilon$. The claimed improvement is at the cost of significantly worse dependence on M and assuming more in terms of assumptions. The claim that for optimization algorithms, it is sufficient to look at the dependence on $\epsilon$ is summarily false. Especially in FL, where M can potentially be in the order of millions.
> > > > >
> > > > > "To the best of our knowledge, there is no algorithm simultaneously bearing training-efficient and memory-efficient."
> > > > >
> > > > > Unfortunately, most FL use cases especially cross-device use cases involve severe restrictions on memory and compute, which needs to be kept in mind while designing algorithms.
> > > > >
> > > > > Thanks to the rebuttal. Some of my concerns are resolved, but my concerns regarding the worse dependence on the total number of clients and the memory efficiency still remain. Having said that, I will raise my score, but I feel the paper needs more work in terms of theory.

---

> > > > > > ### Author Response · Authors · 2022-12-11
> > > > > > **Response to Reviewer Moju's Post-Rebuttal Updates**
> > > > > >
> > > > > > Dear Reviewer Moju,
> > > > > >
> > > > > > Thanks a lot for your follow-up comments and for raising the score.
> > > > > >
> > > > > > From your response, it seems you are still unsatisfied with our work, thereby voting for borderline reject. To address your concerns, we would like to clarify further the disputes on the convergence rate and the contribution of our work.
> > > > > >
> > > > > > ***
> > > > > >
> > > > > > > **[P1]** Bounds, where M is of the order or larger than that of $O(1/\epsilon)$, M will dominate instead of $\epsilon$. The claimed improvement is at the cost of significantly worse dependence on M and assuming more in terms of assumptions. The claim that for optimization algorithms, it is sufficient to look at the dependence on $\epsilon$ is summarily false. Especially in FL, where M can potentially be in the order of millions.
> > > > > >
> > > > > > Thanks for your question concerning the convergence rate. You are making a very interesting argument that $\epsilon \geq 1/M$, where $M$ is in the order of millions. If we achieve $\min_{t \in [T]} \|\nabla F(\tilde{x}_t)\|_2^2 \leq 1/M$ under non-convex objectives, then we obtain the lower bound on $F(\tilde{x}_t) - F_*$ according to Equation (144):
> > > > > >
> > > > > > $F(\tilde{x}\_t) - F\_* \geq \frac{1}{4L} (1 + \frac{2M}{Ap} \sqrt{1-p^A})^{-1} (1 - p^A) \cdot \frac{1}{M}$
> > > > > >
> > > > > > where the learning rate $\eta_s$ and $\eta_l$ follow the setting in Theorem 2. In optimization work, $M$ is a system parameter, which can be arbitrarily large rather than infinitely large. Therefore, **by no means does the model $\tilde{x}_t$ approach the optimal solution** because there is always a constant gap.
> > > > > >
> > > > > > Furthermore, **only when all clients participate in the training** do the algorithms achieve the convergence rate inversely with respect to the number of clients $M$. However, when $M$ is in the order of millions, full client participation rarely happens in the real world.
> > > > > >
> > > > > > As for the algorithm better than FedAvg under partial client participation, their convergence results contain the terms of which $M$ is in the numerator SCAFFOLD under  (i.e., $(\frac{M}{A})^{2/3} \frac{1}{\epsilon}$) [1], FedDyn (i.e., $\frac{M}{A} \frac{1}{\epsilon}$ with $\sigma=0$ in Bounded Noise) [2], and EF-21/EF-BV (i.e., $\frac{M}{A} \frac{1}{\epsilon}$ using a full gradient for each local update) [3, 4].
> > > > > >
> > > > > > ### **References:**
> > > > > >
> > > > > > [1] Karimireddy et al., SCAFFOLD: Stochastic Controlled Averaging for Federated Learning, ICML 2020
> > > > > >
> > > > > > [2] Acar et al., Federated Learning Based on Dynamic Regularization, ICLR 2021
> > > > > >
> > > > > > [3] Richtárik et al., "EF21: A new, simpler, theoretically better, and practically faster error feedback," NeurIPS 2021
> > > > > >
> > > > > > [4] Condat et al., "EF-BV: A Unified Theory of Error Feedback and Variance Reduction Mechanisms for Biased and Unbiased Compression in Distributed Optimization," NeurIPS 2022
> > > > > >
> > > > > > ***
> > > > > >
> > > > > > > **[P2]** Unfortunately, most FL use cases especially cross-device use cases involve severe restrictions on memory and compute, which needs to be kept in mind while designing algorithms.
> > > > > >
> > > > > > Thanks for your question. Our work focuses on **how to accelerate the federated learning and theoretically establish an improved convergence rate**, and it is not our concern whether a large model is able to train in a tiny device. Although the proposed algorithm occupies twice the computation memory as FedAvg, the required number of communication rounds is theoretically $\epsilon$-time as that of FedAvg, which is a significant improvement. In addition, there are numerous ways to cope with the memory challenge. For example, model compression (e.g., quantization [1, 2] and pruning [3, 4, 5]) can effectively diminish the model size such that our approach can be applied to the extreme case you mention.
> > > > > >
> > > > > > ### **References:**
> > > > > >
> > > > > > [1] Lin et al., "On-Device Training Under 256KB Memory," NeurIPS 2022
> > > > > >
> > > > > > [2] Zhou et al., "Octo: INT8 Training with Loss-aware Compensation and Backward Quantization for Tiny On-device Learning," ATC 2021
> > > > > >
> > > > > > [3] Liu et al., "Rethinking the value of network pruning," ICLR 2018
> > > > > >
> > > > > > [4] Zhu and Gupta, "To prune, or not to prune: exploring the efficacy of pruning for model compression"
> > > > > >
> > > > > > [5] Sui et al., "CHIP: CHannel independence-based pruning for compact neural networks," NeurIPS 2021
> > > > > >
> > > > > > ***
> > > > > >
> > > > > > Thank you again for engaging in the discussion and for your insightful questions and suggestions! We sincerely hope that our responses above have clarified your doubts and they will result in a further increase in your score.

---

> ### Author Response · Authors · 2022-11-26
> **Thank you and look forward to your post-rebuttal updates**
>
> Dear Reviewer Moju,
>
> First and foremost, we are grateful for your commitment to reviewing our paper.
>
> As you know, we have submitted a response regarding your concerns in the first-phase review, but we haven't heard from you yet. We understand that you may not be available to read our response thoroughly, so we summarize each point of our response in one or two sentences:
>
> - **[P1]** According to the existing work like SCAFFOLD and FedDyn, the initialization step (Line 1 - 6 in Algorithm 1) should be treated as an **input** of FedAMD, where the clients can provide such information to launch the training. Besides, the gradient inversion attack is **unlikely to happen** because the cached gradient of each client is calculated with a large dataset, e.g., the entire local training set.
> - **[P2]** We provide a solution that alters the aggregation scheme such that the proposed FedAMD fits the scenarios of imbalanced training sets and/or client-specific weights. In practice, $b'$ should be set as other existing works do, and $b$ should be very large in online learning or the size of the entire training sets in finite-sum cases.
> - **[P3]** FedNova alleviates objective inconsistency caused by system heterogeneity (i.e., the numbers of local updates vary among clients), **which is not a target question of our paper**. As for the comparison with FedDyn, we conduct an empirical study and demonstrate **the superiority of FedAMD in all roundness**.
> - **[P4]** The constant probability $p$ is **not** related to the number of samples on each device and $\sigma^2$.
> - **[P5]** There is **no** such algorithm that possesses optimal training efficiency and minimum memory consumption at the same time. Existing works like SCAFFOLD and FedDyn also require an additional memory cost to improve the convergence performance.
> - **[P6]** Our algorithm may be worst than other baselines **only when the targeted loss is large (or improperly set)**, which makes no sense in the optimization works because they aim at finding a model with the loss **approaching zero**. Additionally, some real-world cases satisfy Assumption 3, and many state-of-the-art works also use this assumption.
>
> We hope our explanation makes you clear about our work and some basic concepts of federated optimization. If you are still not convinced to accept this work, please show us your concerns, and we will try our best to solve them. We are **eager** to receive your post-rebuttal updates.
>
> Best,
>
> Authors of Paper 3563 (**Anchor Sampling for Federated Learning with Partial Client Participation**)

---

> ### Author Response · Authors · 2022-11-28
> **A gentle reminder**
>
> Dear Reviewer Moju,
>
> As the second-phase discussion will end in **two weeks**, we are yet to receive any updates from you. Could you please kindly take a look at our response? We are happy to discuss our work with you through OpenReview platform. Thanks a lot.
>
> Best,
>
> Authors of Paper 3563 (**Anchor Sampling for Federated Learning with Partial Client Participation**)

---

> ### Author Response · Authors · 2022-12-01
> **Your voice matters**
>
> Dear Reviewer Moju,
>
> Thanks a lot for your insightful comments in the first-round review.
>
> We have posted our response for over two weeks, but we are still waiting to hear you. We would like to know if you have read our response. Does it fully address your concerns? Does our work deserve acceptance? Feel free to write something through the OpenReview platform. We will highly appreciate your prompt feedback.
>
> Best,
>
> Paper 3563 Authors

---

> ### Author Response · Authors · 2022-12-02
> **We hope you can read our rebuttal and give a response!**
>
> Dear Reviewer Moju,
>
> It is the last ten days that we can discuss with each other through the OpenReview platform. Could you please kindly take a look at our response? It would be highly appreciated if you could provide follow-up comments. Thanks a lot.
>
> Lastly, we sincerely wish you a nice weekend.
>
> Best,
>
> Paper 3563 Authors

---

> ### Author Response · Authors · 2022-12-05
> **Look forward to your reponse as the discussion period ends in one week**
>
> Dear Reviewer Moju,
>
> As the discussion will end in one week, we treasure the final chance to discuss our work with you. Could you feel free to go through our rebuttal and drop some notes? Thanks a lot.
>
> Best,
>
> Paper 3563 Authors

---

> ### Author Response · Authors · 2022-12-06
> **Do you have more comments?**
>
> Thanks again for your efforts and insightful suggestions in reviewing. As the discussion ends shortly, do you mind sparing some time to read our response and leave your comments (or acknowledge you have read it)? It won't take you too much time.
>
> Best,
> Authors

---

> ### Author Response · Authors · 2022-12-09
> **Your response is always welcome and highly appreciated**
>
> Dear Reviewer Moju,
>
> We hope this kind reminder finds you well.
>
> Our response has been posted for over three weeks. Unfortunately, we are yet to receive your follow-up feedback. We understand everyone is very busy when the year comes to an end. To make your time more efficient, we are happy to summarize our response based on your first-phase comments: (*W refers to the weaknesses your comments listed at https://openreview.net/forum?id=VLnODGVVAsL&noteId=ntAG4Ck5qz; P indicates the points of our response at [1/4] https://openreview.net/forum?id=VLnODGVVAsL&noteId=BAfeaZbZv_, [2/4] https://openreview.net/forum?id=VLnODGVVAsL&noteId=kUuotG3Yx6r, [3/4] https://openreview.net/forum?id=VLnODGVVAsL&noteId=UzYhNudKdcG, [4/4] https://openreview.net/forum?id=VLnODGVVAsL&noteId=Nl27djyFTQ.*)
>
> - **[W1, see P1 for details]** According to the existing work like SCAFFOLD and FedDyn, the initialization step (Line 1 - 6 in Algorithm 1) should be treated as an **input** of FedAMD, where the clients can provide such information to launch the training. Besides, the gradient inversion attack is **unlikely** to happen because the cached gradient of each client is calculated with a large dataset, e.g., the entire local training set.
> - **[W2, see P6 for details]** Our algorithm may be worst than other baselines **only when the targeted loss is large (or improperly set)**, which makes no sense in the optimization works because they aim at finding a model with the loss **approaching zero**. Additionally, some real-world cases satisfy Assumption 3, and many state-of-the-art works also use this assumption.
> - **[W3, see P2 for details]** We provide a solution that alters the aggregation scheme such that the proposed FedAMD fits the scenarios of imbalanced training sets and/or client-specific weights. In practice, $b'$ should be set as other existing works do, and $b$ should be very large in online learning or the size of the entire training sets in finite-sum cases.
> - **[W4, see P3 for details]** FedNova alleviates objective inconsistency caused by system heterogeneity (i.e., the numbers of local updates vary among clients), **which is not a target question of our paper**. As for the comparison with FedDyn, we conduct an empirical study and demonstrate **the superiority of FedAMD in all roundness**.
> - **[W5, see P4 for details]** The constant probability $p$ is **not** related to the number of samples on each device and $\sigma^2$.
> - **[W6, see P5 for details]** There is **no** such algorithm that possesses optimal training efficiency and minimum memory consumption at the same time. Existing works like SCAFFOLD and FedDyn also require an additional memory cost to improve the convergence performance.
>
> Should you have any concerns, please don't hesitate to let us know. If we cannot hear you in the next few days, we believe that our response has fully addressed your concerns. In other words, our work has properly settled all weaknesses you mentioned, and you CANNOT see any deficiencies in our work up to now.
>
> Thanks a lot for your time and efforts.

---

> ### Author Response · Authors · 2022-12-12
> **Final call for your feedback**
>
> Dear Reviewer Moju,
>
> Thanks a lot for your engagement in reviewing our work and for your participation in the discussion.
>
> We hope our responses have fully addressed your concerns. As your future comments on our work may not be visible to the authors and the public, we sincerely hope you can positively recommend our work to AC. If you have further comments/suggestions, don't hesitate to let us know. It seems that we still have some time to discuss the work.
>
> Best,
>
> Authors of Paper 3563 (**Anchor Sampling for Federated Learning with Partial Client Participation**)

---

### Official Review · Reviewer_B281 · 2022-10-27

**Confidence:** 3
**Correctness:** 4
**Technical Novelty And Significance:** 3
**Empirical Novelty And Significance:** 3
**Recommendation:** 6

**Clarity, Quality, Novelty And Reproducibility:**

The paper is well-written and provides a novel algorithm. I could not check proofs carefully.

**Strength And Weaknesses:**

Strength: The paper proposes a novel way to enable clients to share their gradients and use this information to update the model locally. The paper proves the convergence of the proposed algorithm and provide convincing experimental results to validate the theoretical analysis provided in the paper.

Weaknesses:

- Using the proposed algorithm, the server has to store a caching gradient matrix whose dimensions scales with the number of clients. When the number of clients is so large, to implement the proposed algorithm a server with a very large memory is required.

- It is beneficial for the paper if the convergence rate of the proposed FedAMD is compared to FedAvg to see what are the benefits of FedAMD compared to FedAvg in terms of convergence. FedAMD requires larger memory from the server side and it is interesting to see if FedAMD uses this memory successfully to obtain better convergence rate than that of FedAvg.

- From reading the Algorithm 1 on page 4, the role of cached gradients in updating models is not clear. It would be great if Algorithm 1 can be revised such that the role of cached gradient become obvious.

**Summary Of The Paper:**

The paper proposes a new federated learning algorithm which allows clients to perform their updates on the model using the gradient of the model obtained by other clients. This involves data samples stored on other clients memory in updating the model locally by a client. The paper provides convergence analysis for the proposed algorithm and examines its performance through experiments.

**Summary Of The Review:**

In summary, the paper proposes a new federated learning algorithm at the cost of the need for a server with a large memory capacity. The paper proves the convergence of the algorithm. However, the paper lacks a detailed comparison of the proposed algorithm's convergence with the convergence of FedAvg.

---

> ### Author Response · Authors · 2022-11-13
> **Response to Reviewer B281**
>
> We would like to appreciate your insightful comments on our work. Based on your comments, we would like to clarify your concerns as follows.
>
> ***
>
> > **[P1]** Using the proposed algorithm, the server has to store a caching gradient matrix whose dimensions scales with the number of clients. When the number of clients is so large, to implement the proposed algorithm a server with a very large memory is required.
>
> Thanks for pointing out the shortages in our proposed algorithm. In Algorithm 1, the server stores the caching gradient $v = \{v^{(1)}, \dots, v^{(M)}\}$ with the size of $M$, the number of the clients. Apparently, the server is not always able to shoulder the storage burden, especially under cross-device settings. Here is a solution that offloads the burden to all clients $i \in [M]$: At round $t$,
>
> - If client $i$ does not participate in the training, or the client acts as a miner, its local caching gradient $v^{(i)}$ remains unchanged. This implies $\wedge^{(i)} = \mathbf{0}$.
> - If client $i$ acts as an anchor, then
>   - calculate $v\_{new}^{(i)} = \nabla f\_i (\tilde{x}\_t, \mathcal{B}\_{i, t})$ using $\mathcal{B}_{i, t} \sim \mathcal{D}_i$ with the size of $b$ (**same as** Line 13 in Algorithm 1)
>   -  communicate the update $\wedge^{(i)} = v_{new}^{(i)} - v^{(i)}$ with the server and indicate the update of caching gradient (**different from** Line 14 in Algorithm 1)
>   - update the local caching gradient $v^{(i)} = v_{new}^{(i)}$.
>
> Therefore, the averaged caching gradient on the server follows the update that $\tilde{g}\_{t+1} = \tilde{g}\_{t} - \frac{1}{M} \sum\_{i \in [M]} \wedge^{(i)}$. By this means, the server no longer maintains the list of the caching gradient $v$. It is worth noting that our revised version discusses it in **Section 3 (Discussion on Massive-Client Settings)**.
>
> ***
>
> > **[P2]** It is beneficial for the paper if the convergence rate of the proposed FedAMD is compared to FedAvg to see what are the benefits of FedAMD compared to FedAvg in terms of convergence. FedAMD requires larger memory from the server side and it is interesting to see if FedAMD uses this memory successfully to obtain better convergence rate than that of FedAvg.
>
> Thanks for your question about the comparison between FedAMD and FedAvg. To the best of our knowledge, Yang et al. [1] provide the tightest bound for FedAvg under non-convex objectives, which requires a total of $O(\frac{K}{A \epsilon^2} + \frac{1}{\epsilon})$ communication rounds and $O(\frac{K^2}{\epsilon^2} + \frac{AK}{\epsilon})$ gradient computations to achieve $\epsilon$-accuracy. In contrast, FedAMD requires a total of $O(\frac{M}{A\epsilon})$ communication rounds and $O(\frac{\sigma^2}{\epsilon^2} + \frac{M K}{\epsilon})$ gradient computations. It is obvious that FedAMD **significantly diminishes** the required communication rounds, but we cannot learn which has efficient computation consumption at first glance because both are at an order of $O(1/\epsilon^2)$. It is noted that [1] implicitly assumes $K \geq \sigma^2$, and therefore, FedAMD **reduces** the computation consumption. As for memory usage on the server, FedAMD always requires more than FedAvg during the training, even if we use an alternative scheme mentioned in the first response (which is twice as FedAvg). It is worth noting that we compare FedAvg and FedAMD under non-convex objectives in **Section 4 (Comparison with FedAvg)**.
>
> ### **Reference:**
>
> [1] Yang et al., Achieving Linear Speedup with Partial Worker Participation in Non-IID Federated Learning, ICLR 2021
>
> ***
>
> > **[P3]** From reading the Algorithm 1 on page 4, the role of cached gradients in updating models is not clear. It would be great if Algorithm 1 can be revised such that the role of cached gradient become obvious.
>
> Thanks for your question regarding the role of cached gradients. As we can see in Line 16 of Algorithm 1, a miner (client $i$) initializes the first update $g_{t, 0}^{(i)}$ with the cached gradients $\tilde{g}_t$, and the subsequent local updates are formed directly or indirectly based on the first update. In specific, the recurrent relation between two successive global models is formulated in Appendix C.
>
> As presented in Equation (73) and Lemma 5, the role of cached gradients in the theoretical analysis is to replace the bound between two local models (FedAvg) with that between two global models (FedAMD). Without the cached gradient, an estimation of the path toward the optimal solution, the local update will be biased to the optimal local solution, which deteriorates the training efficiency.

---

> ### Author Response · Authors · 2022-11-26
> **Thank you and look forward to your post-rebuttal updates**
>
> Dear Reviewer B281,
>
> First and foremost, we are grateful for your commitment to reviewing our paper.
>
> As you know, we have submitted a response regarding your concerns in the first-phase review, but we haven't heard from you yet. We understand that you may not be available to read our response thoroughly, so we summarize each point of our response in one or two sentences:
>
> - **[P1]** To cast off a memory burden of the server, we introduce an alternative way by offloading it to the clients, for which no extra cost is required.
> - **[P2]** We provide a detailed comparison between the proposed FedAMD and FedAvg. For instance, in terms of the convergence rate (or $\epsilon$-accuracy, $\epsilon \rightarrow 0$), the number of communication rounds required by FedAMD (i.e., $O(\frac{M}{A\epsilon})$) is $O(\epsilon)$-time as that by FedAvg (i.e., $O(\frac{K}{A \epsilon^2} + \frac{1}{\epsilon})$).
> - **[P3]** The cached gradient is initialized as the first local update of the miner clients and recursively affects the subsequent local updates. As a result, it determines how the global model changes, and the recurrent relation is formulated in Appendix C.
>
> This letter does not aim to force you to raise your rating and confidence; we want to know anything else we can do to improve our paper. Feel free to let us know if you have any comments. We are **eager** to receive your post-rebuttal updates.
>
> Best,
>
> Authors of Paper 3563 (**Anchor Sampling for Federated Learning with Partial Client Participation**)

---

> ### Author Response · Authors · 2022-11-28
> **A gentle reminder**
>
> Dear Reviewer B281,
>
> As the second-phase discussion will end in **two weeks**, we are yet to receive any updates from you. Could you please kindly take a look at our response? We are happy to discuss our work with you through OpenReview platform. Thanks a lot.
>
> Best,
>
> Authors of Paper 3563 (**Anchor Sampling for Federated Learning with Partial Client Participation**)

---

> ### Author Response · Authors · 2022-12-01
> **Your voice matters**
>
> Dear Reviewer B281,
>
> Thanks a lot for your insightful comments in the first-round review.
>
> We have posted our response for over two weeks, but we are still waiting to hear you. We would like to know if you have read our response. Does it fully address your concerns? Does our work deserve acceptance? Feel free to write something through the OpenReview platform. We will highly appreciate your prompt feedback.
>
> Best,
>
> Paper 3563 Authors

---

> ### Author Response · Authors · 2022-12-02
> **We hope you can read our rebuttal and give a response!**
>
> Dear Reviewer B281,
>
> It is the last ten days that we can discuss with each other through the OpenReview platform. Could you please kindly take a look at our response? It would be highly appreciated if you could provide follow-up comments. Thanks a lot.
>
> Lastly, we sincerely wish you a nice weekend.
>
> Best,
>
> Paper 3563 Authors

---

> ### Author Response · Authors · 2022-12-05
> **Look forward to your reponse as the discussion period ends in one week**
>
> Dear Reviewer B281,
>
> As the discussion will end in one week, we treasure the final chance to discuss our work with you. Could you feel free to go through our rebuttal and drop some notes? Thanks a lot.
>
> Best,
>
> Paper 3563 Authors

---

> ### Author Response · Authors · 2022-12-06
> **Do you have more comments?**
>
> Thanks again for your efforts and insightful suggestions in reviewing. As the discussion ends shortly, do you mind sparing some time to read our response and leave your comments (or acknowledge you have read it)? It won't take you too much time.
>
> Best,
> Authors

---

> ### Author Response · Authors · 2022-12-09
> **Your response is always welcome and highly appreciated**
>
> Dear Reviewer B281,
>
> We hope this kind reminder finds you well.
>
> Our response has been posted for over three weeks. Unfortunately, we are yet to receive your follow-up feedback. We understand everyone is very busy when the year comes to an end. To make your time more efficient, we are happy to summarize our response based on your first-phase comments: (*W refers to the weaknesses your comments listed at https://openreview.net/forum?id=VLnODGVVAsL&noteId=1Ah36DRJdF; P indicates the points at our response in https://openreview.net/forum?id=VLnODGVVAsL&noteId=o4jRoqhj8U.*)
>
> - **[W1, see P1 for details]** To cast off a memory burden of the server, we introduce an alternative way by offloading it to the clients, for which no extra cost is required.
> - **[W2, see P2 for details]** We provide a detailed comparison between the proposed FedAMD and FedAvg. For instance, in terms of the convergence rate (or $\epsilon$-accuracy, $\epsilon \rightarrow 0$), the number of communication rounds required by FedAMD (i.e., $O(\frac{M}{A\epsilon})$) is $O(\epsilon)$-time as that by FedAvg (i.e., $O(\frac{K}{A \epsilon^2} + \frac{1}{\epsilon})$).
> - **[W3, see P3 for details]** The cached gradient is initialized as the first local update of the miner clients and recursively affects the subsequent local updates. As a result, it determines how the global model changes, and the recurrent relation is formulated in Appendix C.
>
> Should you have any concerns, please don't hesitate to let us know. If we cannot hear you in the next few days, we believe that our response has fully addressed your concerns. In other words, our work has properly settled all weaknesses you mentioned, and you CANNOT see any deficiencies in our work up to now.
>
> Thanks a lot for your time and efforts.

---

> ### Author Response · Authors · 2022-12-12
> **Final call for your feedback**
>
> Dear Reviewer B281,
>
> Thanks a lot for your engagement in reviewing our work.
>
> We hope our response has fully addressed your concerns. As your future comments on our work may not be visible to the authors and the public, we sincerely hope you can positively recommend our work to AC. If you have further comments/suggestions, don't hesitate to let us know. It seems that we still have some time to discuss the work.
>
> Best,
>
> Authors of Paper 3563 (**Anchor Sampling for Federated Learning with Partial Client Participation**)

---

### Author Response · Authors · 2022-11-13
**General Response and Appreciation to All Reviewers and ACs**

First and foremost, we would like to show our sincere gratitude and respect to all reviewers and area chairs. In your comments, we see some strong points of our works: (i) Theoretical analysis provides a **non-trivial improvement** compared to the state-of-the-art work (Reviewer B281, Reviewer Moju, Reviewer UWdf); (ii) The experimental results are **convincing** and present the **dominance** of our proposed algorithm (Reviewer B281, Reviewer Moju); (iii) It is the **first** work to analyze the effectiveness of large batches under partial client participation (Reviewer UWdf). We would like to express our special thanks to Reviewer UWdf who carefully checks our proof and affirms its **correctness**.

We also notice that the reviewers raise some concerns or questions. We seriously go through all your comments and make revisions and explanations accordingly. We hope all reviewers and ACs take the revised version and the individual comments into consideration for re-evaluating the paper. In this general response, we highlight what changes have been made in our paper (p.s. The corresponding texts have been marked in blue).

***

**Change #1:** A new paragraph is added to the end of Section 3

> **Discussion on Massive-Client Settings.** A typical example of this scenario is cross-device FL [53]. In this setting, it is not a wise option for the server to preserve all the caching gradients for clients. Therefore, the clients retain their caching gradients while the server keeps their average. Firstly, at $t$-th round, client $i \in [M]$ copies their caching gradient to $(t+1)$-th round, i.e., $v_{t+1}^{(i)} = v_{t}^{(i)}$. For the client $i$ in the anchor group, they will follow Line 13 in Algorithm 1 to update $v_{t+1}^{(i)}$ and push $\wedge_{t}^{(i)} = v_{t+1}^{(i)} - v_{t}^{(i)}$ to the server. After the server receives the updates of all local caching gradients, it performs $v_{t+1} = v_{t} + \frac{1}{M} \sum \wedge_{t}$, where $\wedge_{t}$ aggregates $\wedge_{t}^{(i)}$ where client $i$ is in the anchor group.

Although the current approach to caching the gradients in Algorithm 1 is the most intuitive solution, it aggravates the storage burden on the server, especially when there are massive clients. As a result, we propose an alternative approach that can achieve a similar effect and lessen the burden on the server.

***

**Change #2:** Remove the training loss from Figure 1.

Due to the limited space, we have to move some contents to the supplementary material. The readers can find the training loss against the communication rounds in Figure 3. Note that the training loss originally in Figure 1 was placed in Figure 3 (j) -- (l).

***

Again, thanks a lot for your time and efforts. We sincerely hope that the explanations and the revisions help you better understand our work and also motivate your future research.

---

> ### Author Response · Authors · 2022-11-19
> **A New Update in Appendix F**
>
> We would like to highlight our **Change #3** in our revision:
>
> **Change #3:** Explain the experimental results, and add EMNIST dataset for empirical evaluation in Appendix F (Additional Experiments)
>
> We revise the experiment in our supplementary materials to strengthen our theoretical findings. Firstly, we reorganize the entire section and introduce the connections among figures and settings. Besides, we point out some interesting discoveries in the experiments and provide justifications for why it happens. Reviewer UWdf mentioned that it is insufficient to support our statements with one single dataset. Therefore, we utilize EMNIST to train a non-convex objective to show the effectiveness of our proposed algorithm and enhance our work from an empirical perspective.

---

### Author Response · Authors · 2022-11-21
**Look forward to your follow-up comments**

Dear Reviewers and Area Chairs,

We hope this letter finds you well.

First and foremost, we are grateful for your time and efforts in reading our paper and providing insightful and constructive feedback.

As we know, the stage-1 discussion has ended, and we can no longer revise the paper. Despite no further comments received since the review was released on Nov. 4, we are confident that our current revision and responses address your concerns. Since our work only receives three official comments, all your evaluations matter a lot. We look forward to your active involvement in the stage-2 discussion. Please don't hesitate to let us know if you have any questions. It is highly encouraged and appreciated if you can leave the notes point by point to indicate you have read our responses, even though you don't intend to change your rating and confidence.

Many thanks for your time and efforts.

Best,

Authors

---

### Author Response · Authors · 2022-12-13
**Closing Remarks for Rolling-based Discussion**

First and foremost, it is our great honor to receive your comments on our work. Special thanks to Reviewer Moju, who participates in the second-phase discussion. We are confident that the concerns from Reviewer B281 and Reviewer UWdf have been completely addressed because they do not raise more questions for us. Since the discussion is ending soon, we are here to summarize the main points from the reviewers and our corresponding responses:

***

- **Reviewer B281** appreciates our work for the theoretical contribution and the dominant experimental results. At the same time, the reviewer is concerned about how the server caches a large number of gradients, but we propose a solution that offloads the burden to each client. Besides, we present a comprehensive comparison with FedAvg and demonstrate the superiority of our work in terms of theoretical convergent performance.

***

- **Reviewer Moju** highlights our work for non-trivial theoretical improvement and comprehensive empirical studies. Besides, the reviewer mentions most of our response has addressed his/her concerns except two needing more clarification. First, the reviewer argues that the number of clients $M$ can be in the order of millions. We show that, no matter how $M$ is chosen, the final output model is never equivalent to the optimal one. Second, the reviewer indicates that the algorithm does not support large model training on the tiny device. This concern is out of the scope of our work because we mainly focus on how to theoretically accelerate federated learning and establish an improved convergence rate. However, we introduce some possible approaches that leverage state-of-the-art techniques to settle the issue without harming the performance of the proposed algorithm.

***

- **Reviewer UWdf** has carefully checked our proof details and shows the correctness of our work. Additionally, he/she believes this is the first work to analyze the effectiveness of large batches under partial client participation. However, the reviewer thinks our experimental analysis needs to be revised in the first draft. Then, we polish the paper to enhance our empirical studies and include the model training on EMNIST digits. The results also demonstrate the superiority of the proposed algorithm FedAMD compared to other baselines.

***

We believe anchor sampling will be a new trend in federated learning, which properly utilizes large batches to accelerate the model training. We also hope this idea can be applied to other fields and facilitate future research. It would be highly appreciated if the paper could be recommended for acceptance.

Again, thank all reviewers and area chairs for your hard work, not just evaluating our work but contributing to the ICLR community.

---

### Decision · Program_Chairs · 2023-01-20

**Decision:**

Reject

**Justification For Why Not Higher Score:**

see above

**Justification For Why Not Lower Score:**

see above

**Metareview: Summary, Strengths And Weaknesses:**

After careful consideration of the paper, the reviews, and the discussion points on this paper, it has been decided that the paper is not yet ready for publication in ICLR. The main issue is that none of the reviewers were enthusiastic about the paper, relatively. This meas and there were just too many other papers that were much more highly considered (and rated) by reviewers than the give paper, and ICLR has a low acceptance rate. On my own reading of the paper, I think the writing can be improved as well --- for example, even reading through the abstract, there are things like the use of the word "disjoints", which is rarely used a a verb, a better word would just be to use "separates." There are a variety of such improvements that would help throughout the paper, but that wasn't a key reason for the final decision. Some of the key reasons for this decision are (and ways you can address this in the next version to a future conference):

* Requirement for full client participation
* Remaining questions about $\epsilon$-accuracy compared to baselines, this needs
to be very clear in the initial submission
* The issue of imbalance needs to be made much more clear in the initial paper submission, and the sensitivity should be tested and evaluated.
* Comparisons with methods such as FedNova needs to be more clearly stated and done in the original submission.
* Improvements in convergence proofs,
* Could some improved analysis about the influence of initial model

It is recommended to consider all of the reviewer comments in the next version of the paper and submit to the next conference to ensure that these questions do not arise again.



**Summary Of Ac-Reviewer Meeting:**

see above